# Prediction-Powered Adaptive Inference with Pretrained AI Models for Contextual Bandits

**Gabriel Sargent** [1]  **Will Wei Sun** [2]  **Zhengwu Zhang** [1]  **Yufeng Liu** [3]

## Abstract

In adaptive experiments, statistical inference is essential for reliable decision-making and scientific discovery. Often in these settings, collecting labeled data is expensive, but decision-makers have access to large unlabeled datasets and strong pretrained AI models that can predict outcomes. Effectively leveraging these predictions in online experiments poses fundamental challenges: AI predictions may be inaccurate, and data collected under adaptive policies are inherently non-i.i.d., invalidating classical inference techniques. To address these challenges, we propose a Prediction-Powered Adaptive Inference (PPAI) estimator that integrates unlabeled data, predicted labels, and adaptively collected labeled data. We establish asymptotic normality of the PPAI estimator under mild conditions on the data-collection policy, enabling valid confidence intervals and hypothesis tests for a broad class of Z-functionals. The estimator incorporates a data-driven tuning mechanism that weights AI predictions according to their informativeness, guaranteeing that the asymptotic variance is no worse than that of the labeled-only baseline, and is strictly smaller when predictions are informative. Numerical experiments and a movie recommendation application further support the theory, illustrating efficiency gains with informative AI predictions and robust performance with inaccurate predictions.

[1]Department of Statistics and Operations Research, University of North Carolina at Chapel Hill, Chapel Hill, NC, USA [2]Department of Quantitative Methods, Purdue University, West Lafayette, IN, USA [3]Department of Statistics, University of Michigan, Ann Arbor, MI, USA. Correspondence to: Yufeng Liu <yufliu@umich.edu>, Will Wei Sun <sun244@purdue.edu>.

*Proceedings of the $43^{rd}$ International Conference on Machine Learning*, Seoul, South Korea. PMLR 306, 2026. Copyright 2026 by the author(s).

## 1. Introduction

Adaptive experimental designs are used in a wide variety of modern decision-making problems, like clinical trials (Varatharajah & Berry, 2022; Villar et al., 2015), mobile health (Liao et al., 2020; Yom-Tov et al., 2017), recommendation systems (Tang et al., 2014; Li et al., 2010; Zhu & Roy, 2023; McInerney et al., 2018), and education technology (Shaikh et al., 2019; Liu et al., 2014), because they enable agents to adjust decision-making strategies in real time. For instance, clinicians tailor treatments in response to observed health outcomes, recommender systems adjust recommendations based on recent engagement, and tutoring systems dynamically alter assignments based on student performance.

While the primary objective in these settings is often to optimize outcomes (e.g., minimize regret), an equally important goal is to support *statistical inference* about underlying design parameters. Inference enables scientific discovery and principled decision-making by answering questions such as "What is the average treatment effect?" or "Do outcomes differ across subpopulations?" through valid confidence intervals and hypothesis tests. Moreover, inferential insights can guide the design of future experiments and improve downstream decision-making (Shi et al., 2021; 2024; Zhang et al., 2021; 2022; Simchi-Levi & Wang, 2025).

A central bottleneck is that reliable inference typically requires substantial amounts of *labeled* data, yet collecting labels in adaptive experiments can be expensive, slow, or ethically constrained. At the same time, many applications provide abundant *unlabeled* covariates and access to externally trained AI models that can generate cheap outcome predictions. For instance, in clinical trials one may have rich covariates from electronic health records (EHRs) and pretrained models that predict patient outcomes (Wornow et al., 2025). Similarly, in recommendation systems, models trained on historical logs can predict clicks, and in online education, models can predict student performance. These predictions are not substitutes for experimental outcomes, but they constitute valuable *auxiliary information* that could, in principle, improve inferential efficiency.

This motivates a central and technically nontrivial question

of this project: *how can adaptively collected experimental data be combined with AI-generated predictions from potentially inaccurate external models to enable valid and efficient statistical inference?* Treating AI-generated predictions as ground truth can introduce large bias, while ignoring them discards potentially substantial information. Addressing adaptivity and prediction bias simultaneously therefore requires new estimator construction and new asymptotic analysis.

We consider a contextual bandit setting, in which the decision-maker sequentially observes a context $\boldsymbol{X}_t$ (e.g., covariates describing a patient), selects an action $A_t$ (e.g., a treatment) based on a bandit policy, and receives a reward $Y_t$ (e.g., a measure of the patient's health outcome), which depends on $\boldsymbol{X}_t$ and $A_t$. In addition, the decision-maker has a large pool of unlabeled contexts $\{\tilde{\boldsymbol{X}}_i\}_{i=1}^N$ and a pretrained AI model $f_a(\cdot)$ predicting the rewards of taking an action $a$ for various contexts. The inferential goal is to estimate a parameter $\boldsymbol{\theta}_a^*$ of a working reward function for some action $a$ and construct its confidence interval.

To address this problem, we propose a prediction-powered adaptive estimator $\hat{\boldsymbol{\theta}}_{a,T}$ whose form is explicitly designed to accommodate adaptively collected data $\{(\boldsymbol{X}_t, A_t, Y_t)\}_{t=1}^{T-1}$ (i.e. the experimental history), unlabeled contexts $\{\tilde{\boldsymbol{X}}_i\}_{i=1}^N$, and potentially inaccurate AI predictions $\{f_a(\tilde{\boldsymbol{X}}_i)\}_{i=1}^N$. At each time $T$, the decision-maker observes the current context $\boldsymbol{X}_T$, updates the estimate $\hat{\boldsymbol{\theta}}_{a,T}$, selects an action $A_T$ based on the bandit policy, and receives a reward $Y_T$, which is then added to the labeled dataset (see Figure 1).

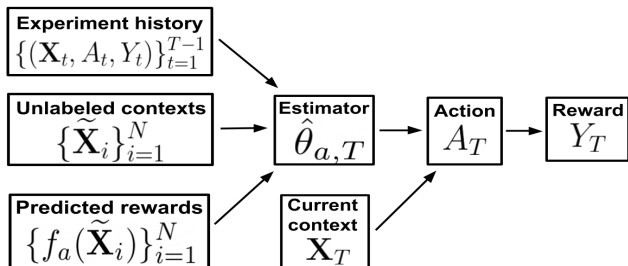

*Figure 1.* A high-level illustration of our online prediction-powered estimation procedure. At time $T$, our estimator for $\boldsymbol{\theta}_a^*$ uses adaptively collected labeled data from times $t = 1, \ldots, T-1$ and predicted rewards for selecting action $a$ with the unlabeled contexts. The decision-maker then selects an action and receives a reward, which is then added to the labeled dataset.

On the theoretical side, we establish asymptotic normality of our estimator under mild assumptions on the bandit policy used to collect the data (Theorem 3.7). The proof relies on a martingale central limit theorem tailored to a prediction-powered estimating equation under adaptivity, a setting not covered by existing results. With this result, one can construct valid confidence intervals and hypothesis tests for a broad class of Z-functionals of the reward distribution. Additionally, our estimator adapts to the quality of the AI

predictions by employing a data-driven tuning procedure that weights predictions with a parameter $\lambda \in \mathbb{R}$ based on their accuracy. We derive the optimal tuning parameter $\lambda^*$ and a consistent estimate $\hat{\lambda}_T$ (Theorem 3.8). By weighting according to $\hat{\lambda}_T$, our estimator achieves the minimal asymptotic variance (Corollary 3.10). We compare our estimator to an analogous one that only uses labeled data, which we call the "labeled-only baseline." If the AI prediction model $f_a$ is strong, our estimator leverages the predictions and unlabeled data to achieve a much smaller asymptotic variance than the labeled-only baseline. On the other hand, if $f_a$ is poor, our estimator downweights the predictions, reverting to this baseline. Thus our estimator incorporates the predictions safely, as its asymptotic variance is never larger than the labeled-only baseline and is much smaller if the predictions are informative.

Our **contributions** can be summarized as follows.

1. (**Method**) We propose a prediction-powered adaptive inference framework for contextual bandits that integrates unlabeled contexts and black-box AI predictions while preserving valid inference under adaptive data collection and misspecification. This is not a straightforward combination of existing prediction-powered and adaptive inference methods: naïvely applying prediction-powered estimators to adaptively collected data breaks unbiasedness, while directly treating predictions as additional labels in adaptive inference also induces bias. Our approach resolves these challenges through a new estimator design tailored to both data adaptivity and prediction bias.

2. (**Theory**) We establish asymptotic normality for general Z-functionals of the reward distribution under mild regularity. In adaptive experiments, classical i.i.d. asymptotic analysis is no longer applicable, leading to invalid confidence intervals (Chen et al., 2021; Zhang et al., 2021; Guo & Xu, 2025). To remedy this, we incorporate inverse propensity weighting into the prediction-powered estimating equation to correct for the bias induced by data adaptivity (see Section 3). This construction enables us to establish asymptotic normality of the resulting estimator via a martingale central limit theorem.

3. (**Robustness and Efficiency**) We propose a data-driven tuning rule $\hat{\lambda}_T$ that adaptively weights AI predictions according to their informativeness. The resulting estimator achieves oracle-optimal weighting, guarantees no first-order efficiency loss relative to a labeled-only baseline, and yields strict efficiency gains when predictions are informative. Importantly, our estimation procedure can be efficiently updated in an online fashion and hence is computationally efficient.

## 1.1. Related Work

Our work connects two largely separate lines of research: prediction-powered inference and statistical inference in adaptive experiments.

**Prediction-Powered Inference.** Prediction-Powered Inference (PPI) leverages predictions from black-box AI models to improve statistical efficiency when only a small i.i.d. labeled dataset is available alongside a large unlabeled dataset (Angelopoulos et al., 2023). PPI++ further improves upon this framework by introducing a power-tuning procedure that adapts to prediction quality and enhances both statistical and computational efficiency (Angelopoulos et al., 2024). More recently, Xu et al. (2025) reinterpret PPI through a semi-supervised learning lens, showing that independently trained predictors do not improve inference for well-specified models and proposing an estimator that improves upon PPI++ in terms of efficiency.

Despite advances, existing PPI-type methods (Angelopoulos et al., 2023; 2024; Kluger et al., 2025; Ji et al., 2025; Xu et al., 2025; Zrnic & Candès, 2024; Wang et al., 2020) are fundamentally designed for offline settings with i.i.d. labeled data. They are therefore ill-suited for adaptive experiments, where data are collected sequentially and actions depend on historical observations. In such settings, naïvely applying PPI-type estimators fails to yield valid inference, as the underlying estimating equations are no longer unbiased and standard asymptotic guarantees break down.

**Inference in Adaptive Experiments.** A complementary line of work studies valid statistical inference under adaptive data collection (Hadad et al., 2021; Deshpande et al., 2018; Khamaru & Zhang, 2024; Waudby-Smith et al., 2024; Zhang et al., 2020), particularly in contextual bandit and reinforcement learning settings. In the contextual bandit literature, Chen et al. (2021) develop inference procedures for linear reward models under $\epsilon$-greedy policies, while Zhang et al. (2021) propose a general M-estimation framework for adaptive inference. Guo & Xu (2025) extend these results to settings with model misspecification, and Han et al. (2025) study inference for bandits with matrix-valued contexts. Related work in reinforcement learning includes confidence interval construction for policy value functions in Markov decision processes (Shi et al., 2021), as well as extensions to confounded and doubly inhomogeneous environments (Shi et al., 2024; Bian et al., 2025).

However, existing adaptive inference methods rely exclusively on labeled data and do not incorporate unlabeled data or external predictive models. To the best of our knowledge, our work is the first to develop a prediction-powered adaptive inference framework that remains valid under adaptive experimental design, and to demonstrate, both theoretically and empirically, that predictive models can substantially improve inference efficiency in online adaptive experiments.

## 1.2. PPI++ Background

Before introducing our method, we briefly review PPI++ (Angelopoulos et al., 2023; 2024), which forms the conceptual foundation of our approach. PPI++ assumes access to i.i.d. labeled data $\{(\boldsymbol{X}_i, Y_i)\}_{i=1} \overset{i.i.d.}{\sim} \mathcal{P}$, unlabeled data $\tilde{\boldsymbol{X}}_i \overset{i.i.d.}{\sim} P_X$, and an AI model $f(\cdot)$ predicting labels. Here $\mathcal{P}_X$ is the marginal of $\mathcal{P}$. They consider the task of inferring a parameter $\boldsymbol{\theta}^* = \underset{\boldsymbol{\theta} \in \mathbb{R}^d}{\arg\min} \mathbb{E}[\ell(\boldsymbol{X}, Y; \boldsymbol{\theta})]$, for some loss function $\ell(\boldsymbol{x}, y; \boldsymbol{\theta})$, e.g., squared loss for linear regression $\ell_{\boldsymbol{\theta}}(\boldsymbol{x}_i, y_i; \boldsymbol{\theta}) = (\boldsymbol{x}_i^\top \boldsymbol{\theta} - y_i)^2$. PPI++ augments the classical estimating equation using predictions on unlabeled data together with a correction term estimated from labeled data by minimizing a "rectified" loss function:

$$\hat{\boldsymbol{\theta}}^{\mathrm{PP}} = \underset{\theta}{\arg\min} L_\lambda^{\mathrm{PP}}(\boldsymbol{\theta}), \quad \text{where} \tag{1}$$

$$L_\lambda^{\mathrm{PP}}(\boldsymbol{\theta}) = \frac{1}{n} \sum_{i=1}^n \ell(\boldsymbol{X}_i, Y_i; \boldsymbol{\theta})$$
$$+ \lambda \left( \frac{1}{N} \sum_{i=1}^N \ell(\tilde{\boldsymbol{X}}_i, f(\tilde{\boldsymbol{X}}_i); \boldsymbol{\theta}) - \frac{1}{n} \sum_{i=1}^n \ell(\boldsymbol{X}_i, f(\boldsymbol{X}_i); \boldsymbol{\theta}) \right).$$

Here, the first term of $L_\lambda^{\mathrm{PP}}(\boldsymbol{\theta})$ is the loss of the classical M-estimator, and the second term is a correction that incorporates the unlabeled data and predictions. The $\lambda$ is a weighting parameter that controls the influence of the predictions $f(\tilde{\boldsymbol{X}}_i)$. Heuristically, PPI++ sets $\lambda = 0$ when the predictions are uninformative and $\lambda = 1$ when they are very accurate. If $\lambda = 0$, we ignore predictions and unlabeled data, relying only on the labeled data; this recovers the classical estimator. On the other hand, if $\lambda = 1$, Equation (1) can be interpreted as an $f$-based loss plus a correction term, which is computed with the differences between the losses of the true and predicted labels on the labeled dataset. This decomposition illustrates a simple idea behind PPI++: to evaluate the bias of $f$, one can compare the true labels $Y_t$ against the predictions $f(\boldsymbol{X}_t)$ on the labeled dataset.

Finally, we note that Equation (1) expresses $\hat{\boldsymbol{\theta}}^{\mathrm{PP}}$ as an M-estimator, but by setting the derivative of $L_\lambda^{\mathrm{PP}}(\boldsymbol{\theta})$ to zero, we can equivalently write $\hat{\boldsymbol{\theta}}^{\mathrm{PP}}$ as a Z-estimator solving

$$\boldsymbol{G}_n^\lambda(\hat{\boldsymbol{\theta}}^{\mathrm{PP}}) = 0, \tag{2}$$

where $\boldsymbol{g}(\boldsymbol{x}, y; \boldsymbol{\theta}) = \nabla_{\boldsymbol{\theta}} \ell(\boldsymbol{x}, y; \boldsymbol{\theta})$ and

$$\boldsymbol{G}_n^\lambda(\boldsymbol{\theta}) = \frac{1}{n} \sum_{i=1}^n \boldsymbol{g}(\boldsymbol{X}_i, Y_i; \boldsymbol{\theta})$$
$$+ \lambda \left( \frac{1}{N} \sum_{i=1}^N \boldsymbol{g}(\tilde{\boldsymbol{X}}_i, f(\tilde{\boldsymbol{X}}_i); \boldsymbol{\theta}) - \frac{1}{n} \sum_{i=1}^n \boldsymbol{g}(\boldsymbol{X}_i, f(\boldsymbol{X}_i); \boldsymbol{\theta}) \right).$$
$$\tag{3}$$

prove inference efficiency in online adaptive experiments.

## 2. Problem Setting

We consider a contextual bandit setting, in which data are collected via the following procedure.

At time $T$,

1. The environment reveals a context $\boldsymbol{X}_T \in \mathcal{X} \subseteq \mathbb{R}^d$, drawn from a distribution $\mathcal{P}_X$.
2. The decision-maker selects an action $A_T \in \mathcal{A}$ according to a bandit policy $\pi_T(\cdot|\boldsymbol{X}_T, \mathcal{H}_{T-1}) \in \Delta(\mathcal{A})$, which depends on the current context $\boldsymbol{X}_T$ and the history $\mathcal{H}_{T-1} = \sigma(\{\boldsymbol{X}_t, A_t, Y_t\}_{t=1}^{T-1})$. Here $\Delta(\mathcal{A})$ is the set of probability distributions over the action space $\mathcal{A}$.
3. The decision-maker receives a reward $Y_T \in \mathbb{R}$. We let $\{Y_T(a) : a \in \mathcal{A}\}$ denote the potential outcomes (the rewards the decision-maker would have received for playing each action), so that $Y_T = Y_T(A_T)$.

In addition to the online experiment data, the decision-maker has an externally trained model $f_a : \mathcal{X} \to \mathbb{R}$ for $a \in \mathcal{A}$, where $f_a(\boldsymbol{x})$ is a prediction of the reward for playing action $a$ under context $\boldsymbol{x}$. In addition, the decision-maker has $N$ unlabeled contexts $\{\tilde{\boldsymbol{X}}_1, \ldots, \tilde{\boldsymbol{X}}_N\}$ sampled i.i.d. from $\mathcal{P}_X$. In Appendix F, we discuss extensions to settings with possible covariate shift, where the unlabeled data may come from a distribution different from the target population.

Our inferential target is the parameter $\boldsymbol{\theta}_a^\star \in \mathbb{R}^d$ associated with a specific action $a \in \mathcal{A}$, defined through a general Z-estimation problem:

$$\mathbb{E}\big[\boldsymbol{g}(\boldsymbol{X}, Y(a); \boldsymbol{\theta}_a^\star)\big] = \boldsymbol{0}, \tag{4}$$

where $\boldsymbol{g}(\cdot)$ is a known score function. Unlike many prior works on inference with adaptively collected data (Shi et al., 2021; 2024; Zhang et al., 2021; Han et al., 2025), we do not assume a well-specified outcome model. In particular, the validity of our inference procedure does not rely on correctly modeling the conditional distribution of $Y_T$ given $A_T$ and $\boldsymbol{X}_T$. This feature makes our approach especially well-suited for adaptive experiments conducted in complex environments, where model misspecification is common. In Section 2.1, we introduce our method via a key special case of misspecified linear bandits (Chen et al., 2021). Other common choices of $\boldsymbol{g}$ can model bandits with noisy contexts, generalized linear models, and off-policy evaluation (Guo & Xu, 2025).

### 2.1. Warm-Up: Misspecified Linear Bandits

In the misspecified linear bandits (Chen et al., 2021), the score function in (4) is $g(\boldsymbol{x}, y; \boldsymbol{\theta}) = \boldsymbol{x}(y - \boldsymbol{x}^\top \boldsymbol{\theta})$ and hence the inferential parameter $\boldsymbol{\theta}_a^\star$ solves $\mathbb{E}\big[\boldsymbol{X}(Y(a) - \boldsymbol{X}^\top \boldsymbol{\theta}_a^\star)\big] = 0$. This score function corresponds to the best linear approximation of $Y(a)$ based on $\boldsymbol{X}$ for arm $a$. When the true dependence of $Y(a)$ on $\boldsymbol{X}$ and $a$ is linear, this yields the true linear parameter. Otherwise, it defines the best linear projection of $Y(a)$ onto the covariates $\boldsymbol{X}$ in the least squares sense.

When ignoring the actions, the classical least squares estimator with labeled data $\{\boldsymbol{X}_t, Y_t\}_{t=1}^T$ is the OLS estimator:

$$\hat{\boldsymbol{\theta}}^{\text{OLS}} = \left[\sum_{t=1}^T \boldsymbol{X}_t \boldsymbol{X}_t^\top\right]^{-1} \sum_{t=1}^T \boldsymbol{X}_t Y_t. \tag{5}$$

For adaptive data with actions collected via a bandit policy, Chen et al. (2021) proposes a Weighted Least Squares (WLS) estimator, which uses inverse propensity weights to account for the adaptive bias:

$$\hat{\boldsymbol{\theta}}_{a,T}^{\text{WLS}} = \left[\sum_{t=1}^T \frac{1_{A_t=a}}{\pi_t(a)} \boldsymbol{X}_t \boldsymbol{X}_t^\top\right]^{-1} \sum_{t=1}^T \frac{1_{A_t=a}}{\pi_t(a)} \boldsymbol{X}_t Y_t. \tag{6}$$

Here $\pi_t(a) = \pi_t(a|\boldsymbol{X}_t, \mathcal{H}_{t-1}) := \mathbb{E}[1_{A_t=a}|\mathcal{H}_{t-1}, \boldsymbol{X}_t]$ was the probability of selecting action $a$ at time $t$. For example, the $\epsilon$-greedy policy (Lattimore & Szepesvári, 2020) plays the apparently best arm with probability $1 - \epsilon/2$ and chooses randomly with probability $\epsilon/2$. In a 2-armed linear bandit, it plays arm 1 at time $t$ with probability

$$\pi_t(1) = (1 - \epsilon)1_{\boldsymbol{X}_t^\top \hat{\boldsymbol{\theta}}_{1,t-1} > \boldsymbol{X}_t^\top \hat{\boldsymbol{\theta}}_{0,t-1}} + \frac{\epsilon}{2}.$$

Another example is the softmax policy (Lattimore & Szepesvári, 2020), which computes selection probabilities via exponential weighting:

$$\pi_t(1) = \frac{\exp(\boldsymbol{X}_t^\top \hat{\boldsymbol{\theta}}_{1,t-1})}{\exp(\boldsymbol{X}_t^\top \hat{\boldsymbol{\theta}}_{0,t-1}) + \exp(\boldsymbol{X}_t^\top \hat{\boldsymbol{\theta}}_{1,t-1})}.$$

Dividing by these probabilities removes the adaptive bias induced by $A_t$; observe that

$$\mathbb{E}\left[\frac{1_{A_t=a}}{\pi_t(a)} \boldsymbol{X}_t Y_t | \mathcal{H}_{t-1}\right]$$
$$= \mathbb{E}\left[\mathbb{E}\left[\frac{1_{A_t=a}}{\pi_t(a)} \boldsymbol{X}_t Y_t(a)|\mathcal{H}_{t-1}, \boldsymbol{X}_t, Y_t(a)\right]|\mathcal{H}_{t-1}\right]$$
$$= \mathbb{E}\left[\mathbb{E}\left[1_{A_t=a}|\mathcal{H}_{t-1}, \boldsymbol{X}_t\right]\frac{1}{\pi_t(a)} \boldsymbol{X}_t Y_t(a)|\mathcal{H}_{t-1}\right]$$
$$= \mathbb{E}\left[\boldsymbol{X}_t Y_t(a)|\mathcal{H}_{t-1}\right].$$

Here the first equality uses the law of iterated expectations; the second uses that $\pi_t(a)$, $\boldsymbol{X}_t$, and $Y_t(a)$ are measurable with respect to the inner conditioning and independence of $A_t$ and $Y_t(a)$; and the last uses the definition of $\pi_t(a)$.

Our proposed prediction-powered adaptive inference (PPAI) framework integrates unlabeled contexts $\{\tilde{\boldsymbol{X}}_i\}_{i=1}^N$ and AI predictions $f_a$ while preserving valid inference under adaptive data collection and misspecification. Specifically, our

PPAI estimator for the misspecified linear bandits is:

$$\hat{\boldsymbol{\theta}}_{a,T}^{\text{PPAI}} = \left[ \frac{1-\lambda}{T} \sum_{t=1}^{T} \frac{1_{A_t=a}}{\pi_t(a)} \boldsymbol{X}_t \boldsymbol{X}_t^T + \frac{\lambda}{N} \sum_{i=1}^{N} \tilde{\boldsymbol{X}}_i \tilde{\boldsymbol{X}}_i^T \right]^{-1}$$

$$\times \left[ \frac{1}{T} \sum_{t=1}^{T} \frac{1_{A_t=a}}{\pi_t(a)} \boldsymbol{X}_t Y_t + \lambda \left( \frac{1}{N} \sum_{i=1}^{N} \tilde{\boldsymbol{X}}_i f_a(\tilde{\boldsymbol{X}}_i) \right. \right.$$

$$\left. \left. - \frac{1}{T} \sum_{t=1}^{T} \frac{1_{A_t=a}}{\pi_t(a)} \boldsymbol{X}_t f_a(\boldsymbol{X}_t) \right) \right].$$

(7)

Like the PPI++ estimator (1), our PPAI estimator uses a tuning parameter $\lambda$ that weights the AI model based on its informativeness. If $\lambda = 0$, we ignore predictions and unlabeled data, relying only on the labeled data; this recovers the WLS estimator (6). On the other hand, if $\lambda = 1$, we weight the predictions highly, and the second bracketed term of (7) can be rewritten as

$$\frac{1}{N} \sum_{i=1}^{N} \tilde{\boldsymbol{X}}_i f_a(\tilde{\boldsymbol{X}}_i) + \frac{1}{T} \sum_{t=1}^{T} \frac{1_{A_t=a}}{\pi_t(a)} \boldsymbol{X}_t \left( Y_t - f_a(\boldsymbol{X}_t) \right).$$

This can be interpreted as an $f_a$-based prediction plus a correction term, which represents the bias of $f_a$.

## 3. Our Method under General Score Function

We now develop our PPAI estimator for the general inferential target $\boldsymbol{\theta}_a^*$ defined in (4) that solves $\mathbb{E}[\boldsymbol{g}(\boldsymbol{X}, Y(a); \boldsymbol{\theta}_a^*)] = 0$ for some known score function $\boldsymbol{g}(\cdot)$. Our PPAI estimator $\hat{\boldsymbol{\theta}}_{a,T}$ is defined as the solution to

$$\boldsymbol{G}_{a,T}^{\lambda_{a,T}}(\boldsymbol{\theta}) := \frac{\lambda_{a,T}}{N} \sum_{i=1}^{N} \boldsymbol{g}(\tilde{\boldsymbol{X}}_i, f_a(\tilde{\boldsymbol{X}}_i); \boldsymbol{\theta})$$

$$+ \frac{1}{T} \sum_{t=1}^{T} \frac{1_{\{A_t=a\}}}{\pi_t(a)} \left( \boldsymbol{g}(\boldsymbol{X}_t, Y_t; \boldsymbol{\theta}) \right. \quad (8)$$

$$\left. - \lambda_{a,T} \boldsymbol{g}(\boldsymbol{X}_t, f_a(\boldsymbol{X}_t); \boldsymbol{\theta}) \right) = 0.$$

Our estimator extends the PPI++ estimator in (2) to the adaptive data setting by incorporating the inverse propensity weighting. Specifically, the indicator $1_{A_t=a}$ restricts the calculations to times where arm $a$ was played, and inverse propensity weights $1/\pi_t(a)$ cancels the adaptive bias. Like PPI++, $\{\lambda_{a,T}\}_{T \geq 1}$ is a (data-dependent) real-valued sequence that controls the weighting of the AI model over the course of the experiment. Crucially, the quantity weighted by the inverse propensity weights is a small residual $\boldsymbol{g}(\boldsymbol{X}_t, Y_t(a); \boldsymbol{\theta}) - \lambda_{a,T} \boldsymbol{g}(\boldsymbol{X}_t, f_a(\boldsymbol{X}_t); \boldsymbol{\theta})$, which mitigates the high variance introduced by small weights. We will soon exhibit a specific choice of $\{\lambda_{a,T}\}_{T \geq 1}$ that shrinks this residual by weighting AI predictions according to their informativeness (Corollary 3.10).

Before establishing the theoretical properties of PPAI, we first introduce all regularity conditions. We assume $\{\boldsymbol{X}_t, Y_t(a) : a \in \mathcal{A}\} \overset{i.i.d.}{\sim} \mathcal{P}$ for all $t \geq 1$. That is, the environment's draw of contexts and potential outcomes are independent across time. This is a standard assumption in contextual bandit work (Guo & Xu, 2025; Zhang et al., 2021; Chen et al., 2021). While the contexts and potential outcomes are independent, the actions are not, as they are chosen based on the bandit policy and depend on the evolving experimental history. Consequently, our labeled data $\{(\boldsymbol{X}_t, Y_t) : A_t = a\}$ is not i.i.d., unlike the standard PPI setup. We also emphasize that we do not assume a functional form of the reward model $Y_t(a)|\boldsymbol{X}_t$, so our method does not require a well-specified reward model.

We first make the following assumptions on the policy $\pi$.

**Assumption 3.1** (Policy Convergence). There exists a stationary policy $\bar{\pi} : \mathcal{X} \to \Delta(\mathcal{A})$ such that

$$\pi_t(a|\boldsymbol{X}_t, \mathcal{H}_{t-1}) - \bar{\pi}(a|\boldsymbol{X}_t) \overset{p}{\to} 0 \quad \text{as } t \to \infty.$$

**Assumption 3.2** (Minimum sampling probability). For $a \in \mathcal{A}$, there exists a constant $\pi_{\min} \in (0,1)$ such that $\pi_t(a) \geq \pi_{\min}$ almost surely for all $t \geq 1$.

Policy convergence says that the probability of selecting arm $a$ given each context $\boldsymbol{X}_t$ stabilizes in the long run. This ensures an experiment's replicability by requiring that the policy's long-term behavior is always the same, regardless of the initial observations. This condition is satisfied by a broad class of commonly employed policies and is frequently assumed in adaptive inference works under misspecification (Chen et al., 2021; Guo & Xu, 2025). Minimum sampling probabilities are essential for inference under misspecification, as they can ensure sufficient exploration in a poorly understood environment, and they are also commonly assumed in the adaptive inference literature (Zhang et al., 2021; Chen et al., 2021; Zhang et al., 2023; Guo & Xu, 2025; Simchi-Levi & Wang, 2025; Han et al., 2025).

We also make the following causal assumption.

**Assumption 3.3** (Unconfoundedness). $A_t \perp \{Y_t(a)\}_{a \in \mathcal{A}} | (\mathcal{H}_{t-1}, \boldsymbol{X}_t)$ for $t = 1, \ldots, T$.

Assumption 3.3 is a common unconfoundedness assumption in bandit causal framing (Zhang et al., 2021; Guo & Xu, 2025; Han et al., 2025); it says that the decision-maker acts without knowledge of the potential outcomes beyond what they have already observed. We also adopt the following regularity conditions on the score function, which can be easily satisfied under misspecified linear bandits, bandits with noisy contexts, and score functions corresponding to other common settings (Guo & Xu, 2025).

**Assumption 3.4** (Well-separated solution). $\forall \epsilon > 0$, $\inf_{||\boldsymbol{\theta} - \boldsymbol{\theta}_a^*||_2 > \epsilon} ||\mathbb{E}[\boldsymbol{g}(\boldsymbol{X}_t, Y_t(a); \boldsymbol{\theta})]||_2 > 0$.

**Assumption 3.5** (Bounded moments). There exist constants $R_\theta, M_2, M_3, M_4$ such that, a.e. $\boldsymbol{X}_t$,

(i) $||\mathbb{E}[\boldsymbol{g}(\boldsymbol{X}_t, Y_t(a); \boldsymbol{\theta}_a^*)\boldsymbol{g}(\boldsymbol{X}_t, Y_t(a); \boldsymbol{\theta}_a^*)^\mathsf{T}|\boldsymbol{X}_t]||_2 \leq M_2$, $||\mathbb{E}[\boldsymbol{g}(\boldsymbol{X}_t, f_a(\boldsymbol{X}_t); \boldsymbol{\theta}_a^*)\boldsymbol{g}(\boldsymbol{X}_t, f_a(\boldsymbol{X}_t); \boldsymbol{\theta}_a^*)^\mathsf{T}|\boldsymbol{X}_t]||_2 \leq M_3$, $||\mathbb{E}[\boldsymbol{g}(\boldsymbol{X}_t, Y_t(a); \boldsymbol{\theta}_a^*)\boldsymbol{g}(\boldsymbol{X}_t, f_a(\boldsymbol{X}_t); \boldsymbol{\theta}_a^*)^\mathsf{T}|\boldsymbol{X}_t]||_2 \leq M_4$;

(ii) $||\boldsymbol{\theta}_a^*||_2 < R_\theta$, $\sup_{||\boldsymbol{\theta}|| \leq R_\theta} \mathbb{E}[||\boldsymbol{g}(\boldsymbol{X}_t, Y_t(a); \boldsymbol{\theta})||_2^2] < \infty$;

(iii) $\mathbb{E}[||\boldsymbol{g}(\boldsymbol{X}_t, Y_t(a); \boldsymbol{\theta}_a^*)||_2^4], \mathbb{E}[||\boldsymbol{g}(\boldsymbol{X}_t, f_a(\boldsymbol{X}_t); \boldsymbol{\theta}_a^*)||_2^4] < \infty$.

**Assumption 3.6** (Smoothness). (i) The function $\boldsymbol{g}(\boldsymbol{x}, y; \boldsymbol{\theta})$ is twice differentiable with respect to $\boldsymbol{\theta}$, with $\mathbb{E}[\nabla\boldsymbol{g}(\boldsymbol{X}_t, Y_t(a); \boldsymbol{\theta}_a^*)]$ nonsingular;

(ii) There exists a function $\phi : \mathbb{R}^{d_X} \times \mathbb{R} \to \mathbb{R}$ such that $\forall \boldsymbol{x}, y, \sup_{||\boldsymbol{\theta}|| \leq R_\theta} ||\nabla\boldsymbol{g}(\boldsymbol{x}, y; \boldsymbol{\theta})||_2 \leq \phi(\boldsymbol{x}, y)$, and $\mathbb{E}[\phi(\boldsymbol{X}_t, Y_t(a))^2] < \infty$;

(iii) There exists a constant $\epsilon_0 > 0$ and a function $\Phi : \mathbb{R}^{d_X} \times \mathbb{R} \to \mathbb{R}$ such that $\sup_{||\boldsymbol{\theta} - \boldsymbol{\theta}^*|| \leq \epsilon_0, i \in [d]} ||\nabla^2\boldsymbol{g}^{(i)}(\boldsymbol{x}, y; \boldsymbol{\theta})||_2 \leq \Phi(\boldsymbol{x}, y)$ and $\mathbb{E}[\Phi(\boldsymbol{X}_t, Y_t(a))] < \infty$, where $\boldsymbol{g}^{(i)}(\boldsymbol{x}, y; \boldsymbol{\theta})$ is the $i$th entry of $\boldsymbol{g}(\boldsymbol{x}, y; \boldsymbol{\theta})$.

Assumption 3.4 is a standard assumption in Z-estimation that requires the expected score to be bounded away from zero outside any neighborhood of the true parameter $\boldsymbol{\theta}_a^*$; this ensures that $\boldsymbol{\theta}_a^*$ is identifiable (Vaart, 1998). Assumption 3.5 imposes mild regularity on the moments of the score functions $\boldsymbol{g}(\boldsymbol{x}, y; \boldsymbol{\theta}_a^*)$ and $\boldsymbol{g}(\boldsymbol{x}, f(\boldsymbol{x}); \boldsymbol{\theta}_a^*)$ with true and predicted labels, respectively. These types of boundedness assumptions are standard for establishing asymptotic normality of estimators in adaptive experiments (Zhang et al., 2021; Chen et al., 2021; Guo & Xu, 2025). Finally, part (i) of Assumption 3.6 is required for identifiability of $\boldsymbol{\theta}_a^*$, and parts (ii) and (iii) impose smoothness on $\boldsymbol{g}$; all three are standard for Z-estimation in both classical and adaptive settings (Vaart, 1998; Zhang et al., 2023).

In what follows, we let $r := \lim_{T \to \infty} T/N$ denote the limit of the (typically small) ratio of the sizes of the labeled and unlabeled datasets, and we let $\overline{\lambda}_a$ denote the limit in probability of $\lambda_{a,T}$. Also, for $\boldsymbol{v} \in \mathbb{R}^d$, we write $\boldsymbol{v}^{\otimes 2} := \boldsymbol{v}\boldsymbol{v}^\top$. For brevity, we sometimes suppress dependence on the action $a$ in our notation, writing $\lambda_T$ for $\lambda_{a,T}$, $\overline{\lambda}$ for $\overline{\lambda}_a$, $f$ for $f_a$, $\widehat{\boldsymbol{\theta}}_T$ for $\widehat{\boldsymbol{\theta}}_{a,T}$, $\boldsymbol{\theta}^*$ for $\boldsymbol{\theta}_a^*$, and $\boldsymbol{G}_T^{\lambda_T}(\boldsymbol{\theta})$ for $\boldsymbol{G}_{a,T}^{\lambda_{a,T}}(\boldsymbol{\theta})$.

**Theorem 3.7** (Asymptotic Normality). *Suppose Assumptions 3.1-3.6 hold for an action $a \in \mathcal{A}$. Then there exists an estimator sequence $\{\widehat{\boldsymbol{\theta}}_T\}_{T \geq 1}$ satisfying the estimating equation (8) with $||\widehat{\boldsymbol{\theta}}_T||_2 \leq R_\theta$ for all $T$. Moreover, for any such sequence, as $T \to \infty$,*

$$\sqrt{T}(\widehat{\boldsymbol{\theta}}_T - \boldsymbol{\theta}^*) \xrightarrow{d} \mathcal{N}(\boldsymbol{0}, \boldsymbol{\Sigma}_{\overline{\lambda}}), \tag{9}$$

*where*

$$\boldsymbol{\Sigma}_{\overline{\lambda}} = \boldsymbol{J}^{-1}(\overline{\lambda}^2 r\text{Cov}(\boldsymbol{g}(\boldsymbol{X}_t, f(\boldsymbol{X}_t); \boldsymbol{\theta}^*)) + \boldsymbol{V})\boldsymbol{J}^{-1\top}, \tag{10}$$

$$\boldsymbol{J} = \mathbb{E}\nabla\boldsymbol{g}(\boldsymbol{X}_t, Y_t(a); \boldsymbol{\theta}^*),$$

$$\boldsymbol{V} = \mathbb{E}\left[\frac{1}{\overline{\pi}(a)}\left(\boldsymbol{g}(\boldsymbol{X}_t, Y_t(a); \boldsymbol{\theta}^*) - \overline{\lambda}\boldsymbol{g}(\boldsymbol{X}_t, f(\boldsymbol{X}_t); \boldsymbol{\theta}^*)\right)^{\otimes 2}\right]$$
$$- \overline{\lambda}^2 \mathbb{E}[\boldsymbol{g}(\boldsymbol{X}_t, f(\boldsymbol{X}_t); \boldsymbol{\theta}^*)]^{\otimes 2}.$$

In Appendix C, we give a consistent estimator of (10), enabling construction of confidence intervals. Also, note that if $\overline{\lambda} = 0$, (10) reduces to

$$\Sigma_0 = \boldsymbol{J}^{-1}\mathbb{E}\left[\frac{1}{\overline{\pi}(a)}\boldsymbol{g}(\boldsymbol{X}_t, Y_t(a); \boldsymbol{\theta}^*)^{\otimes 2}\right]\boldsymbol{J}^{-1\top},$$

which coincides with the variance from Theorem 3.2 of Guo & Xu (2025).

Establishing this result is technically nontrivial for several reasons. Firstly, we explicitly allow misspecification of the reward model, which invalidates classical likelihood-based arguments. Secondly, since the labeled data are collected adaptively, the standard empirical process methods used to derive PPI asymptotics are not applicable; instead, we employ martingale limit theory.

A key step in our proof is to derive the asymptotic distribution of the scaled estimating equation $\sqrt{T}\boldsymbol{G}_T^{\lambda_T}(\boldsymbol{\theta}^*)$. This term depends on both unlabeled and labeled data, which are coupled by the weighting parameter $\lambda_T$. We analyze this expression by decomposing it into its labeled and unlabeled components. While the unlabeled component consists of i.i.d. data amenable to the central limit theorem, the labeled component involves dependent data. We analyze the latter by expressing it as a normalized sum of a martingale difference sequence, which enables the application of a martingale central limit theorem.

The asymptotic normality result (9) holds for any fixed choice of the weighting parameter $\overline{\lambda}$, ensuring valid inference regardless of how AI predictions are weighted. However, different choices of $\overline{\lambda}$ lead to different asymptotic variances. To fully exploit the auxiliary information provided by the predictions, we now characterize the choice of $\overline{\lambda}$ that minimizes the asymptotic variance of the estimator. In particular, we derive the power-tuning parameter $\lambda^*$ that minimizes the trace of $\boldsymbol{\Sigma}_{\overline{\lambda}}$.

**Theorem 3.8** (Optimal $\overline{\lambda}$). *The trace of $\boldsymbol{\Sigma}_{\overline{\lambda}}$ is minimized at*

$$\lambda^* = \frac{\text{tr}\left(\boldsymbol{J}^{-1}\mathbb{E}[\frac{1}{\overline{\pi}(a)}(\boldsymbol{g}_f^*\boldsymbol{g}_Y^{*\mathsf{T}} + \boldsymbol{g}_Y^*\boldsymbol{g}_f^{*\mathsf{T}})]\boldsymbol{J}^{-1\top}\right)}{2\text{tr}\left(\boldsymbol{J}^{-1}(r\text{Cov}(\boldsymbol{g}_f^*) + E[\frac{\boldsymbol{g}_f^{*\otimes 2}}{\overline{\pi}(a)}] - E[\boldsymbol{g}_f^*]^{\otimes 2})\boldsymbol{J}^{-1\top}\right)}, \tag{11}$$

*where $\boldsymbol{g}_Y^* = \boldsymbol{g}(\boldsymbol{X}, Y(a); \boldsymbol{\theta}^*), \boldsymbol{g}_f^* = \boldsymbol{g}(\boldsymbol{X}, f(\boldsymbol{X}); \boldsymbol{\theta}^*)$, and $\boldsymbol{J} = E[\nabla\boldsymbol{g}_Y^*]$.*

**Algorithm 1** PPAI Online Update

**Input:** Experiment history $\{(\boldsymbol{X}_t, A_t, Y_t)\}_{t=1}^T$, unlabeled contexts $\{\tilde{\boldsymbol{X}}_i\}_{i=1}^{N(T+1)}$, predictive model $f$, current estimate $\hat{\boldsymbol{\theta}}$, intermediate sums $\boldsymbol{B}, \boldsymbol{C}, \boldsymbol{D}, \boldsymbol{E}, \boldsymbol{F}(\boldsymbol{\theta}), \boldsymbol{H}(\boldsymbol{\theta}), \boldsymbol{I}(\boldsymbol{\theta})$
At time $T + 1$, observe $\boldsymbol{X}_{T+1}$, update
$\boldsymbol{B} \leftarrow \boldsymbol{B} + \frac{1_{A_T=a}}{\pi_T(a)} \nabla \boldsymbol{g}_{Y,T}$
$\boldsymbol{C} \leftarrow \boldsymbol{C} + \frac{1_{\{A_T=a\}}}{\pi_T(a)^2} \left( \boldsymbol{g}_{f,T} \boldsymbol{g}_{Y,T}^\top + \boldsymbol{g}_{Y,T} \boldsymbol{g}_{f,T}^\top \right)$
$\boldsymbol{D} \leftarrow \boldsymbol{D} + \frac{1_{A_T=a}}{\pi_T(a)} \boldsymbol{g}_{f,T}^{\otimes 2}$
$\boldsymbol{E} \leftarrow \boldsymbol{E} + \frac{1_{A_T=a}}{\pi_T(a)} \boldsymbol{g}_{f,T}$
$\boldsymbol{F} \leftarrow \boldsymbol{F} + \frac{1_{A_T=a}}{\pi_T(a)^2} \boldsymbol{g}_{f,T} \boldsymbol{g}_{f,T}^\top$
$\hat{\lambda} \leftarrow \frac{\mathrm{tr}(\boldsymbol{B}^{-1} \boldsymbol{C} \boldsymbol{B}^{-1})}{2\mathrm{tr}(\boldsymbol{B}^{-1}(\frac{T}{N}(\boldsymbol{D} - \frac{1}{T}\boldsymbol{E}\boldsymbol{E}^\top) + \boldsymbol{F} - \frac{1}{T}\boldsymbol{E}\boldsymbol{E}^\top)\boldsymbol{B}^{-1})}$
$\boldsymbol{F}(\boldsymbol{\theta}) \leftarrow F(\boldsymbol{\theta}) + \frac{1_{A_T=a}}{\pi_T(a)} \boldsymbol{g}_{Y,T}$
$\boldsymbol{H}(\boldsymbol{\theta}) \leftarrow H(\boldsymbol{\theta}) + \sum_{i=N(T)+1}^{N(T+1)} \boldsymbol{g}(\tilde{\boldsymbol{X}}_i, f(\tilde{\boldsymbol{X}}_i); \boldsymbol{\theta})$
$\boldsymbol{I}(\boldsymbol{\theta}) \leftarrow I(\boldsymbol{\theta}) + \frac{1_{A_T=a}}{\pi_T(a)} \boldsymbol{g}_{f,T}$
$\boldsymbol{G}(\boldsymbol{\theta}) \leftarrow \frac{1}{T+1} F(\boldsymbol{\theta}) + \hat{\lambda} \left( \frac{1}{N(T+1)} H(\boldsymbol{\theta}) - \frac{1}{T+1} I(\boldsymbol{\theta}) \right)$
$\hat{\boldsymbol{\theta}} = \mathrm{root}(\boldsymbol{G}(\boldsymbol{\theta}))$
**return** $\hat{\boldsymbol{\theta}}$

The optimal $\bar{\lambda}$ balances two objectives: limiting variance contributed by the AI model (which favors small $\lambda$) and minimizing the IPW-weighted residual (which favors $\lambda$ that aligns $\lambda g_f$ with $g_Y$). The denominator acts as a normalizing factor that penalizes high variance in $f(\boldsymbol{X})$, reflecting the first objective. The numerator of $\lambda^*$ measures the (weighted) alignment of the AI predictions and true rewards, reflecting the second objective. If they are well aligned, $\lambda^* \approx 1$, and if they are poorly aligned, $\lambda^* \approx 0$.

We also remark that $\lambda^*$ can be efficiently estimated in an online fashion, and consequently, $\hat{\boldsymbol{\theta}}_T$ can be updated online as well. Algorithm 1 summarizes the online procedure for updating $\hat{\boldsymbol{\theta}}_T$ to $\hat{\boldsymbol{\theta}}_{T+1}$. To avoid repeatedly summing over the labeled and unlabeled datasets, it stores intermediate sums $B, C, D, E, F$ to compute an estimate $\hat{\lambda}$. It also stores sums $F(\boldsymbol{\theta}) = \sum_{t=1}^T \frac{1_{A_t=a}}{\pi_t(a)} \boldsymbol{g}(\boldsymbol{X}_t, Y_t; \boldsymbol{\theta})$, $H(\boldsymbol{\theta}) = \frac{1}{N} \sum_{i=1}^N \boldsymbol{g}(\tilde{\boldsymbol{X}}_i, f_a(\tilde{\boldsymbol{X}}_i); \boldsymbol{\theta})$, and $I(\boldsymbol{\theta}) = \sum_{t=1}^T \frac{1_{A_t=a}}{\pi_t(a)} \boldsymbol{g}(\boldsymbol{X}_t, f_a(\boldsymbol{X}_t); \boldsymbol{\theta})$ to compute the estimating function $G_{T+1}^{\hat{\lambda}}(\boldsymbol{\theta})$. Since $\hat{\boldsymbol{\theta}}_{T+1}$ is defined as a root of $\boldsymbol{G}_{T+1}^{\hat{\lambda}}(\boldsymbol{\theta})$, it can be computed with standard root-finding algorithms like Newton's method.

Having derived an estimate $\hat{\lambda}$ of the optimal $\lambda^*$, we next prove its consistency; then we establish the minimal asymptotic variance of our estimator. In what follows, we denote $\boldsymbol{g}_{f,t} := \boldsymbol{g}(\boldsymbol{X}_t, f(\boldsymbol{X}_t); \hat{\boldsymbol{\theta}}_t)$ and $\boldsymbol{g}_{Y,t} := \boldsymbol{g}(\boldsymbol{X}_t, Y_t; \hat{\boldsymbol{\theta}}_t)$.

**Corollary 3.9.** *A consistent estimate of $\lambda^*$ is*

$$\hat{\lambda}_T := \frac{\mathrm{tr}(\boldsymbol{B}_T^{-1} \boldsymbol{C}_T \boldsymbol{B}_T^{-1})}{2\mathrm{tr}(\boldsymbol{B}_T^{-1}(\frac{T}{N}(\boldsymbol{D}_T - \boldsymbol{E}_T \boldsymbol{E}_T^\top) + \boldsymbol{F}_T - \boldsymbol{E}_T \boldsymbol{E}_T^\top) \boldsymbol{B}_T^{-1})}, \tag{12}$$

*where*

$$\boldsymbol{B}_T = \frac{1}{T-1} \sum_{t=1}^{T-1} \frac{1_{A_t=a}}{\pi_t(a)} \nabla \boldsymbol{g}_{Y,t},$$

$$\boldsymbol{C}_T = \frac{1}{T-1} \sum_{t=1}^{T-1} \frac{1_{\{A_t=a\}}}{\pi_t(a)^2} \left( \boldsymbol{g}_{f,t} \boldsymbol{g}_{Y,t}^\top + \boldsymbol{g}_{Y,t} \boldsymbol{g}_{f,t}^\top \right)$$

$$\boldsymbol{D}_T = \frac{1}{T-1} \sum_{t=1}^{T-1} \frac{1_{\{A_t=a\}}}{\pi_t(a)} \boldsymbol{g}_{f,t} \boldsymbol{g}_{f,t}^\top,$$

$$\boldsymbol{E}_T = \frac{1}{T-1} \sum_{t=1}^{T-1} \frac{1_{\{A_t=a\}}}{\pi_t(a)} \boldsymbol{g}_{f,t},$$

$$\boldsymbol{F}_T = \frac{1}{T-1} \sum_{t=1}^{T-1} \frac{1_{A_t=a}}{\pi_t(a)^2} \boldsymbol{g}_{f,t} \boldsymbol{g}_{f,t}^\top.$$

**Corollary 3.10.** *Let $\lambda_T = \hat{\lambda}_T$ from (12). Then the asymptotic variance of $\hat{\boldsymbol{\theta}}_T$ from Algorithm 1 has minimal trace among all variances of the form (10).*

This adaptive power-tuning enables our estimator to adapt to the quality of $f_a$. If $f_a$ is accurate ($f_a(\boldsymbol{X}_t) \approx Y_t(a)$), then $\boldsymbol{g}(\boldsymbol{X}_t, Y_t(a); \boldsymbol{\theta}^*) \approx \boldsymbol{g}(\boldsymbol{X}_t, f_a(\boldsymbol{X}_t); \boldsymbol{\theta}^*)$ and $\lambda^* \approx 1$, so $\boldsymbol{V} \approx 0$. In this case, the asymptotic variance is roughly $r\boldsymbol{J}^{-1}\mathrm{Cov}(\boldsymbol{g}(\boldsymbol{X}, f(\boldsymbol{X}); \boldsymbol{\theta}^*))\boldsymbol{J}^{-1}$. When we have abundant unlabeled data, $r$ is small, and this variance is small. On the other hand, if $f_a$ is not informative, then $\lambda^* = 0$, in which case the asymptotic variance is $\boldsymbol{J}^{-1}\boldsymbol{V}\boldsymbol{J}^{-1}$ which coincides with the variance from Theorem 3.2 of Guo & Xu (2025). In summary, if $f_a$ is accurate, our estimator leverages the predictions to reduce variance, and if $f_a$ is totally uninformative, our estimator ignores the predictions and recovers the variance of the labeled-only baseline. In this way, our estimator incorporates the predictions safely.

## 4. Numerical Results

We simulate our method on a two-armed contextual bandit with reward models

$$Y_t(0) = \exp(0.2\boldsymbol{X}_t^{(1)} - 0.1\boldsymbol{X}_t^{(2)} + 0.4\boldsymbol{X}_t^{(3)}) + e_t,$$

$$Y_t(1) = \exp(0.5\boldsymbol{X}_t^{(1)} + 0.2\boldsymbol{X}_t^{(2)} - 0.1\boldsymbol{X}_t^{(3)}) + e_t,$$

for arms 0 and 1 respectively, where $e_t \sim N(0, \sigma^2)$ with $\sigma = 0.1$. Our inferential target is the least squares parameter $\boldsymbol{\theta}_0^*$ for arm 0, which satisfies $\mathbb{E}[\boldsymbol{X}(Y(0) - \boldsymbol{X}^\top \boldsymbol{\theta}_0^*)] = 0$. The contexts $\boldsymbol{X}_t \sim \mathcal{N}_3(\boldsymbol{0}, \boldsymbol{I})$ are drawn i.i.d. In addition, 100 i.i.d. unlabeled contexts arrive at each time step,

so at time $t$, the decision-maker has unlabeled contexts $\tilde{X}_1, \ldots, \tilde{X}_{100t} \sim \mathcal{N}_3(\mathbf{0}, \mathbf{I})$.

We estimate $\boldsymbol{\theta}_0^*$ with our PPAI estimator $\hat{\boldsymbol{\theta}}_{0,T}$, which leverages the unlabeled dataset and a predictive model $f_0$. We vary the informativeness of $f_0$ by simulating with $f_0^{\mathrm{poor}}$ and $f_0^{\mathrm{strong}}$, given by $f_0^{\mathrm{poor}}(\boldsymbol{x}) = 10$ and

$$f_0^{\mathrm{strong}}(\boldsymbol{x}) = \exp(0.21\boldsymbol{x}^{(1)} - 0.09\boldsymbol{x}^{(2)} + 0.42\boldsymbol{x}^{(3)}).$$

Note that $f_0^{\mathrm{strong}}$ is strongly aligned with the true reward model for arm 0 (i.e. highly informative), whereas $f_0^{\mathrm{poor}}$ is poorly aligned (uninformative). For comparison, we also estimate $\boldsymbol{\theta}_0^*$ with the labeled-only baseline estimator, Weighted Least Squares (WLS) (see Chen et al. (2021)).

Actions are chosen according to an $\epsilon$-greedy policy: at each time $t$, with probability $\epsilon$, we select $A_t \in \{0, 1\}$ randomly, and with probability $1 - \epsilon$, we select $A_t = \operatorname{argmax}_a \boldsymbol{X}_t^T \hat{\boldsymbol{\theta}}_{a,T}$, where $\hat{\boldsymbol{\theta}}_{0,T}$ is described above and $\hat{\boldsymbol{\theta}}_{1,T}$ is the WLS estimator for arm 1. Here we set $\epsilon = 0.3$.

Each simulation lasts $T = 5000$ time steps. The first 100 time steps are devoted to random exploration: we randomly assign 50 of the first 100 times to arm 0 and the rest to arm 1 in order to ensure that each arm has enough samples to form the initial estimators. We run 300 simulations in which $\hat{\boldsymbol{\theta}}_{0,T}$ is PPAI with $f_0^{\mathrm{strong}}$, 300 simulations in which $\hat{\boldsymbol{\theta}}_{0,T}$ is PPAI with $f_0^{\mathrm{poor}}$, and 300 simulations in which $\hat{\boldsymbol{\theta}}_{0,T}$ is WLS.

We approximate the true $\boldsymbol{\theta}_0^*$ with the OLS estimator of $10^8$ random samples, finding $\boldsymbol{\theta}_0^* \approx (0.22, -0.11, 0.44)$. To illustrate our results, we restrict attention to the first component of $\boldsymbol{\theta}_0^*$. Figure 2 compares the sampling distributions of (the first components of) the PPAI estimator for $\boldsymbol{\theta}_0^*$ with $f_0^{\mathrm{strong}}$, PPAI with $f_0^{\mathrm{poor}}$, and the labeled-only baseline (WLS). Our PPAI estimators reduce variance relative to the labeled-only baseline, and this reduction is largest when PPAI uses $f_0^{\mathrm{strong}}$. We also compute 90% confidence intervals for the first com-

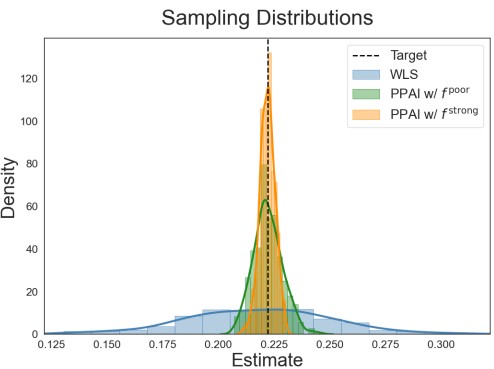

*Figure 2.* Sampling distributions of PPAI with a strong predictive model, PPAI with a poor predictive model, and the labeled-only baseline (WLS), using an $\epsilon$-greedy policy with $\epsilon = 0.3$. The inferential target is approximately 0.22. PPAI with $f^{\mathrm{strong}}$ has the smallest variance, and the labeled-only baseline has the largest.

ponent of $\boldsymbol{\theta}_0^*$ halfway through ($T = 2500$) and at the end of each experiment ($T = 5000$). Further implementation details are discussed in Appendix D. Table 1 summarizes the average width and coverage of the confidence intervals. Both PPAI variants yield narrower intervals relative to WLS, and PPAI with $f_0^{\mathrm{strong}}$ achieves a tenfold reduction in width.

| | | $\hat{\boldsymbol{\theta}}$ | | |
|---|---|---|---|---|
| | | **WLS** | **PPAI** ($f^{\mathrm{poor}}$) | **PPAI** ($f^{\mathrm{strong}}$) |
| **T** | **2500** | 0.161 (92%) | 0.030 (88%) | 0.016 (91%) |
| | **5000** | 0.111 (91%) | 0.021 (90%) | 0.011 (91%) |

*Table 1.* Average width and coverage (in parentheses) of 90% confidence intervals for WLS, PPAI with $f^{\mathrm{poor}}$, and PPAI with $f^{\mathrm{strong}}$. Both PPAI variants yield narrower intervals relative to WLS, and PPAI with $f^{\mathrm{strong}}$ achieves a tenfold reduction in width.

Figure 3 shows the convergence of the power-tuning parameter estimates, summarizing 300 trajectories of $\hat{\lambda}_T$ for each of $f^{\mathrm{strong}}$ and $f^{\mathrm{poor}}$. We approximate the oracle power-tuning parameters by approximating each of the expectations in (11) using Monte Carlo simulations with $10^6$ samples. We find $\lambda^* \approx 0.988$ for $f_0^{\mathrm{strong}}$ and $\lambda^* \approx 0.116$ for $f_0^{\mathrm{poor}}$.

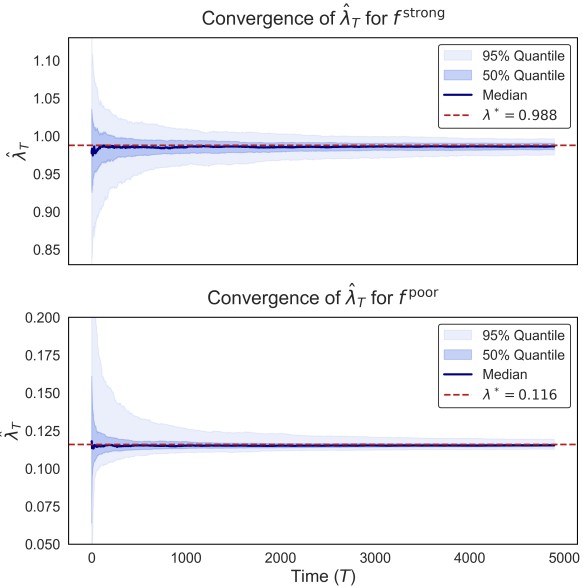

*Figure 3.* Quantile summary of 300 independent trajectories of $\hat{\lambda}_T$ for each predictive model, $f^{\mathrm{strong}}$ and $f^{\mathrm{poor}}$. The oracle parameters are $\lambda^* \approx 0.988$ and $\lambda^* \approx 0.116$, for $f^{\mathrm{strong}}$ and $f^{\mathrm{poor}}$, respectively.

Additional experiments are reported in Appendix D. These results show that PPAI can substantially reduce variance in low-exploration regimes with a strong AI model, that the bias-correction term is necessary for maintaining accuracy, and that adaptive tuning of $\hat{\lambda}_T$ achieves lower variance than a fixed-parameter baseline. We also vary the degree of reward misspecification and find that PPAI maintains robust performance across a range of settings.

# 5. Real Data Analysis

We evaluate PPAI on a movie recommendation task using *The Movies Dataset*, which contains approximately 26 million ratings from 270,000 users. Gemini is used to predict a user's movie ratings based on their individual rating history.

## 5.1. Contexts and Actions

We choose five movies to serve as "representatives": *Independence Day* (1996), *The Lord of the Rings: The Return of the King* (2003), *Star Wars: Episode IV - A New Hope* (1977), *Pulp Fiction* (1994), and *Shrek* (2001). We filter our dataset to include only the 11,198 users who have rated all five of these movies.

Each user's context is defined as a standardized vector $x \in \mathbb{R}^5$ of their ratings for these representative movies. We perform standardization of their raw rating vector $x_{\text{raw}} \in \mathbb{R}^5$ by subtracting the average context $\mu \in \mathbb{R}^5$ and dividing by the vector of standard deviations $\sigma \in \mathbb{R}^5$, i.e. the standardized vector is defined entry-wise as $x = (x_{\text{raw}} - \mu)/\sigma$.

We let the action set $\mathcal{A} = \{0, 1\}$ consist of two actions, which correspond to recommending *X-Men* (2000) or *Good Will Hunting* (1997), respectively.

## 5.2. Reward Model

Since we do not observe the rating (i.e., reward) each user would give to every movie, we train a reward model $y : \mathcal{U} \times \mathcal{M} \rightarrow \{0.5, 1.0, 1.5, \dots, 5.0\}$, where $\mathcal{U}$ denotes the set of 11,198 users and $\mathcal{M}$ denotes the set of movies. We treat $y(u, m)$ as the true rating that user $u \in \mathcal{U}$ would assign to movie $m \in \mathcal{M}$. Further details on the reward model are in Appendix E.1.

In our experiments, we query this model for the "true" ratings, though we use the data to infer the parameters $\beta_0, \beta_1$ of a simpler linear reward model $y = 1_{A_t=0}\beta_0^\top x + 1_{A_t=1}\beta_1^\top x$. These parameters represent how ratings of the two movies in the action set vary linearly with ratings of the five representative movies; for instance, the second coordinate of $\beta_0$ characterizes the linear relationship between a user's ratings of *X-Men* (2000) and *The Lord of the Rings: Return of the King* (2003).

## 5.3. Experiment Setup

We randomly select 10% of the users to be in the labeled dataset and the remaining 90% to be unlabeled. We now simulate the adaptive experiment by applying an $\epsilon$-greedy policy and computing the PPAI least squares estimator (7). To get predicted ratings, we ask Gemini to predict how a user will rate a particular movie given their rating history. For comparison, we repeat the experiment with the Weighted Least Squares estimator (6), which does not use unlabeled

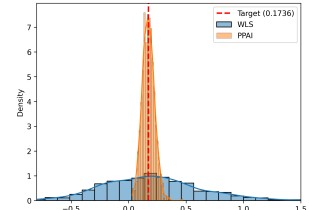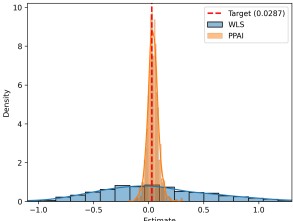

*Figure 4.* Sampling distributions of PPAI vs. WLS estimates of $\beta_0^{(3)} \approx 0.1736$ (left) and $\beta_1^{(1)} \approx 0.0287$ (right).

data or Gemini predictions. For more details, see Appendix E.2.

## 5.4. Results

For illustration, we focus on two estimands: the third coordinate of $\beta_0$, which characterizes the linear relationship between a user's rating of *X-Men* (2000) and *Star Wars: Episode IV - A New Hope* (1977), and the first coordinate of $\beta_1$, which characterizes the linear relationship between a user's rating of *Good Will Hunting* (1997) and *Independence Day* (1996). The true values are $\beta_0^{(3)} \approx 0.1736$ and $\beta_1^{(1)} \approx 0.0287$.

Figure 4 displays the sampling distributions of PPAI and WLS for each estimand, and Table 2 reports their variances.

|  | WLS | PPAI |
|---|---|---|
| $\beta_0^{(3)}$ | 0.170 | 0.003 |
| $\beta_1^{(1)}$ | 0.304 | 0.002 |

*Table 2.* Variance of WLS and PPAI estimators for each inferential target. For both targets, the variance of the PPAI estimator is substantially smaller than that of WLS.

We also construct 90% confidence intervals with both approaches; Table 3 summarizes their average length and coverage, showing that PPAI produces intervals that are approximately 5-7 times shorter than those of WLS while achieving superior coverage.

|  | WLS | PPAI |
|---|---|---|
| $\beta_0^{(3)}$ | 1.23 (82.6%) | 0.21 (95.1%) |
| $\beta_1^{(1)}$ | 1.44 (80.5%) | 0.22 (99.0%) |

*Table 3.* Average length and coverage for 90% confidence intervals $(T = 1000, N = 1000)$. For both targets, PPAI confidence intervals are 5-7 times tighter and achieve superior coverage.

We also perform hypothesis tests to illustrate the inferential benefits of PPAI, finding that it delivers substantially higher testing power (see Appendix E.3).

## Acknowledgments

Gabriel Sargent was partially supported by NSF RTG grant DMS-2134107 and NSF Grant SES-2217440. The research of Will Wei Sun and Yufeng Liu was partially supported by NSF Grant SES-2217440.

## Impact Statement

This paper presents work whose goal is to advance the field of Machine Learning. There are many potential societal consequences of our work, none which we feel must be specifically highlighted here.

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

# A. Proofs of Main Results

## A.1. Proof of Theorem 3.7

*Proof.* The proof of this theorem (and its dependent lemmas) mirrors that of Theorem 3.2 from Guo & Xu (2025) but with the modifications required by our different Z-equation.

For the remainder of this proof, we write $\widehat{\boldsymbol{\theta}}_T$ as $\widehat{\boldsymbol{\theta}}$. Let $\boldsymbol{G}_T^{\lambda_T,(i)}(\boldsymbol{\theta})$ denote the $i$-th entry of $\boldsymbol{G}_T^{\lambda_T}(\boldsymbol{\theta})$. By Taylor expansion, we have for each $i \in \{1, \ldots, d\}$ some $\tilde{\boldsymbol{\theta}}_i$ on the line segment between $\boldsymbol{\theta}^*$ and $\widehat{\boldsymbol{\theta}}$ such that

$$-\boldsymbol{G}_T^{\lambda_T,(i)}(\boldsymbol{\theta}^*) = \boldsymbol{G}_T^{\lambda_T,(i)}(\widehat{\boldsymbol{\theta}}) - \boldsymbol{G}_T^{\lambda_T,(i)}(\boldsymbol{\theta}^*) + o_p(1/\sqrt{T})$$

$$= \langle \nabla \boldsymbol{G}_T^{\lambda_T,(i)}(\boldsymbol{\theta}^*), \widehat{\boldsymbol{\theta}} - \boldsymbol{\theta}^* \rangle + \frac{1}{2}(\widehat{\boldsymbol{\theta}} - \boldsymbol{\theta}^*)^{\mathsf{T}} \nabla^2 \boldsymbol{G}_T^{\lambda_T,(i)}(\tilde{\boldsymbol{\theta}}_i)(\widehat{\boldsymbol{\theta}} - \boldsymbol{\theta}^*) + o_p(1/\sqrt{T}).$$

Stacking these expansions over entries $i = 1, \ldots, d$ gives

$$-\boldsymbol{G}_T^{\lambda_T}(\boldsymbol{\theta}^*) = \nabla \boldsymbol{G}_T^{\lambda_T}(\boldsymbol{\theta}^*)(\widehat{\boldsymbol{\theta}} - \boldsymbol{\theta}^*) + \frac{1}{2}\tilde{\boldsymbol{\delta}}(\widehat{\boldsymbol{\theta}} - \boldsymbol{\theta}^*) + o_p(1/\sqrt{T}),$$

where

$$\tilde{\boldsymbol{\delta}} = \begin{pmatrix} (\hat{\boldsymbol{\theta}} - \boldsymbol{\theta}^*)^{\top} \nabla^2 \boldsymbol{G}_T^{\lambda_T,(1)}(\tilde{\boldsymbol{\theta}}_1) \\ \vdots \\ (\hat{\boldsymbol{\theta}} - \boldsymbol{\theta}^*)^{\top} \nabla^2 \boldsymbol{G}_T^{\lambda_T,(d)}(\tilde{\boldsymbol{\theta}}_d) \end{pmatrix}.$$

Rearranging gives

$$[\nabla \boldsymbol{G}_T^{\lambda_T}(\boldsymbol{\theta}) + \frac{1}{2}\tilde{\boldsymbol{\delta}}]\sqrt{T}(\widehat{\boldsymbol{\theta}} - \boldsymbol{\theta}^*) = -\sqrt{T}\boldsymbol{G}_T^{\lambda_T}(\boldsymbol{\theta}^*) + o_p(1). \tag{13}$$

Since $\tilde{\boldsymbol{\theta}}_i$ is on the line segment between $\boldsymbol{\theta}^*$ and $\widehat{\boldsymbol{\theta}}$, we have by Lemmas B.1 and B.4 that for all $i$,

$$||\nabla^2 \boldsymbol{G}_T^{\lambda_T,(i)}(\tilde{\boldsymbol{\theta}}_i)||_{1,1} \le ||\nabla^2 \boldsymbol{G}_T^{\lambda_T}(\tilde{\boldsymbol{\theta}}_i)||_1$$

$$= ||\nabla^2 \boldsymbol{G}_T^{\lambda_T}(\tilde{\boldsymbol{\theta}}_i)||_1 1_{||\tilde{\boldsymbol{\theta}}_i - \boldsymbol{\theta}^*||_2 \le \epsilon_0} + ||\nabla^2 \boldsymbol{G}_T^{\lambda_T}(\tilde{\boldsymbol{\theta}}_i)||_1 1_{||\tilde{\boldsymbol{\theta}}_i - \boldsymbol{\theta}^*||_2 > \epsilon_0}$$

$$\le \sup_{||\boldsymbol{\theta} - \boldsymbol{\theta}^*|| \le \epsilon_0} ||\nabla^2 \boldsymbol{G}_T^{\lambda_T}(\boldsymbol{\theta})||_1 + ||\nabla^2 \boldsymbol{G}_T^{\lambda_T}(\tilde{\boldsymbol{\theta}}_i)||_1 1_{||\tilde{\boldsymbol{\theta}}_i - \boldsymbol{\theta}^*||_2 > \epsilon_0}.$$

where for a matrix $\boldsymbol{B} \in \mathbb{R}^{d_1 \times d_2}$, we define $||\boldsymbol{B}||_{1,1} = \sum_{i \in [d_1], j \in [d_2]} |\boldsymbol{B}_{i,j}|$. Note Lemma B.1 gives $\widehat{\boldsymbol{\theta}} - \boldsymbol{\theta}^* = o_p(1)$, and Lemma B.2 ensures convergence of $\nabla \boldsymbol{G}_T^{\lambda_T}(\boldsymbol{\theta}^*)$. Together they give

$$\nabla \boldsymbol{G}_T^{\lambda_T}(\boldsymbol{\theta}^*) + \frac{1}{2}\tilde{\boldsymbol{\delta}} \xrightarrow{p} \mathbb{E}[\nabla \boldsymbol{g}(\boldsymbol{X}_t, Y_t(a); \boldsymbol{\theta}^*)].$$

Applying this result and Lemma B.4 to (13) gives the desired result. $\square$

## A.2. Proof of Theorem 3.8

*Proof.* Using the linearity of the trace, we can write $\text{tr}(\boldsymbol{\Sigma}_{\lambda^*}) = a\lambda^2 + b\lambda + c$, where

$$a = \text{tr}\left( \boldsymbol{J}^{-1}(r\text{Cov}(\boldsymbol{g}_f^*) + E[\frac{1}{\bar{\pi}(a|\boldsymbol{X}_t)}\boldsymbol{g}_f^*\boldsymbol{g}_f^{*\mathsf{T}}] - E[\boldsymbol{g}_f^*]E[\boldsymbol{g}_f^{*\mathsf{T}}])\boldsymbol{J}^{-1} \right)$$

$$b = -\text{tr}\left( \boldsymbol{J}^{-1}\mathbb{E}[\frac{1}{\bar{\pi}(a|\boldsymbol{X}_t)}(\boldsymbol{g}_f^*\boldsymbol{g}_Y^{*\mathsf{T}} + \boldsymbol{g}_Y^*\boldsymbol{g}_f^{*\mathsf{T}})]\boldsymbol{J}^{-1} \right)$$

$$c = \text{tr}\left( \boldsymbol{J}^{-1}(\mathbb{E}[\boldsymbol{g}_Y^*\boldsymbol{g}_Y^{*\mathsf{T}}]\boldsymbol{J}^{-1} \right).$$

Since the minimizer is $\lambda^* = -\frac{b}{2a}$, the result follows. For consistency of $\hat{\lambda}_T$, by continuous mapping theorem, it suffices to show that

$$\boldsymbol{B}_T \xrightarrow{p} \mathbb{E}[\nabla \boldsymbol{g}(\boldsymbol{X}, Y(a); \boldsymbol{\theta}^*)]$$

$$\boldsymbol{C}_T \xrightarrow{p} \mathbb{E}\left[\frac{1}{\overline{\pi}(a)}(\boldsymbol{g}_f^* \boldsymbol{g}_Y^{*\mathsf{T}} + \boldsymbol{g}_Y^* \boldsymbol{g}_f^{*\mathsf{T}})\right]$$

$$\boldsymbol{D}_T \xrightarrow{p} \mathbb{E}\left[\boldsymbol{g}_f^* \boldsymbol{g}_f^{*\mathsf{T}}\right]$$

$$\boldsymbol{E}_T \xrightarrow{p} \mathbb{E}\left[\boldsymbol{g}_f^*\right]$$

$$\boldsymbol{F}_T \xrightarrow{p} \mathbb{E}\left[\frac{1}{\overline{\pi}(a)} \boldsymbol{g}_f^* \boldsymbol{g}_f^{*\mathsf{T}}\right].$$

We will show convergence of $\boldsymbol{B}_T$ and $\boldsymbol{F}_T$; the arguments for the rest are similar. First note

$$\boldsymbol{B}_T = \frac{1}{T}\sum_{t=1}^{T} \frac{1_{A_t=a}}{\pi_t(a)} \nabla \boldsymbol{g}(\boldsymbol{X}_t, Y_t(a); \boldsymbol{\theta}^*) - \frac{1}{T}\sum_{t=1}^{T} \frac{1_{A_t=a}}{\pi_t(a)}\left(\nabla \boldsymbol{g}(\boldsymbol{X}_t, Y_t(a); \boldsymbol{\theta}^*) - \nabla \boldsymbol{g}(\boldsymbol{X}_t, Y_t(a); \hat{\boldsymbol{\theta}}_t)\right),$$

so for convergence of $\boldsymbol{B}_T$, it suffices to show the following two facts:

$$\frac{1}{T}\sum_{t=1}^{T} \frac{1_{A_t=a}}{\pi_t(a)} \nabla \boldsymbol{g}(\boldsymbol{X}_t, Y_t(a); \boldsymbol{\theta}^*) \xrightarrow{p} \mathbb{E}[\nabla \boldsymbol{g}(\boldsymbol{X}, Y(a); \boldsymbol{\theta}^*)] \tag{14}$$

$$\frac{1}{T}\sum_{t=1}^{T} \frac{1_{A_t=a}}{\pi_t(a)}\left(\nabla \boldsymbol{g}(\boldsymbol{X}_t, Y_t(a); \boldsymbol{\theta}^*) - \nabla \boldsymbol{g}(\boldsymbol{X}_t, Y_t(a); \hat{\boldsymbol{\theta}}_t)\right) \xrightarrow{p} 0. \tag{15}$$

For (14), it suffices to show that for any $\boldsymbol{c}, \boldsymbol{c}' \in \mathbb{R}^d$, $\boldsymbol{c}^\mathsf{T} \frac{1}{T}\sum_{t=1}^{T} \frac{1_{A_t=a}}{\pi_t(a)} \nabla \boldsymbol{g}(\boldsymbol{X}_t, Y_t(a); \boldsymbol{\theta}^*) \boldsymbol{c}' \xrightarrow{p} \boldsymbol{c}^\mathsf{T} \mathbb{E}[\nabla \boldsymbol{g}(\boldsymbol{X}, Y(a); \boldsymbol{\theta}^*)] \boldsymbol{c}'$. To that end, define $Z_t = \boldsymbol{c}^\mathsf{T} \frac{1_{A_t=a}}{\pi_t(a)} \nabla \boldsymbol{g}(\boldsymbol{X}_t, Y_t(a); \boldsymbol{\theta}^*) \boldsymbol{c}'$ and let $Z = \boldsymbol{c}^\mathsf{T} \frac{1}{\pi_{\min}} \nabla \boldsymbol{g}(\boldsymbol{X}_t, Y_t(a); \boldsymbol{\theta}^*) \boldsymbol{c}'$. Note $\mathbb{E}|A| < \infty$ and $|Z_t| \le |Z|$, so $\mathbb{P}(|Z_t| > x) \le \mathbb{P}(|Z| > x)$ for all $x$. Thus by Theorem 2.19 from Hall & Heyde (1980),

$$\frac{1}{T}\sum_{t=1}^{T}(Z_t - \mathbb{E}[Z_t|\mathcal{H}_{t-1}]) \xrightarrow{p} 0.$$

Since $\mathbb{E}[Z_t|\mathcal{H}_{t-1}] = \boldsymbol{c}^\mathsf{T} \mathbb{E}[\nabla \boldsymbol{g}(\boldsymbol{X}_t, Y_t(a); \boldsymbol{\theta}^*)] \boldsymbol{c}$, this implies

$$\frac{1}{T}\sum_{t=1}^{T} Z_t \xrightarrow{p} \boldsymbol{c}^\mathsf{T} \mathbb{E}[\nabla \boldsymbol{g}(\boldsymbol{X}_t, Y_t(a); \boldsymbol{\theta}^*)] \boldsymbol{c}',$$

establishing (14). For (15), we can apply an argument from Guo & Xu (2025): Observe for all $i$,

$$\left\|\frac{1}{T}\sum_{t=1}^{T} \frac{1_{A_t=a}}{\pi_t(a)}\left(\nabla \boldsymbol{g}^{(i)}(\boldsymbol{X}_t, Y_t(a); \boldsymbol{\theta}^*) - \nabla \boldsymbol{g}^{(i)}(\boldsymbol{X}_t, Y_t(a); \hat{\boldsymbol{\theta}}_t)\right)\right\|_2$$

$$\le \frac{1}{T}\sum_{t=1}^{T} \frac{1_{A_t=a}}{\pi_t(a)} \left\|\nabla \boldsymbol{g}^{(i)}(\boldsymbol{X}_t, Y_t(a); \boldsymbol{\theta}^*) - \nabla \boldsymbol{g}^{(i)}(\boldsymbol{X}_t, Y_t(a); \hat{\boldsymbol{\theta}}_t)\right\|_2$$

$$\le \frac{1}{\pi_{\min}T}\sum_{t=1}^{T} \left\|\int_0^1 \nabla^2 \boldsymbol{g}^{(i)}(\boldsymbol{X}_t, Y_t(a); \boldsymbol{\theta}^* + u(\hat{\boldsymbol{\theta}}_T - \boldsymbol{\theta}^*))\mathrm{d}u \cdot (\hat{\boldsymbol{\theta}}_T - \boldsymbol{\theta}^*)\right\|_2$$

$$\le \frac{1}{\pi_{\min}T}\left(\sum_{t=1}^{T} \Phi(\boldsymbol{X}_t, Y_t(a))\right) \cdot \left\|\hat{\boldsymbol{\theta}}_T - \boldsymbol{\theta}^*\right\|_2$$

$$+1_{||\widehat{\boldsymbol{\theta}}_T - \boldsymbol{\theta}^*||_2 > \epsilon_0} \cdot \frac{1}{\pi_{\min} T} \sum_{t=1}^{T} \left\| \int_0^1 \nabla^2 \boldsymbol{g}^{(i)}(\boldsymbol{X}_t, Y_t(a); \boldsymbol{\theta}^*) + u(\widehat{\boldsymbol{\theta}}_T - \boldsymbol{\theta}^*) \mathrm{d}u \cdot (\widehat{\boldsymbol{\theta}}_T - \boldsymbol{\theta}^*) \right\|_2$$

$$\xrightarrow{p} 0$$

by Assumptions 3.6 and 3.2. This implies (15), establishing convergence of $\boldsymbol{B}_T$. Similarly, to show convergence of $\boldsymbol{F}_T$, it suffices to show

$$\frac{1}{T} \sum_{t=1}^{T} \frac{1_{A_t=a}}{\pi_t(a)^2} \boldsymbol{g}_{f,t}^* \boldsymbol{g}_{f,t}^{*\mathsf{T}} \xrightarrow{p} \mathbb{E}\left[ \frac{1}{\bar{\pi}(a)} \boldsymbol{g}_f^* \boldsymbol{g}_f^{*\mathsf{T}} \right] \tag{16}$$

$$\frac{1}{T} \sum_{t=1}^{T} \frac{1_{A_t=a}}{\pi_t(a)^2} \left( \boldsymbol{g}_{f,t}^* \boldsymbol{g}_{f,t}^{*\mathsf{T}} - \hat{\boldsymbol{g}}_{f,t}^{(T)} \hat{\boldsymbol{g}}_{f,t}^{(T)\mathsf{T}} \right) \xrightarrow{p} 0. \tag{17}$$

For (16), note it suffices to show that $\frac{1}{T} \sum_{t=1}^{T} \boldsymbol{c}^\mathsf{T} \frac{1_{A_t=a}}{\pi_t(a)^2} \boldsymbol{g}_{f,t}^* \boldsymbol{g}_{f,t}^{*\mathsf{T}} \boldsymbol{c}' \xrightarrow{p} \boldsymbol{c}^\mathsf{T} \mathbb{E}\left[ \frac{1}{\bar{\pi}(a)} \boldsymbol{g}_f^* \boldsymbol{g}_f^{*\mathsf{T}} \right] \boldsymbol{c}'$ for any $\boldsymbol{c}, \boldsymbol{c}' \in \mathbb{R}^d$. Let $Z_t = \boldsymbol{c}^\mathsf{T} \frac{1_{A_t=a}}{\pi_t(a)^2} \boldsymbol{g}_{f,t}^* \boldsymbol{g}_{f,t}^{*\mathsf{T}} \boldsymbol{c}'$ and $Z = \boldsymbol{c}^\mathsf{T} \frac{1}{\pi_{\min}} \boldsymbol{g}_{f,t}^* \boldsymbol{g}_{f,t}^{*\mathsf{T}} \boldsymbol{c}'$. Note $|Z_t| \le |Z|$ clearly, and $E|Z| < \infty$ by Assumption 3.5. Thus by Theorem 2.19 from Hall & Heyde (1980),

$$\frac{1}{T} \sum_{t=1}^{T} (Z_t - \mathbb{E}[Z_t | \mathcal{H}_{t-1}]) \xrightarrow{p} 0.$$

Since $\mathbb{E}[Z_t | \mathcal{H}_{t-1}] = \boldsymbol{c}^\mathsf{T} \mathbb{E}\left[ \frac{1}{\bar{\pi}(a)} \boldsymbol{g}_f^* \boldsymbol{g}_f^{*\mathsf{T}} \right] \boldsymbol{c}'$, this implies

$$\frac{1}{T} \sum_{t=1}^{T} Z_t \xrightarrow{p} \boldsymbol{c}^\mathsf{T} \mathbb{E}\left[ \frac{1}{\bar{\pi}(a)} \boldsymbol{g}_f^* \boldsymbol{g}_f^{*\mathsf{T}} \right] \boldsymbol{c}',$$

which establishes (16). For (17), we can again apply an argument from (Guo & Xu, 2025): Note

$$\left\| \frac{1}{T} \sum_{t=1}^{T} \frac{1_{A_t=a}}{\pi_t(a)^2} \left( \boldsymbol{g}_{f,t}^* \boldsymbol{g}_{f,t}^{*\mathsf{T}} - \hat{\boldsymbol{g}}_{f,t}^{(T)} \hat{\boldsymbol{g}}_{f,t}^{(T)\mathsf{T}} \right) \right\|_2$$

$$\le \frac{1}{T} \cdot \frac{1}{\pi_{\min}^2} \sum_{t=1}^{T} \left\| \boldsymbol{g}_{f,t}^* \boldsymbol{g}_{f,t}^{*\mathsf{T}} - \hat{\boldsymbol{g}}_{f,t}^{(T)} \hat{\boldsymbol{g}}_{f,t}^{(T)\mathsf{T}} \right\|_2$$

$$\le \frac{1}{\pi_{\min}^2 T} \sum_{t=1}^{T} \left( ||\widehat{\boldsymbol{g}}_{f,t}^{(T)} \widehat{\boldsymbol{g}}_{f,t}^{(T)\mathsf{T}} - \widehat{\boldsymbol{g}}_{f,t}^{(T)} \widehat{\boldsymbol{g}}_{f,t}^{*\mathsf{T}}||_2 + ||\widehat{\boldsymbol{g}}_{f,t}^{(T)} \widehat{\boldsymbol{g}}_{f,t}^{*\mathsf{T}} - \widehat{\boldsymbol{g}}_{f,t}^* \widehat{\boldsymbol{g}}_{f,t}^{*\mathsf{T}}||_2 \right)$$

$$\le \frac{1}{\pi_{\min}^2 T} \sum_{t=1}^{T} ||\widehat{\boldsymbol{g}}_{f,t}^{(T)} - \boldsymbol{g}_{f,t}^*||_2 \cdot (||\widehat{\boldsymbol{g}}_{f,t}^{(T)}||_2 + ||\boldsymbol{g}_{f,t}^*||_2)$$

$$\le \frac{1}{\pi_{\min}^2 T} \sum_{t=1}^{T} ||\widehat{\boldsymbol{g}}_{f,t}^{(T)} - \boldsymbol{g}_{f,t}^*||_2 \cdot (||\widehat{\boldsymbol{g}}_{f,t}^{(T)} - \boldsymbol{g}_{f,t}^*||_2 + 2||\boldsymbol{g}_{f,t}^*||_2)$$

$$\le \frac{1}{\pi_{\min}^2 T} \sum_{t=1}^{T} \phi(\boldsymbol{X}_t, Y_t(a)) ||\widehat{\boldsymbol{\theta}}_T - \boldsymbol{\theta}^*||_2 [\phi(\boldsymbol{X}_t, Y_t(a)) ||\widehat{\boldsymbol{\theta}}^{(T)} - \boldsymbol{\theta}^*||_2 + 2||\boldsymbol{g}_{f,t}^*||_2]$$

$$= \frac{1}{\pi_{\min}^2 T} \left[ \sum_{t=1}^{T} \phi(\boldsymbol{X}_t, Y_t(a))^2 \right] ||\widehat{\boldsymbol{\theta}}_T - \boldsymbol{\theta}^*||_2^2 + \frac{2}{\pi_{\min}^2 T} \left[ \sum_{t=1}^{T} \phi(\boldsymbol{X}_t, Y_t(a)) ||\boldsymbol{g}(\boldsymbol{X}_t, Y_t(a); \boldsymbol{\theta}^*||_2 \right] \cdot ||\widehat{\boldsymbol{\theta}}_T - \boldsymbol{\theta}^*||_2$$

$$\le \frac{1}{\pi_{\min}^2 T} \left[ \sum_{t=1}^{T} \phi(\boldsymbol{X}_t, Y_t(a))^2 \right] \cdot ||\widehat{\boldsymbol{\theta}}_T - \boldsymbol{\theta}^*||_2^2 + \frac{2}{\pi_{\min}^2} \sqrt{\frac{1}{T} \sum_{t=1}^{T} \phi(\boldsymbol{X}_t, Y_t(a))^2 \cdot \frac{1}{T} \sum_{t=1}^{T} ||\boldsymbol{g}(\boldsymbol{X}_t, Y_t(a); \boldsymbol{\theta}^*))||_2^2} \cdot ||\widehat{\boldsymbol{\theta}}_T - \boldsymbol{\theta}^*||_2$$

$$\xrightarrow{p} 0.$$

The fifth inequality follows from Assumption 3.6. The last inequality uses Cauchy-Schwarz, and the last equality holds because $\frac{1}{T}\sum_{t=1}^{T}\phi(\boldsymbol{X}_t, Y_t(a))^2 = \mathcal{O}_p(1)$ and $\frac{1}{T}\sum_{t=1}^{T}||\boldsymbol{g}(\boldsymbol{X}_t, Y_t(a); \boldsymbol{\theta}^*)||_2^2 = \mathcal{O}_p(1)$ (by law of large numbers), and $||\widehat{\boldsymbol{\theta}}_T - \boldsymbol{\theta}^*||_2 \xrightarrow{p} 0$ by Lemma B.1. This establishes (17), completing the proof of convergence of $\boldsymbol{F}_T$.

$\square$

### A.3. Proof of Corollary 3.10

*Proof.* By Theorem 3.8, $\hat{\lambda}_T \xrightarrow{p} \lambda^*$. Thus if $\lambda_T = \hat{\lambda}_T$, then by Theorem 3.7, the asymptotic variance of $\widehat{\boldsymbol{\theta}}_T$ is $\boldsymbol{\Sigma}_{\lambda^*}$, which has minimal trace by definition of $\lambda^*$. $\square$

## B. Lemmas

**Lemma B.1** (Existence, boundedness, consistency of $\widehat{\boldsymbol{\theta}}_T$)**.** *Under the assumptions of Theorem 3.7, there exists a sequence of estimators $\{\widehat{\boldsymbol{\theta}}_T\}_{T\geq 1}$ such that (8) holds, and $||\widehat{\boldsymbol{\theta}}_T||_2 \leq R_\theta$ for all $T$. Moreover, for any such sequence, as $T \to \infty$, $\widehat{\boldsymbol{\theta}}_T \xrightarrow{p} \boldsymbol{\theta}^*$.*

*Proof.* We first show that for $R_\theta$ from Assumption 3.5, as $T \to \infty$,

$$\sup_{||\boldsymbol{\theta}||_2 \leq R_\theta} ||\boldsymbol{G}_T^{\lambda_T}(\boldsymbol{\theta}) - \mathbb{E}\boldsymbol{G}_T^{\lambda_T}(\boldsymbol{\theta})||_2 \xrightarrow{p} 0. \tag{18}$$

It suffices to show the coordinate-wise result for each $i = 1, \ldots, d$:

$$\sup_{||\boldsymbol{\theta}||_2 \leq R_\theta} |\boldsymbol{G}_T^{\lambda_T,(i)}(\boldsymbol{\theta}) - \mathbb{E}\boldsymbol{G}_T^{\lambda_T,(i)}(\boldsymbol{\theta})| \xrightarrow{p} 0. \tag{19}$$

Fix $\epsilon > 0$ and let $\Theta_\epsilon = \{\boldsymbol{\theta}_j : j = 1, \ldots, N_\epsilon\}$ be an $\epsilon$-net of the set $\Theta := \overline{\mathcal{B}(\boldsymbol{0}, R_\theta)}$ with finite cardinality $N_\epsilon$. By Assumption 3.6, we have

$$|\boldsymbol{g}^{(i)}(\boldsymbol{X}_t, Y_t(a); \boldsymbol{\theta}) - \boldsymbol{g}^{(i)}(\boldsymbol{X}_t, Y_t(a); \boldsymbol{\theta}_j)| \leq ||\boldsymbol{g}(\boldsymbol{X}_t, Y_t(a); \boldsymbol{\theta}) - \boldsymbol{g}(\boldsymbol{X}_t, Y_t(a); \boldsymbol{\theta}_j)||_2$$

$$\leq \left|\left|\int_0^1 \nabla\boldsymbol{g}(\boldsymbol{X}_t, Y_t(a); \boldsymbol{\theta}_j + u(\boldsymbol{\theta} - \boldsymbol{\theta}_j))du \cdot (\boldsymbol{\theta} - \boldsymbol{\theta}_j)\right|\right|_2$$

$$\leq \phi(\boldsymbol{X}_t, Y_t(a)) \cdot ||\boldsymbol{\theta} - \boldsymbol{\theta}_j||_2$$

$$\leq \epsilon\phi(\boldsymbol{X}_t, Y_t(a)).$$

(Here $\boldsymbol{g}^{(i)}(\boldsymbol{X}_t, Y_t(a); \boldsymbol{\theta})$ denotes the $i$th entry of $\boldsymbol{g}(\boldsymbol{X}_t, Y_t(a); \boldsymbol{\theta})$.) Thus,

$$l_j(\boldsymbol{X}_t, Y_t(a)) \leq \boldsymbol{g}^{(i)}(\boldsymbol{X}_t, Y_t(a); \boldsymbol{\theta}) \leq u_j(\boldsymbol{X}_t, Y_t(a)) \tag{20}$$

where we define $l_j(\boldsymbol{X}_t, Y_t(a)) = \boldsymbol{g}^{(i)}(\boldsymbol{X}_t, Y_t(a); \boldsymbol{\theta}_j) - \epsilon\phi(\boldsymbol{X}_t, Y_t(a))$ and $u_j(\boldsymbol{X}_t, Y_t(a)) = \boldsymbol{g}^{(i)}(\boldsymbol{X}_t, Y_t(a); \boldsymbol{\theta}_j) + \epsilon\phi(\boldsymbol{X}_t, Y_t(a))$. Now notice that, for all $t \in [T]$,

$$\mathbb{E}\boldsymbol{G}_T(\boldsymbol{\theta}) = \mathbb{E}[\boldsymbol{g}(\boldsymbol{X}_t, Y_t(a); \boldsymbol{\theta})] = \mathbb{E}\left[\frac{1_{\{A_t=a\}}}{\pi_t(a)}\boldsymbol{g}(\boldsymbol{X}_t, Y_t; \boldsymbol{\theta})|\mathcal{H}_{t-1}\right]. \tag{21}$$

Combining this with (20) gives

$$\sup_{\boldsymbol{\theta}\in\Theta} |\boldsymbol{G}_T^{\lambda_T,(i)}(\boldsymbol{\theta}) - \mathbb{E}\boldsymbol{G}_T^{\lambda_T,(i)}(\boldsymbol{\theta})|$$

$$= \sup_{\boldsymbol{\theta}\in\Theta}\left|\frac{1}{T}\sum_{t=1}^{T}\frac{1_{\{A_t=a\}}}{\pi_t(a)}\boldsymbol{g}^{(i)}(\boldsymbol{X}_t, Y_t; \boldsymbol{\theta}) + \frac{1}{N}\sum_{i=1}^{N}\lambda_T\boldsymbol{g}^{(i)}(\tilde{\boldsymbol{X}}_i, f(\tilde{\boldsymbol{X}}_i), \boldsymbol{\theta}) - \frac{1}{T}\sum_{t=1}^{T}\frac{1_{\{A_t=a\}}}{\pi_t(a)}\lambda_T\boldsymbol{g}^{(i)}(\boldsymbol{X}_t, f(\boldsymbol{X}_t), \boldsymbol{\theta})\right.$$

$$\left.-\mathbb{E}\left[\frac{1_{\{A_t=a\}}}{\pi_t(a)}\boldsymbol{g}^{(i)}(\boldsymbol{X}_t, Y_t; \boldsymbol{\theta}) \mid \mathcal{H}_{t-1}\right]\right| \tag{22}$$

$$\leq \sup_{\boldsymbol{\theta} \in \Theta} |A_T| + \lambda_T \sup_{\boldsymbol{\theta} \in \Theta} |B_N| + \lambda_T \sup_{\boldsymbol{\theta} \in \Theta} |C_T|,$$

where

$$A_T = \frac{1}{T} \sum_{t=1}^{T} \left( \frac{1_{A_t=a}}{\pi_t(a)} \boldsymbol{g}^{(j)}(\boldsymbol{X}_t, Y_t; \boldsymbol{\theta}) - \mathbb{E}\left[ \frac{1_{A_t=a}}{\pi_t(a)} \boldsymbol{g}^{(j)}(\boldsymbol{X}_t, Y_t; \boldsymbol{\theta}) | \mathcal{H}_{t-1} \right] \right)$$

$$B_N = \frac{1}{N} \sum_{i=1}^{N} \boldsymbol{g}^{(j)}(\tilde{\boldsymbol{X}}_i, f(\tilde{\boldsymbol{X}}_i); \theta) - \mathbb{E}[\boldsymbol{g}^{(j)}(\boldsymbol{X}, f(\boldsymbol{X}); \boldsymbol{\theta})]$$

$$C_T = \frac{1}{T} \sum_{t=1}^{T} \left( \mathbb{E}\left[ \frac{1_{A_t=a}}{\pi_t(a)} \boldsymbol{g}^{(j)}(\boldsymbol{X}_t, f(\boldsymbol{X}_t); \boldsymbol{\theta}) | \mathcal{H}_{t-1} \right] - \frac{1_{A_t=a}}{\pi_t(a)} \boldsymbol{g}^{(j)}(\boldsymbol{X}_t, f(\boldsymbol{X}_t); \boldsymbol{\theta}) \right).$$

Since $\lambda_T \xrightarrow{p} \overline{\lambda}$, it suffices to show that $\sup_{\theta \in \Theta} |A_T|, \sup_{\theta \in \Theta} |B_N|, \sup_{\theta \in \Theta} |C_T| \xrightarrow{p} 0$. The proof for $A_T$ is provided in Lemma B.1 of Guo & Xu (2025), but we reproduce it here for completeness. Note

$$\sup_{\boldsymbol{\theta} \in \Theta} A_T \leq \max_{k \in [N_\epsilon]} \frac{1}{T} \sum_{t=1}^{T} \left\{ \frac{1_{A_t=a}}{\pi_t(a)} u_k(\boldsymbol{X}_t, Y_t(a)) - \mathbb{E}\left[ \frac{1_{A_t=a}}{\pi_t(a)} l_k(\boldsymbol{X}_t, Y_t(a)) | \mathcal{H}_{t-1} \right] \right\}$$

$$\leq \Delta_1 + \Delta_2, \tag{23}$$

where

$$\Delta_1 = \max_{k \in [N_\epsilon]} \frac{1}{T} \sum_{t=1}^{T} \left\{ \frac{1_{A_t=a}}{\pi_t(a)} u_k(\boldsymbol{X}_t, Y_t(a)) - \mathbb{E}\left[ \frac{1_{A_t=a}}{\pi_t(a)} u_k(\boldsymbol{X}_t, Y_t(a)) | \mathcal{H}_{t-1} \right] \right\}$$

$$\Delta_2 = \max_{k \in [N_\epsilon]} \frac{1}{T} \sum_{t=1}^{T} \mathbb{E}\left[ \frac{1_{A_t=a}}{\pi_t(a)} (u_k(\boldsymbol{X}_t, Y_t(a)) - l_k(\boldsymbol{X}_t, Y_t(a))) | \mathcal{H}_{t-1} \right].$$

We first study $\Delta_1$. Observe

$$\Delta_1 \leq \sum_{k \in [N_\epsilon]} \frac{1}{T} \sum_{t=1}^{T} \left\{ \frac{1_{A_t=a}}{\pi_t(a)} u_k(\boldsymbol{X}_t, Y_t(a)) - \mathbb{E}\left[ \frac{1_{A_t=a}}{\pi_t(a)} u_k(\boldsymbol{X}_t, Y_t(a)) | \mathcal{H}_{t-1} \right] \right\}.$$

For all $\epsilon' > 0$,

$$\mathbb{P}\left( \left| \frac{1}{T} \sum_{t=1}^{T} \left\{ \frac{1_{A_t=a}}{\pi_t(a)} u_j(\boldsymbol{X}_t, Y_t(a)) - \mathbb{E}\left[ \frac{1_{A_t=a}}{\pi_t(a)} u_j(\boldsymbol{X}_t, Y_t(a)) | \mathcal{H}_{t-1} \right] \right\} \right| > \epsilon' \right)$$

$$\leq \frac{1}{T^2 \epsilon'^2} \mathbb{E}\left( \sum_{t=1}^{T} \left\{ \frac{1_{A_t=a}}{\pi_t(a)} u_k(\boldsymbol{X}_t, Y_t(a)) - \mathbb{E}\left[ \frac{1_{A_t=a}}{\pi_t(a)} u_k(\boldsymbol{X}_t, Y_t(a)) | \mathcal{H}_{t-1} \right] \right\} \right)^2$$

$$= \frac{1}{T^2 \epsilon'^2} \sum_{t=1}^{T} \mathbb{E}\left( \frac{1_{A_t=a}}{\pi_t(a)} u_k(\boldsymbol{X}_t, Y_t(a)) - \mathbb{E}\left[ \frac{1_{A_t=a}}{\pi_t(a)} u_k(\boldsymbol{X}_t, Y_t(a)) | \mathcal{H}_{t-1} \right] \right)^2$$

$$\leq \frac{1}{T^2 \epsilon'^2} \sum_{t=1}^{T} \mathbb{E}\left( \frac{1_{A_t=a}}{\pi_t(a)} u_k(\boldsymbol{X}_t, Y_t(a)) \right)^2$$

$$\leq \frac{1}{T^2 \epsilon'^2 \pi_{\min}^2} \sum_{t=1}^{T} \mathbb{E} u_j^2(\boldsymbol{X}_t, Y_t(a))$$

$$= \frac{1}{T^2 \epsilon'^2} \sum_{t=1}^{T} \frac{1}{\pi_{\min}^2} \mathbb{E}(\boldsymbol{g}^{(j)}(\boldsymbol{X}_t, Y_t(a); \boldsymbol{\theta}_j) + \epsilon \phi(\boldsymbol{X}_t, Y_t(a)))^2$$

$$\leq \frac{1}{\pi_{\min}^2 T^2 \epsilon'^2} \cdot T \mathbb{E}[2(\boldsymbol{g}^{(i)}(\boldsymbol{X}_1, Y_1(a); \boldsymbol{\theta}_k))^2 + 2\epsilon^2 \phi^2(\boldsymbol{X}_1, Y_1(a))]$$

$$\leq \frac{2M_2' + \epsilon^2 M_\phi}{\pi_{\min}^2 T \epsilon'^2} \xrightarrow{p} 0,$$

where $M_2' = \sup_{||\boldsymbol{\theta}||_2 \leq R_\theta} \mathbb{E}||\boldsymbol{g}(\boldsymbol{X}_t, Y_t(a); \boldsymbol{\theta})||_2^2$, $M_\phi = \mathbb{E}\phi(X_t, Y_t(a))^2$. The first inequality follows from Chebyshev, and the first equality uses the following fact: if $u_{j,t} = \frac{1_{A_t=a}}{\pi_t(a)} u_k(\boldsymbol{X}_t, Y_t(a))$. Then for $t_1 < t_2$,

$$\mathbb{E}[u_{j,t_1} - \mathbb{E}[u_{j,t_1}|\mathcal{H}_{t_1-1}])(u_{j,t_2} - \mathbb{E}[u_{j,t_2}|\mathcal{H}_{t_2-1}])]$$

$$= \mathbb{E}[\mathbb{E}[(u_{j,t_1} - \mathbb{E}[u_{j,t_1}|\mathcal{H}_{t_1-1}])(u_{j,t_2} - \mathbb{E}[u_{j,t_2}|\mathcal{H}_{t_2-1}])|\mathcal{H}_{t-1}]$$

$$= \mathbb{E}[(u_{j,t_1} - \mathbb{E}[u_{j,t_1}|\mathcal{H}_{t_1-1}]) \cdot \mathbb{E}[u_{j,t_2}|\mathcal{H}_{t_2-1}]|\mathcal{H}_{t_1-1}]$$

$$= \mathbb{E}[(u_{j,t_1} - \mathbb{E}[u_{j,t_1}|\mathcal{H}_{t_1-1}]) \cdot 0] = 0.$$

The penultimate inequality uses that $(a + b)^2 \leq 2(a^2 + b^2)$ for any $a, b \in \mathbb{R}$, and the final inequality uses Assumptions 3.5 and 3.6, which implies $M_2' < \infty$ and $M_\infty < \infty$. This establishes that $\Delta_1 \xrightarrow{p} 0$. We now study $\Delta_2$. Note

$$\Delta_2 \leq \max_{k \in [N_\epsilon]} \frac{1}{T} \sum_{t=1}^{T} \frac{1}{\pi_{\min}} \mathbb{E}\left[u_j(\boldsymbol{X}_t, Y_t(a)) - l_j(\boldsymbol{X}_t, Y_t(a))|\mathcal{H}_{t-1}\right]$$

$$\leq \max_{j \in [N_\epsilon]} \frac{1}{T} \sum_{t=1}^{T} \frac{1}{\pi_{\min}} \mathbb{E}\left[2\epsilon\phi(\boldsymbol{X}_t, Y_t(a))|\mathcal{H}_{t-1}\right]$$

$$= \frac{2\epsilon}{\pi_{\min}} \mathbb{E}\phi(\boldsymbol{X}_1, Y_1(a))$$

$$\leq \frac{2\epsilon}{\pi_{\min}} \sqrt{\mathbb{E}\phi(\boldsymbol{X}_1, Y_1(a))^2}$$

$$\leq \frac{2\epsilon\sqrt{M_\phi}}{\pi_{\min}}.$$

Plugging in the analysis of $\Delta_1$ and $\Delta_2$ into (23) gives that for all $\epsilon > 0$,

$$\mathbb{P}\left(\sup_{\theta \in \Theta} A_T > \frac{2\epsilon\sqrt{M_\phi}}{\pi_{\min}}\right) \xrightarrow{p} 0.$$

This implies that for all $\epsilon > 0$,

$$\mathbb{P}\left(\sup_{\theta \in \Theta} A_T > \epsilon\right) \xrightarrow{p} 0.$$

By a similar argument,

$$\mathbb{P}\left(\sup_{\theta \in \Theta} -A_T > \epsilon\right) \xrightarrow{p} 0,$$

so

$$\sup_{\theta \in \Theta} |A_T| \xrightarrow{p} 0.$$

Similar reasoning shows $\sup_{\theta \in \Theta} |B_N| \xrightarrow{p} 0$ and $\sup_{\theta \in \Theta} |C_T| \xrightarrow{p} 0$. Thus by (21),

$$\sup_{\boldsymbol{\theta} \in \Theta} |\boldsymbol{G}_T^{\lambda_T,(i)}(\boldsymbol{\theta}) - \mathbb{E}\boldsymbol{G}_T^{\lambda_T,(i)}(\boldsymbol{\theta})| \xrightarrow{p} 0.$$

So (19) holds and therefore (18) holds.

We now focus on the main results. Define

$$\widehat{\boldsymbol{\theta}}_T = \underset{||\boldsymbol{\theta}||_2 \leq R_\theta}{\arg\min} ||\boldsymbol{G}_T^{\lambda_T}(\boldsymbol{\theta})||_2$$

and

$$\tilde{\boldsymbol{\theta}}_T = \boldsymbol{\theta}^* - [\nabla\mathbb{E}\boldsymbol{g}(\boldsymbol{X}_t, Y_t(a); \boldsymbol{\theta}^*)]^{-1} \boldsymbol{G}_T^{\lambda_T}(\boldsymbol{\theta}^*).$$

Note Lemma B.4 implies
$$\tilde{\boldsymbol{\theta}}_T - \boldsymbol{\theta}^* = O_p(1/\sqrt{T}) \tag{24}$$

Moreover, Taylor expansion gives
$$\boldsymbol{G}_T^{\lambda_T}(\tilde{\boldsymbol{\theta}}_a) = \boldsymbol{G}_T^{\lambda_T}(\boldsymbol{\theta}^*) + \nabla \boldsymbol{G}_T^{\lambda_T}(\boldsymbol{\theta}^*)(\tilde{\boldsymbol{\theta}}_a - \boldsymbol{\theta}^*) + o_p(||\tilde{\boldsymbol{\theta}}_T - \boldsymbol{\theta}^*||_2)$$

$$= \nabla G_T^{\lambda_T}(\boldsymbol{\theta}^*) - \nabla \boldsymbol{G}_T^{\lambda_T}(\boldsymbol{\theta}^*)[\nabla \mathbb{E}\boldsymbol{g}(\boldsymbol{X}_t, Y_t(a); \boldsymbol{\theta}^*)]^{-1} \boldsymbol{G}_T^{\lambda_T}(\boldsymbol{\theta}^*) + o_p(1/\sqrt{T}).$$

$$= \boldsymbol{G}_T^{\lambda_T}(\boldsymbol{\theta}^*) - (1 + o_p(1))\boldsymbol{G}_T^{\lambda_T}(\boldsymbol{\theta}^*) + o_p(1/\sqrt{T})$$

$$= o_p(1/\sqrt{T}). \tag{25}$$

The second equality follows from the definition of $\tilde{\theta}_T$ and (24). The third follows from Lemma B.2 and the fact that, by (21 and Assumption 3.6), $\nabla \boldsymbol{G}(\boldsymbol{\theta}^*) = \frac{\partial}{\partial \boldsymbol{\theta}} \mathbb{E}\boldsymbol{G}_T^{\lambda_T}(\boldsymbol{\theta})_{\boldsymbol{\theta}=\boldsymbol{\theta}^*} = \frac{\partial}{\partial \boldsymbol{\theta}} \mathbb{E}\boldsymbol{g}(\boldsymbol{X}_t, Y_t(a); \boldsymbol{\theta})_{\boldsymbol{\theta}=\boldsymbol{\theta}^*} = \mathbb{E}\frac{\partial}{\partial \boldsymbol{\theta}} \boldsymbol{g}(\boldsymbol{X}_t, Y_t(a); \boldsymbol{\theta})_{\boldsymbol{\theta}=\boldsymbol{\theta}^*}$ is nonsingular. The last equality follows from Lemma B.4.

Combining (24) (which implies $\mathbb{P}(||\tilde{\boldsymbol{\theta}}_T||_2 > R_\theta \to 0)$) and (25), we have
$$||\boldsymbol{G}_T^{\lambda_T}(\widehat{\boldsymbol{\theta}}_T)||_2 = ||\boldsymbol{G}_T^{\lambda_T}(\widehat{\boldsymbol{\theta}}_a)||_2 \mathbb{1}_{\{||\tilde{\boldsymbol{\theta}}_T||_2 \leq R_\theta\}} + ||\boldsymbol{G}_T^{\lambda_T}(\widehat{\boldsymbol{\theta}}_a)||_2 \mathbb{1}_{\{||\tilde{\boldsymbol{\theta}}_T||_2 > R_\theta\}}$$

$$\leq ||\boldsymbol{G}_T^{\lambda_T}(\widehat{\boldsymbol{\theta}}_a)||_2 + o_p(1/\sqrt{T})$$

$$= o_p(1/\sqrt{T}).$$

Thus this choice of $\widehat{\boldsymbol{\theta}}_T$ satisfies (8). This completes the existence proof.

Lastly, we show consistency. Suppose $\widehat{\boldsymbol{\theta}}_T$ satisfies $||\widehat{\boldsymbol{\theta}}_T|| \leq R_\theta$ and (8). By Assumption 3.4, for any $\epsilon > 0$,
$$\delta_\epsilon := \inf_{||\boldsymbol{\theta}-\boldsymbol{\theta}^*||_2 > \epsilon} ||\mathbb{E}\boldsymbol{g}(\boldsymbol{X}_t, Y_t(a); \boldsymbol{\theta})||_2 > 0.$$

From (21), we have
$$\inf_{||\boldsymbol{\theta}-\boldsymbol{\theta}^*||_2 > \epsilon} ||\mathbb{E}\boldsymbol{G}_T^{\lambda_T}(\boldsymbol{\theta})||_2 = \inf_{||\boldsymbol{\theta}-\boldsymbol{\theta}^*||_2 > \epsilon} ||\mathbb{E}\boldsymbol{g}(\boldsymbol{X}_t, Y_t(a); \boldsymbol{\theta})||_2 \geq \delta_\epsilon.$$

Combining this with (18) gives
$$\inf_{\boldsymbol{\theta} \in \boldsymbol{\Theta}, ||\boldsymbol{\theta}-\boldsymbol{\theta}^*||_2 > \epsilon} ||\boldsymbol{G}_T^{\lambda_T}(\boldsymbol{\theta})||_2 \geq \inf_{\boldsymbol{\theta}-\boldsymbol{\theta}^*||_2 > \epsilon} ||\mathbb{E}\boldsymbol{G}_T^{\lambda_T}(\boldsymbol{\theta})||_2 - \sup_{\boldsymbol{\theta} \in \Theta} ||\boldsymbol{G}_T^{\lambda_T}(\boldsymbol{\theta}) - \mathbb{E}\boldsymbol{G}_T^{\lambda_T}(\boldsymbol{\theta})||_2 \geq \delta_\epsilon - o_p(1)$$

where $\Theta = \overline{\mathcal{B}(\boldsymbol{0}, R_\theta)}$. This implies
$$\lim_{T\to\infty} \mathbb{P}\left(\inf_{\boldsymbol{\theta} \in \boldsymbol{\Theta}, ||\boldsymbol{\theta}-\boldsymbol{\theta}^*||_2 > \epsilon} ||\boldsymbol{G}_T^{\lambda_T}(\boldsymbol{\theta})||_2 \leq \frac{1}{2}\delta_\epsilon\right) = 0. \tag{26}$$

We also have, for any $\epsilon' > 0$,
$$\lim_{T\to\infty} \mathbb{P}\left(||\boldsymbol{G}_T^{\lambda_T}(\widehat{\boldsymbol{\theta}}_T)||_2 > \epsilon'/\sqrt{T}\right) = 0.$$

Therefore
$$\lim_{T\to\infty} \left(||\boldsymbol{G}_T^{\lambda_T}(\widehat{\boldsymbol{\theta}}_T)||_2 > \frac{1}{2}\delta_\epsilon\right) = 0. \tag{27}$$

Combining (26) and (27) gives
$$\mathbb{P}(||\widehat{\boldsymbol{\theta}}_T - \boldsymbol{\theta}^*||_2 > \epsilon)$$

$$= \mathbb{P}\left(||\boldsymbol{\theta}_T - \boldsymbol{\theta}^*||_2 > \epsilon, ||\boldsymbol{G}_T^{\lambda_T}(\widehat{\boldsymbol{\theta}}_T)||_2 \leq \frac{1}{2}\delta_\epsilon\right) + \mathbb{P}\left(||\widehat{\boldsymbol{\theta}}_T - \boldsymbol{\theta}^*||_2 > \epsilon, ||\boldsymbol{G}_T^{\lambda_T}(\widehat{\boldsymbol{\theta}}_T)||_2 > \frac{1}{2}\delta_\epsilon\right)$$

$$\leq \mathbb{P}\left(\inf_{\boldsymbol{\theta} \in \boldsymbol{\Theta}, ||\boldsymbol{\theta}-\boldsymbol{\theta}^*||_2 > \epsilon} ||\boldsymbol{G}_T^{\lambda_T}(\boldsymbol{\theta})||_2 \leq \frac{1}{2}\delta_\epsilon\right) + \mathbb{P}\left(||\boldsymbol{G}_T^{\lambda_T}(\widehat{\boldsymbol{\theta}}_T)||_2 > \frac{1}{2}\delta_\epsilon\right)$$

$$\to 0$$

as $T \to \infty$. This establishes consistency of $\widehat{\boldsymbol{\theta}}_T$. $\qquad \square$

**Lemma B.2.** *Under the assumptions of Theorem* 3.7, *as* $T \to \infty, \nabla \boldsymbol{G}_T^{\lambda_T}(\boldsymbol{\theta}^*) \xrightarrow{p} \mathbb{E}\nabla \boldsymbol{g}(\boldsymbol{X}_t, Y_t(a); \boldsymbol{\theta}^*)$.

*Proof.* It suffices to show that $\boldsymbol{c}^\mathsf{T} \nabla \boldsymbol{G}_T^{\lambda_T}(\boldsymbol{\theta}^*)\boldsymbol{c}' \xrightarrow{p} \boldsymbol{c}^\mathsf{T}\mathbb{E}\nabla \boldsymbol{g}(\boldsymbol{X}_t, Y_t(a); \boldsymbol{\theta}^*)\boldsymbol{c}'$ for any nonrandom vectors $\boldsymbol{c}, \boldsymbol{c}' \in \mathbb{R}^d$. To that end, fix $\boldsymbol{c}, \boldsymbol{c}' \in \mathbb{R}^d$ and note

$$\boldsymbol{c}^\mathsf{T} \nabla \boldsymbol{G}_T^{\lambda_T}(\boldsymbol{\theta}^*)\boldsymbol{c}' = A_T + \lambda_T B_T - \lambda_T C_T, \tag{28}$$

where

$$A_T = \frac{1}{T}\sum_{t=1}^T \frac{\mathbb{1}_{\{A_t=a\}}}{\pi_t(a)}\boldsymbol{c}^\mathsf{T}\nabla \boldsymbol{g}(\boldsymbol{X}_t, Y_t; \boldsymbol{\theta}^*)\boldsymbol{c}'$$

$$B_T = \frac{1}{N}\sum_{i=1}^N \boldsymbol{c}^\mathsf{T}\nabla \boldsymbol{g}(\tilde{\boldsymbol{X}}_i, f(\tilde{\boldsymbol{X}}_i); \boldsymbol{\theta}^*)\boldsymbol{c}'$$

$$C_T = \frac{1}{T}\sum_{t=1}^T \frac{\mathbb{1}_{\{A_t=a\}}}{\pi_t(a)}\boldsymbol{c}^\mathsf{T}\nabla \boldsymbol{g}(\boldsymbol{X}_t, f(\boldsymbol{X}_t); \boldsymbol{\theta}^*)\boldsymbol{c}'.$$

We first show that $A_T \xrightarrow{p} \boldsymbol{c}^\mathsf{T}\mathbb{E}\nabla \boldsymbol{g}(\boldsymbol{X}_t, Y_t(a); \boldsymbol{\theta}^*)\boldsymbol{c}'$. Write $Z_t = \frac{\mathbb{1}_{\{A_t=a\}}}{\pi_t(a)}\boldsymbol{c}^\mathsf{T}\nabla \boldsymbol{g}(\boldsymbol{X}_t, Y_t; \boldsymbol{\theta}^*)\boldsymbol{c}'$ so that $A_T = \sum_{t=1}^T Z_t$. Let $Z = \frac{1}{\pi_{\min}}\boldsymbol{c}^\mathsf{T}\nabla \boldsymbol{g}(\boldsymbol{X}_t, Y_t; \boldsymbol{\theta}^*)\boldsymbol{c}'$, and note for all $x$, $\mathbb{P}(|Z_t| \geq x) \leq \mathbb{P}(|Z| \geq x)$ and $\mathbb{E}|Z| < \infty$. To see integrability of $A$, note by our assumptions,

$$||\nabla \boldsymbol{g}(\boldsymbol{X}_t, Y_t(a); \boldsymbol{\theta}^*)||_2 \leq \sup_{||\boldsymbol{\theta}|| \leq R_\theta} ||\nabla \boldsymbol{g}(\boldsymbol{X}_t, Y_t(a); \boldsymbol{\theta})||_2 \leq \phi(\boldsymbol{X}_t, Y_t(a)). \tag{29}$$

Our assumptions also imply that $\mathbb{E}[\phi(\boldsymbol{X}_t, Y_t(a))] < \infty$, so taking the expectation of (29) gives $\mathbb{E}||\nabla \boldsymbol{g}(\boldsymbol{X}_t, Y_t(a); \boldsymbol{\theta}^*)||_2 < \infty$. Thus by Cauchy-Schwarz,

$$\mathbb{E}|A| \leq \frac{1}{\pi_{\min}}||c||_2||c'||_2 E||\nabla \boldsymbol{g}(\boldsymbol{X}_t, Y_t(a); \boldsymbol{\theta}^*)||_2 < \infty.$$

We can now apply Theorem 2.19 from Hall & Heyde (1980) to get

$$\frac{1}{T}\sum_{t=1}^T (Z_t - \mathbb{E}[Z_t|\mathcal{H}_{t-1}]) \xrightarrow{p} 0.$$

Since $\mathbb{E}[Z_t|\mathcal{H}_{t-1}] = \boldsymbol{c}^\mathsf{T}\nabla \boldsymbol{g}(\boldsymbol{X}_t, Y_t(a); \boldsymbol{\theta}^*)\boldsymbol{c}'$, it follows that

$$A_T \xrightarrow{p} \boldsymbol{c}^\mathsf{T}\mathbb{E}\nabla \boldsymbol{g}(\boldsymbol{X}, Y(a); \boldsymbol{\theta}^*)\boldsymbol{c}'.$$

A similar argument shows that $C_T \xrightarrow{p} \boldsymbol{c}^\mathsf{T}\mathbb{E}\nabla \boldsymbol{g}(\boldsymbol{X}, f(\boldsymbol{X}); \boldsymbol{\theta}^*)\boldsymbol{c}'$. And by the Law of Large Numbers, $B_T \xrightarrow{p} \boldsymbol{c}^\mathsf{T}\mathbb{E}\nabla \boldsymbol{g}(\boldsymbol{X}, f(\boldsymbol{X}); \boldsymbol{\theta}^*)\boldsymbol{c}'$ as well. Taking the limit of (28), the result follows.

$\square$

**Lemma B.3.** *Under the assumptions of Theorem* 3.7, $\sup_{||\boldsymbol{\theta}-\boldsymbol{\theta}^*||_2 \leq \epsilon_0} ||\nabla^2 \boldsymbol{G}_T^{\lambda_T}(\boldsymbol{\theta})||_1 = \mathcal{O}_p(1)$, *where the tensor norm is defined as* $||\boldsymbol{B}||_1 = \sum_{i\in[d_1], j\in[d_2], k\in[d_3]} |\boldsymbol{B}_{i,j,k}|$ *for* $\boldsymbol{B} \in \mathbb{R}^{d_1 \times d_2 \times d_3}$.

*Proof.* Note that

$$\nabla^2 \boldsymbol{G}_T^{\lambda_T}(\boldsymbol{\theta}) = \frac{1}{T}\sum_{t=1}^T \frac{\mathbb{1}_{\{A_t=a\}}}{\pi_t(a)}\nabla^2\boldsymbol{g}(\boldsymbol{X}_t, Y_t; \boldsymbol{\theta}) + \frac{1}{N}\sum_{i=1}^N \lambda_T\nabla^2\boldsymbol{g}(\tilde{\boldsymbol{X}}_i, f(\tilde{\boldsymbol{X}}_i), \boldsymbol{\theta}) - \frac{1}{T}\sum_{t=1}^T \frac{\mathbb{1}_{\{A_t=a\}}}{\pi_t(a)}\lambda_T\nabla^2\boldsymbol{g}(\boldsymbol{X}_t, f(\boldsymbol{X}_t), \boldsymbol{\theta})).$$

By triangle inequality,

$$||\nabla^2 \boldsymbol{G}_T^\lambda(\boldsymbol{\theta})||_1 \leq \frac{1}{\pi_{\min}}A_T + \lambda_T B_T + \frac{1}{\pi_{\min}}\lambda_T C_T$$

where

$$A_T(\boldsymbol{\theta}) = \frac{1}{T}\sum_{t=1}^{T}||\nabla^2 \boldsymbol{g}(\boldsymbol{X}_t, Y_t(a); \boldsymbol{\theta})||_1$$

$$B_T(\boldsymbol{\theta}) = \frac{1}{N}\sum_{i=1}^{N}||\nabla^2 \boldsymbol{g}(\tilde{\boldsymbol{X}}_i, f(\tilde{\boldsymbol{X}}_i); \boldsymbol{\theta})||_1$$

$$C_T(\boldsymbol{\theta}) = \frac{1}{T}\sum_{t=1}^{T}||\nabla^2 \boldsymbol{g}(\boldsymbol{X}_t, f(\boldsymbol{X}_t); \boldsymbol{\theta})||_1.$$

Note $\lambda_T$ is tight, so to establish tightness of $\sup_{||\boldsymbol{\theta}-\boldsymbol{\theta}^*||_2\leq\epsilon_0}||\nabla^2 \boldsymbol{G}_T^\lambda(\boldsymbol{\theta})||_1$, it suffices to show tightness of $\sup_{||\boldsymbol{\theta}-\boldsymbol{\theta}^*||_2\leq\epsilon_0} A_T(\theta)$, $\sup_{||\boldsymbol{\theta}-\boldsymbol{\theta}^*||_2\leq\epsilon_0} B_T(\theta)$, and $\sup_{||\boldsymbol{\theta}-\boldsymbol{\theta}^*||_2\leq\epsilon_0} C_T(\theta)$. Assumption 3.6 implies that for all $\boldsymbol{\theta} \in \overline{\mathcal{B}(\boldsymbol{\theta}^*, \epsilon_0)}$,

$$||A_T(\boldsymbol{\theta})||_1 \leq \frac{1}{T}\sum_{t=1}^{T}d^2\sup_{i\in[d]}||\nabla^2 \boldsymbol{g}^{(i)}(\boldsymbol{X}_t, Y_t(a); \boldsymbol{\theta})||_2 \leq \frac{d^2}{T}\sum_{t=1}^{T}\Phi(\boldsymbol{X}_t, Y_t(a)).$$

Thus

$$\sup_{||\boldsymbol{\theta}-\boldsymbol{\theta}^*||_2\leq\epsilon_0}||A_T(\boldsymbol{\theta})||_1 \leq d^2 \cdot \frac{1}{T}\sum_{t=1}^{T}\Phi(\boldsymbol{X}_t, Y_t(a)) = \mathcal{O}_p(1).$$

A similar argument shows

$$\sup_{||\boldsymbol{\theta}-\boldsymbol{\theta}^*||_2\leq\epsilon_0} B_T(\boldsymbol{\theta}) = \mathcal{O}_p(1)$$

$$\sup_{||\boldsymbol{\theta}-\boldsymbol{\theta}^*||_2\leq\epsilon_0} C_T(\boldsymbol{\theta}) = \mathcal{O}_p(1),$$

as desired.

$\square$

**Lemma B.4.** *Under the assumptions of Theorem 3.7, $\sqrt{T}\boldsymbol{G}_T^{\lambda_T}(\boldsymbol{\theta}^*) \xrightarrow{d} \mathcal{N}(\boldsymbol{0}, \overline{\lambda}^2 r\mathrm{Cov}(\boldsymbol{g}(\boldsymbol{X}, f(\boldsymbol{X}); \boldsymbol{\theta}^*)) + \boldsymbol{V})$ as $T \to \infty$.*

*Proof.* Note that $\sqrt{T}\boldsymbol{G}_T^{\lambda_T}(\boldsymbol{\theta}^*) - \sqrt{T}\boldsymbol{G}_T^{\overline{\lambda}}(\boldsymbol{\theta}^*) = o_p(1)$, so it suffices to show that

$$\sqrt{T}\boldsymbol{G}_T^{\overline{\lambda}}(\boldsymbol{\theta}^*) \xrightarrow{d} \mathcal{N}(\boldsymbol{0}, \overline{\lambda}^2 r\mathrm{Cov}(\boldsymbol{g}(\boldsymbol{X}, f(\boldsymbol{X}); \boldsymbol{\theta}^*)) + \boldsymbol{V}). \tag{30}$$

To that end, note

$$\sqrt{T}\boldsymbol{G}_T^{\overline{\lambda}}(\boldsymbol{\theta}^*) = \overline{\lambda}\sqrt{\frac{T}{N}}\boldsymbol{A}_N + \boldsymbol{B}_T, \tag{31}$$

where

$$\boldsymbol{A}_N = \frac{1}{\sqrt{N}}\sum_{i=1}^{N}\left(\boldsymbol{g}(\tilde{\boldsymbol{X}}_i, f(\tilde{\boldsymbol{X}}_i); \boldsymbol{\theta}^*) - \mathbb{E}[\boldsymbol{g}(\boldsymbol{X}, f(\boldsymbol{X}); \boldsymbol{\theta}^*)]\right)$$

$$\boldsymbol{B}_T = \frac{1}{\sqrt{T}}\sum_{t=1}^{T}\left(\frac{\mathbb{1}_{\{A_t=a\}}}{\pi_t(a)}\big[\boldsymbol{g}(\boldsymbol{X}_t, Y_t(a); \boldsymbol{\theta}^*) - \overline{\lambda}\boldsymbol{g}(\boldsymbol{X}_t, f(\boldsymbol{X}_t); \boldsymbol{\theta}^*)\big] + \overline{\lambda}\mathbb{E}[\boldsymbol{g}(\boldsymbol{X}, f(\boldsymbol{X}); \boldsymbol{\theta}^*)]\right).$$

Since the summands of $\boldsymbol{A}_N$ are i.i.d., we can apply the multivariate CLT to get $\boldsymbol{A}_N \xrightarrow{d} \mathcal{N}(\boldsymbol{0}, \mathrm{Cov}(\boldsymbol{g}(\boldsymbol{X}, f(\boldsymbol{X}); \boldsymbol{\theta}^*))$. We now show that $\boldsymbol{B}_T \xrightarrow{d} \mathcal{N}(\boldsymbol{0}, \boldsymbol{V})$, where

$$\boldsymbol{V} = \mathbb{E}\left[\frac{1}{\pi(a)}\left(g(\boldsymbol{X}_t, Y_t(a); \boldsymbol{\theta}^*) - \overline{\lambda}g(\boldsymbol{X}_t, f(\boldsymbol{X}_t); \boldsymbol{\theta}^*)\right)\left(g(\boldsymbol{X}_t, Y_t(a); \boldsymbol{\theta}^*) - \overline{\lambda}g(\boldsymbol{X}_t, f(\boldsymbol{X}_t); \boldsymbol{\theta}^*)\right)^\top\right]$$

$$- \overline{\lambda}^2 \mathbb{E}[g(\boldsymbol{X}, f(\boldsymbol{X}); \boldsymbol{\theta}^*)]E[g(\boldsymbol{X}, f(\boldsymbol{X}); \boldsymbol{\theta}^*)]^\top.$$

By Cramer-Wold, it suffices to show that for any $c \in \mathbb{R}^d$, $c^\mathsf{T} B_T \xrightarrow{d} \mathcal{N}(0, c^\mathsf{T} V c)$. We now verify the conditions of Theorem 3.2 from Hall & Heyde (1980). Firstly, the summands form a martingale difference sequence:

$$\mathbb{E}\left[ c^\mathsf{T} \left( \frac{1_{\{A_t=a\}}}{\pi_t(a)} \left[ g(X_t, Y_t(a); \theta^*) - \bar{\lambda} g(X_t, f(X_t); \theta^*) \right] + \bar{\lambda}\mathbb{E}[g(X, f(X); \theta^*)] | \mathcal{H}_{t-1} \right] \right)$$

$$= c^\mathsf{T} \left( \mathbb{E}\left[ \frac{1_{\{A_t=a\}}}{\pi_t(a)} \left[ g(X_t, Y_t(a); \theta^*) - \bar{\lambda} g(X_t, f(X_t); \theta^*) \right] | \mathcal{H}_{t-1} \right] + \bar{\lambda}\mathbb{E}[g(X, f(X); \theta^*)] \right)$$

$$= c^\mathsf{T} \left( \mathbb{E}\left[ \mathbb{E}\left[ \frac{1_{\{A_t=a\}}}{\pi_t(a)} \left[ g(X_t, Y_t(a); \theta^*) - \bar{\lambda} g(X_t, f(X_t); \theta^*) \right] | \mathcal{H}_{t-1}, X_t \right] | \mathcal{H}_{t-1} \right] + \bar{\lambda}\mathbb{E}[g(X, f(X); \theta^*)] \right)$$

$$= c^\mathsf{T} \left( \mathbb{E}\left[ \frac{1}{\pi_t(a)} \mathbb{E}[1_{\{A_t=a\}} | \mathcal{H}_{t-1}, X_t] \left[ g(X_t, Y_t(a); \theta^*) - \bar{\lambda} g(X_t, f(X_t); \theta^*) \right] | \mathcal{H}_{t-1} \right] + \bar{\lambda}\mathbb{E}[g(X, f(X); \theta^*)] \right)$$

$$= c^\mathsf{T} \left( \mathbb{E}\left[ g(X_t, Y_t(a); \theta^*) - \bar{\lambda} g(X_t, f(X_t); \theta^*) | \mathcal{H}_{t-1} \right] + \bar{\lambda}\mathbb{E}[g(X, f(X); \theta^*)] \right)$$

$$= c^\mathsf{T} \left( \mathbb{E}[g(X_t, Y_t(a); \theta^*)] - \bar{\lambda}\mathbb{E}[g(X, f(X); \theta^*)] + \bar{\lambda}\mathbb{E}[g(X, f(X); \theta^*)] \right)$$

$$= c^\mathsf{T}\mathbb{E}[g(X_t, Y_t(a); \theta^*)]$$

$$= 0.$$

The second equality uses the tower property, the third uses that $g(X_t, Y_t(a); \theta^*) - \bar{\lambda} g(X_t, f(X_t); \theta^*)$ is measurable with respect to $\sigma(X_t)$, the fourth uses that $\pi_t(a) := \mathbb{E}[1_{\{A_t=a\}} | \mathcal{H}_{t-1}, X_t]$, and the last uses that $X_t$ is independent of the history $\mathcal{H}_{t-1}$.

Secondly, we must show

$$\frac{1}{T} \sum_{t=1}^{T} \mathrm{Var}\left( c^\mathsf{T}\left[ \frac{1_{\{A_t=a\}}}{\pi_t(a)} \left[ g(X_t, Y_t(a); \theta^*) - \bar{\lambda} g(X_t, f(X_t); \theta^*) \right] + \bar{\lambda}\mathbb{E}[g(X, f(X); \theta^*)] \right] | \mathcal{H}_{t-1} \right) \xrightarrow{p} c^\mathsf{T} V c.$$

Since the expression inside the variance has mean zero, we can replace the variance with the expectation of the square. Expanding the square gives

$$\frac{1}{T} \sum_{t=1}^{T} \left( \mathbb{E}\left[ \frac{1_{\{A_t=a\}}}{\pi_t(a)^2} (c^\mathsf{T}(g(X_t, Y_t; \theta^*) - \bar{\lambda} g(X_t, f(X_t); \theta^*)))^2 | \mathcal{H}_{t-1} \right] \right.$$

$$\left. + 2c^\mathsf{T}\mathbb{E}[g(X, f(X); \theta^*)] c^\mathsf{T} \mathbb{E}\left[ \frac{1_{\{A_t=a\}}}{\pi_t(a)} (g(X_t, Y_t; \theta^*) - \bar{\lambda} g(X_t, f(X_t); \theta^*) | \mathcal{H}_{t-1} \right] + (\bar{\lambda} c^\mathsf{T}\mathbb{E}[g(X, f(X); \theta^*)])^2 \right)$$

$$= \frac{1}{T} \sum_{t=1}^{T} \left( \mathbb{E}\left[ \frac{1_{\{A_t=a\}}}{\pi_t(a)^2} (c^\mathsf{T}(g(X_t, Y_t; \theta^*) - \bar{\lambda} g(X_t, f(X_t); \theta^*)))^2 | \mathcal{H}_{t-1} \right] - \bar{\lambda}^2 c^\mathsf{T}\mathbb{E}[g(X, f(X); \theta^*)]\mathbb{E}[g(X, f(X); \theta^*)]^\top c \right)$$

$$= \frac{1}{T} \sum_{t=1}^{T} c^T \frac{1}{\pi_t(a)} \mathbb{E}_{Y_t(a)}[(g(X_t, Y_t(a); \theta^*) - \bar{\lambda} g(X_t, f(X_t); \theta^*))^{\otimes 2}] c - \bar{\lambda}^2 c^T \mathbb{E}[g(X, f(X); \theta^*)^{\otimes 2}] c \quad (32)$$

We now study the limit of this. Let $I_t = \frac{1}{\pi_t(a)} \mathbb{E}_{Y_t(a)}[(g(X_t, Y_t(a); \theta^*) - \bar{\lambda} g(X_t, f(X_t); \theta^*))^{\otimes 2}]$ and $\bar{I}_t = \frac{1}{\bar{\pi}(a|X_t)} \mathbb{E}_{Y_t(a)}[(g(X_t, Y_t(a); \theta^*) - \bar{\lambda} g(X_t, f(X_t); \theta^*))^{\otimes 2}]$, where $v^{\otimes 2} := vv^T$ for $v \in \mathbb{R}^d$. Observe

$$|c^T I_t c - c^T \bar{I}_t c|$$

$$= |\bar{\pi}(a|X_t) - \pi_t(a)| \frac{1}{\bar{\pi}(a|X_t)\pi_t(a)} c^T \mathbb{E}[(g(X_t, Y_t(a); \theta^*) - \bar{\lambda} g(X_t, f(X_t); \theta^*))^{\otimes 2} | X_t] c$$

$$\leq M |\bar{\pi}(a|X_t) - \pi_t(a)|$$

for some $M$, since we have bounded moments by Assumption 3.5 and minimum sampling probabilities by Assumption 3.2. Note the random variables $\{\bar{\pi}(a|\boldsymbol{X}_t) - \pi_t(a)\}$ are uniformly integrable. Thus $\bar{\pi}(a|\boldsymbol{X}_t) - \pi_t(a) \xrightarrow{P} 0$ can be strengthened to convergence in $L^1$:

$$\lim_{t \to \infty} \mathbb{E}|\bar{\pi}(a|\boldsymbol{X}_t) - \pi_t(a)| = 0.$$

Now define $U_{t,c} = \boldsymbol{c}^T \boldsymbol{I}_t \boldsymbol{c} - \boldsymbol{c}^T \bar{\boldsymbol{I}}_t \boldsymbol{c}$. Combining the above facts gives

$$\lim_{t \to \infty} \mathbb{E}|U_{t,c}| = 0.$$

Also, note that

$$\mathbb{E}|\mathbb{E}|U_{t,c}|\mathcal{H}_{t-1}| \leq \mathbb{E}\mathbb{E}[|U_{t,c}|\mathcal{H}_{t-1}] = \mathbb{E}|U_{t,c}|.$$

Thus

$$\lim_{t \to \infty} \mathbb{E}|\mathbb{E}[U_{t,c}|\mathcal{H}_{t-1}] = 0.$$

Note that if a sequence of random variables converges to 0 in $L_1$, then its running average does as well. Therefore

$$\frac{1}{t} \sum_{\tau \leq t} \mathbb{E}[U_{\tau,\boldsymbol{c}}|\mathcal{H}_{\tau-1}] \xrightarrow{L_1} 0.$$

Plugging in the definition of $U_{\tau,\boldsymbol{c}}$ gives

$$\frac{1}{T} \sum_{t \leq T} \mathbb{E}[\boldsymbol{c}^T \boldsymbol{I}_t \boldsymbol{c}|\mathcal{H}_{t-1}] - \frac{1}{T} \sum_{t \leq T} \mathbb{E}[\boldsymbol{c}^T \bar{\boldsymbol{I}}_t \boldsymbol{c}|\mathcal{H}_{t-1}] \xrightarrow{L_1} 0.$$

Note also

$$\frac{1}{T} \sum_{t \leq T} \mathbb{E}[\boldsymbol{c}^T \bar{\boldsymbol{I}}_t \boldsymbol{c}|\mathcal{H}_{t-1}] = \frac{1}{T} \sum_{t \leq T} \boldsymbol{c}^T \mathbb{E}[\bar{\boldsymbol{I}}_t] \boldsymbol{c} = \boldsymbol{c}^T \mathbb{E}[\bar{\boldsymbol{I}}_t] \boldsymbol{c} = \boldsymbol{c}^T \mathbb{E}\left[\frac{1}{\bar{\pi}(a|\boldsymbol{X}_t)}(\boldsymbol{g}(\boldsymbol{X}_t, Y_t(a); \boldsymbol{\theta}^*) - \bar{\lambda}\boldsymbol{g}(\boldsymbol{X}_t, f(\boldsymbol{X}_t); \boldsymbol{\theta}^*))^{\otimes 2}\right] \boldsymbol{c}.$$

These imply that the following convergence holds in $L_1$ and probability:

$$\frac{1}{T} \sum_{t \leq T} \mathbb{E}[\boldsymbol{c}^T \boldsymbol{I}_t \boldsymbol{c}|\mathcal{H}_{t-1}] \to \boldsymbol{c}^T \mathbb{E}\left[\frac{1}{\bar{\pi}(a|\boldsymbol{X}_t)}(\boldsymbol{g}(\boldsymbol{X}_t, Y_t(a); \boldsymbol{\theta}^*) - \bar{\lambda}\boldsymbol{g}(\boldsymbol{X}_t, f(\boldsymbol{X}_t); \boldsymbol{\theta}^*))^{\otimes 2}\right] \boldsymbol{c}.$$

Therefore the limit in probability of (32) is

$$\boldsymbol{c}^T \left(\mathbb{E}\left[\frac{1}{\bar{\pi}(a)}\left(\boldsymbol{g}(\boldsymbol{X}_t, Y_t(a); \boldsymbol{\theta}^*) - \bar{\lambda}\boldsymbol{g}(\boldsymbol{X}_t, f(\boldsymbol{X}_t); \boldsymbol{\theta}^*)\right)^{\otimes 2}\right] - \bar{\lambda}^2 \mathbb{E}[\boldsymbol{g}(\boldsymbol{X}, f(\boldsymbol{X}); \boldsymbol{\theta}^*)]^{\otimes 2}\right) \boldsymbol{c}$$

$$= \boldsymbol{c}^\mathsf{T} \boldsymbol{V} \boldsymbol{c},$$

as desired.

Lastly, we must verify the conditional Lindeberg condition:

$$\frac{1}{T} \sum_{t=1}^{T} \mathbb{E}\left[\left(\boldsymbol{c}^\mathsf{T} Z_t\right)^2 \mathbb{1}_{|\boldsymbol{c}^\mathsf{T} Z_t| > \sqrt{T}\delta}|\mathcal{H}_{t-1}\right] \xrightarrow{P} 0 \quad \forall \delta > 0 \tag{33}$$

where $Z_t = \frac{\mathbb{1}_{A_t=a}}{\pi_t(a)}\left[\boldsymbol{g}(\boldsymbol{X}_t, Y_t(a); \boldsymbol{\theta}^*) - \bar{\lambda}\boldsymbol{g}(\boldsymbol{X}_t, f(\boldsymbol{X}_t); \boldsymbol{\theta}^*)\right] + \bar{\lambda}\mathbb{E}[\boldsymbol{g}(\boldsymbol{X}, f(\boldsymbol{X}); \boldsymbol{\theta}^*)]$. To see this, observe:

$$\frac{1}{T} \sum_{t=1}^{T} \mathbb{E}\left[\left(\boldsymbol{c}^\mathsf{T} Z_t\right)^2 \mathbb{1}_{|\boldsymbol{c}^\mathsf{T} Z_t| > \sqrt{T}\delta}|\mathcal{H}_{t-1}\right]$$

$$\leq \frac{1}{T} \cdot \frac{1}{T\delta^2} \sum_{t=1}^{T} \mathbb{E}[(\boldsymbol{c}^\mathsf{T} Z_t)^4 |\mathcal{H}_{t-1}]$$

$$\leq \frac{1}{T^2\delta^2} \sum_{t=1}^{T} \mathbb{E}\left[c^{\mathsf{T}}(\frac{1}{\pi_{\min}}[g(X_t, Y_t(a); \theta^*) - \bar{\lambda}g(X_t, f(X_t); \theta^*)] + \bar{\lambda}\mathbb{E}[g(X, f(X); \theta^*)])^4\right]$$

$$= \frac{1}{T^2\delta^2} \sum_{t=1}^{T} \mathbb{E}\left[\sum_{k=0}^{4} \frac{1}{\pi_{\min}^k}\binom{4}{k}\left(c^{\mathsf{T}}g(X_t, Y_t(a); \theta^*) - \bar{\lambda}c^{\mathsf{T}}g(X_t, f(X_t); \theta^*)\right)^k (c^{\mathsf{T}}\bar{\lambda}\mathbb{E}[g(X, f(X); \theta^*)])^{4-k}\right]$$

$$= \frac{1}{T^2\delta^2} \sum_{t=1}^{T} \mathbb{E}\left[\sum_{k=0}^{4} \frac{1}{\pi_{\min}^k}\binom{4}{k}\left(\sum_{j=0}^{k}\binom{k}{j}(c^{\mathsf{T}}g(X_t, Y_t(a); \theta^*))^j(-\bar{\lambda}c^{\mathsf{T}}g(X_t, f(X_t); \theta^*)^{k-j})(c^{\mathsf{T}}\bar{\lambda}\mathbb{E}[g(X, f(X); \theta^*)])^{4-k}\right)\right]$$

$$= \sum_{k=0}^{4}\sum_{j=0}^{k} \frac{(-1)^{k-j}\bar{\lambda}^{4-j}\binom{4}{k}\binom{k}{j}}{T^2\delta^2\pi_{\min}^4} \sum_{t=1}^{T} \mathbb{E}\left[(c^{\mathsf{T}}g(X_t, Y_t(a); \theta^*))^j(c^{\mathsf{T}}g(X_t, f(X_t); \theta^*))^{k-j}\right](\mathbb{E}[c^{\mathsf{T}}g(X, f(X); \theta^*)])^{4-k}$$

$$\leq \sum_{k=0}^{4}\sum_{j=0}^{k} \frac{(-1)^{k-j}\bar{\lambda}^{4-j}\binom{4}{k}\binom{k}{j}}{T^2\delta^2\pi_{\min}^4} \sum_{t=1}^{T} \left(\mathbb{E}\left[(c^{\mathsf{T}}g(X_t, Y_t(a); \theta^*)^{j\cdot\frac{k}{j}}\right]\right)^{\frac{j}{k}}\left(\mathbb{E}\left[(c^{\mathsf{T}}g(X_t, f(X_t); \theta^*)^{(k-j)\cdot\frac{k}{k-j}}\right]\right)^{\frac{k-j}{k}}(\mathbb{E}[c^{\mathsf{T}}g(X, f(X); \theta^*)])^{4-k}$$

$$= \sum_{k=0}^{4}\sum_{j=0}^{k} \frac{(-1)^{k-j}\bar{\lambda}^{4-j}\binom{4}{k}\binom{k}{j}}{T^2\delta^2\pi_{\min}^4} \sum_{t=1}^{T} \left(\mathbb{E}\left[(c^{\mathsf{T}}g(X_t, Y_t(a); \theta^*)^k\right]\right)^{\frac{j}{k}}\left(\mathbb{E}\left[(c^{\mathsf{T}}g(X_t, f(X_t); \theta^*)^k\right]\right)^{\frac{k-j}{k}}(\mathbb{E}[c^{\mathsf{T}}g(X, f(X); \theta^*)])^{4-k}$$

$$= \frac{1}{T}\sum_{k=0}^{4}\sum_{j=0}^{k} \frac{(-1)^{k-j}\bar{\lambda}^{4-j}\binom{4}{k}\binom{k}{j}}{\delta^2\pi_{\min}^4} \left(\mathbb{E}\left[(c^{\mathsf{T}}g(X_t, Y_t(a); \theta^*))^k\right]\right)^{\frac{j}{k}}\left(\mathbb{E}\left[(c^{\mathsf{T}}g(X_t, f(X_t); \theta^*))^k\right]\right)^{\frac{k-j}{k}}(c^{\mathsf{T}}\mathbb{E}[g(X, f(X); \theta^*)])^{4-k}$$

$$\xrightarrow{p} 0.$$

The first inequality follows from Chebyshev, the first two equalities follow from the Binomial theorem applied twice, the third equality follows from interchanging sums, and the second inequality uses Holder's inequality with $p = \frac{k}{j}$ and $q = \frac{k}{k-j}$. The convergence holds because the summands are finite: $\mathbb{E}[g(X, f(X); \theta^*)]$ is finite by assumption, and the first two expectations are finite since $j, k \leq 4$ and

$$\mathbb{E}[(c^{\mathsf{T}}g(X_t, Y_t(a); \theta^*))^4] \leq ||c||_2^4\mathbb{E}[||g(X_t, Y_t(a); \theta^*)||_2^4] < \infty$$

$$\mathbb{E}[(c^{\mathsf{T}}g(X_t, f(X_t); \theta^*))^4] \leq ||c||_2^4\mathbb{E}[||g(X_t, f(X_t); \theta^*)||_2]^4 < \infty$$

by Cauchy-Schwarz and Assumption 3.5. This establishes (33), completing the proof that $B_T \xrightarrow{d} \mathcal{N}(0, V)$.

Finally, note $A_N$ and $B_T$ are independent, so they converge jointly. Thus by continuous mapping theorem and Slutsky's Theorem, taking the limit of (31) yields $\sqrt{T}G_T^{\bar{\lambda}}(\theta^*) \xrightarrow{d} \bar{\lambda}\sqrt{r}Z_1 + Z_2$, where $Z_1 \sim \mathcal{N}(0, \text{Cov}(g(X, f(X); \theta^*))$ and $Z_2 \sim \mathcal{N}(0, V)$ are independent. By standard properties of independent normals, this limiting distribution is $\mathcal{N}(0, \bar{\lambda}^2 r\text{Cov}(g(X, f(X); \theta^*)) + V)$, establishing (30). $\qquad\square$

## C. Consistent Estimate of Asymptotic Variance

**Theorem C.1.** *A consistent estimate of $\Sigma_{\bar{\lambda}}$ from Theorem 3.7 is*

$$\hat{\Sigma} = B_T^{-1}\left(\lambda_T\frac{T}{N}(D_T - E_T^{\otimes 2}) + H_T - \lambda_T^2 E_T^{\otimes 2}\right)B_T^{-1}, \tag{34}$$

*where $\lambda_T, B_T, D_T,$ and $E_T$ are defined as in Theorem 3.8 and $H_T = \frac{1}{T-1}\sum_{t=1}^{T-1}\frac{1_{A_t=a}}{\pi_t(a)^2}(g_{Y,t} - \lambda_T g_{f,t})^{\otimes 2}$.*

*Proof.* By assumption, $\lambda_T \xrightarrow{p} \bar{\lambda}$, and $T/N \to r$. Also, from the proof of Theorem 3.8, we have

$$B_T \xrightarrow{p} \mathbb{E}[\nabla g(X, Y(a); \theta^*)]$$

$$D_T \xrightarrow{p} \mathbb{E}[g_f^* g_f^{*\top}]$$

$$\boldsymbol{E}_T \xrightarrow{p} \mathbb{E}[\boldsymbol{g}_f^*].$$

So by continuous mapping theorem, it suffices to show

$$\boldsymbol{H}_T \xrightarrow{p} \mathbb{E}\left[\frac{1}{\bar{\pi}(a)}(\boldsymbol{g}_Y - \bar{\lambda}\boldsymbol{g}_f)^{\otimes 2}\right] \tag{35}$$

To that end, note

$$\boldsymbol{H}_T = \frac{1}{T-1}\sum_{t=1}^{T-1}\frac{1_{A_t=a}}{\pi_t(a)^2}\boldsymbol{g}_{Y,t}^{\otimes 2} + \lambda_T^2 \boldsymbol{F}_T - \lambda_T \boldsymbol{C}_T, \tag{36}$$

where $\boldsymbol{F}_T$ and $\boldsymbol{C}_T$ are defined in the proof of Theorem 3.8. Again from that proof, we have

$$\boldsymbol{F}_T \xrightarrow{p} \mathbb{E}\left[\frac{1}{\bar{\pi}(a)}\boldsymbol{g}_f^*\boldsymbol{g}_f^{*\mathsf{T}}\right]$$

$$\boldsymbol{C}_T \xrightarrow{p} E\left[\frac{1}{\bar{\pi}(a)}(\boldsymbol{g}_f^*\boldsymbol{g}_Y^{*\mathsf{T}} + \boldsymbol{g}_Y^*\boldsymbol{g}_f^{*\mathsf{T}})\right]$$

and by the same argument,

$$\frac{1}{T-1}\sum_{t=1}^{T-1}\frac{1_{A_t=a}}{\pi_t(a)^2}\boldsymbol{g}_{Y,t}^{\otimes 2} \xrightarrow{p} \mathbb{E}\left[\frac{\boldsymbol{g}_{Y,t}^{\otimes 2}}{\bar{\pi}(a)}\right].$$

Thus by continuous mapping theorem applied to (36),

$$\boldsymbol{H}_T \xrightarrow{p} \mathbb{E}\left[\frac{\boldsymbol{g}_{Y,t}^{\otimes 2}}{\bar{\pi}(a)}\right] + \bar{\lambda}^2\mathbb{E}\left[\frac{1}{\bar{\pi}(a)}\boldsymbol{g}_f^*\boldsymbol{g}_f^{*\mathsf{T}}\right] - \bar{\lambda}E\left[\frac{1}{\bar{\pi}(a)}(\boldsymbol{g}_f^*\boldsymbol{g}_Y^{\mathsf{T}} + \boldsymbol{g}_Y^*\boldsymbol{g}_f^{*\mathsf{T}})\right]$$

$$= \mathbb{E}\left[\frac{1}{\bar{\pi}(a)}(\boldsymbol{g}_Y - \bar{\lambda}\boldsymbol{g}_f)^{\otimes 2}\right].$$

establishing (35) and completing the proof.

$\square$

## D. Further Simulations

### D.1. Low-Exploration Regimes

To show strong predictions can reduce variance due to small inverse propensity weights, we reran the experiment of Section 4 with smaller $\epsilon$ and a stronger predictive model $f^{\text{excellent}}(\boldsymbol{x}) = \exp(0.201\boldsymbol{x}^{(1)} - 0.097\boldsymbol{x}^{(2)} + 0.398\boldsymbol{x}^{(3)})$. Table 4 shows the average confidence interval width of 90% confidence intervals with coverage in parentheses. In this low-exploration regime, PPAI confidence intervals are 10-16 times tighter than those of WLS and maintain near-nominal coverage.

| $\epsilon$ | WLS | PPAI ($f^{\text{excellent}}$) |
|---|---|---|
| 0.01 | 0.47 (89%) | 0.03 (88%) |
| 0.05 | 0.19 (90%) | 0.02 (89%) |

*Table 4.* Confidence interval width and coverage for WLS and PPAI under small choices of $\epsilon$. PPAI intervals are substantially tighter and achieve near-nominal coverage.

### D.2. Ablation Study: Importance of the Correction Term

We conduct an ablation study to evaluate the impact of the PPAI correction term, by comparing standard PPAI against an estimator that relies entirely on the AI predictions and unlabeled data:

$$\hat{\boldsymbol{\theta}}_{a,T}^{\text{NC}} = \left[\sum_{i=1}^N \tilde{\boldsymbol{X}}_i \tilde{\boldsymbol{X}}_i^T\right]^{-1} \times \sum_{i=1}^N \tilde{\boldsymbol{X}}_i f_a(\tilde{\boldsymbol{X}}_i). \tag{37}$$

As shown in Table 5, the inclusion of the correction term significantly reduces the mean absolute error (MAE) across 300 trials. As expected, the error reduction is largest when the AI model is poor ($f^{poor}$), demonstrating that the correction term is vital for maintaining accuracy, especially for poorly calibrated AI models.

| | PPAI w/o correction | PPAI |
|---|---|---|
| $f^{poor}$ | 0.222 | 0.005 |
| $f^{strong}$ | 0.013 | 0.003 |

*Table 5.* Mean absolute error over 300 trials comparing PPAI with and without the correction term.

### D.3. Ablation Study: Adaptive Lambda Selection

To assess the effectiveness of the data-driven parameter $\hat{\lambda}_T$, we compare our procedure against an approach where $\lambda$ is fixed at $\lambda = 0.5$. The results in Table 6 show that using $\hat{\lambda}_T$ drastically reduces the confidence interval width while maintaining or improving coverage. For $f^{poor}$ the interval width narrows from 0.355 to 0.021, illustrating that adaptively selecting $\lambda$ effectively reduces variance.

| | PPAI with $\lambda = 0.5$ | PPAI with $\hat{\lambda}_T$ |
|---|---|---|
| $f^{poor}$ | 0.355 (88%) | 0.021 (89%) |
| $f^{strong}$ | 0.054 (93%) | 0.011 (93%) |

*Table 6.* CI width (coverage) comparing fixed $\lambda$ against adaptive $\hat{\lambda}_T$ selection.

### D.4. Misspecification Analysis

We evaluate the robustness of PPAI under varying levels of reward model misspecification. We define the true reward as $Y = X^\top \beta + c \cdot h(X)$, where $\beta = [0.3, 0.1, 0.2]^\top$ and $h(\boldsymbol{x}) = \exp(0.2\boldsymbol{x}^{(1)} - 0.1\boldsymbol{x}^{(2)} + 0.3\boldsymbol{x}^{(3)})$. We test across $c \in \{0, 0.5, 1\}$, comparing a strong predictive model $f^{strong}(\boldsymbol{x}) = \boldsymbol{x}^\top \beta + \exp(0.29\boldsymbol{x}^{(1)} + 0.11\boldsymbol{x}^{(2)} + 0.19\boldsymbol{x}^{(3)})$ and a poor model $f^{poor}(\boldsymbol{x}) = 1$. The remaining experimental setup from Section 4 is unchanged. As shown in Table 7, PPAI CIs consistently outperform standard WLS as the misspecification constant $c$ increases, maintaining reliable coverage with significantly narrower widths.

| $c$ | WLS | PPAI ($f^{poor}$) | PPAI ($f^{strong}$) |
|---|---|---|---|
| 0.0 | 0.009 (91%) | 0.009 (91%) | 0.009 (90%) |
| 0.5 | 0.054 (91%) | 0.031 (89%) | 0.010 (88%) |
| 1.0 | 0.107 (91%) | 0.063 (89%) | 0.011 (92%) |

*Table 7.* Width and coverage of 90% confidence intervals under increasing reward misspecification.

### D.5. Confidence Interval Implementation Details

To construct the PPAI confidence intervals summarized in Table 1, we use the plug-in estimate of the PPAI asymptotic variance $\Sigma_{\lambda^*}$ described in Appendix C. For this problem,

$$\Sigma_{\lambda^*} = \boldsymbol{J}^{-1} \left( \lambda^{*2} r A + \boldsymbol{B} + \lambda^* C - \lambda^* D \right) \boldsymbol{J}^{-1},$$

where

$$\boldsymbol{J} = \mathbb{E}[\boldsymbol{X}\boldsymbol{X}^\top]^{-1}$$

$$A = \mathrm{Cov}(\boldsymbol{X}(Y(0) - \boldsymbol{X}^\top \boldsymbol{\theta}_0^*))$$

$$B = \mathbb{E}\left[ \frac{1}{\bar{\pi}(0)} (Y(0) - \boldsymbol{X}^\top \boldsymbol{\theta}_0^*)^2 \boldsymbol{X}\boldsymbol{X}^\top \right]$$

$$C = \mathbb{E}\left[ \frac{1}{\bar{\pi}(0)} (f_0(\boldsymbol{X}) - \boldsymbol{X}^\top \boldsymbol{\theta}_0^*)^2 \boldsymbol{X}\boldsymbol{X}^\top \right]$$

$$D = 2\mathbb{E}\left[\frac{1}{\bar{\pi}(0)}(Y(0) - \boldsymbol{X}^\top\boldsymbol{\theta}_0^*)(f_0(\boldsymbol{X}) - \boldsymbol{X}^\top\boldsymbol{\theta}_0^*)\boldsymbol{X}\boldsymbol{X}^\top\right] + \mathbb{E}\left[(f_0(\boldsymbol{X}) - \boldsymbol{X}^\top\boldsymbol{\theta}_0^*)\boldsymbol{X}\right]^{\otimes 2},$$

and our plug-in estimate is

$$\hat{\Sigma} = \hat{\boldsymbol{J}}^{-1}\left(\hat{\lambda}^2\frac{T}{N}\hat{A} + \hat{B} + \hat{\lambda}^2\hat{C} - \hat{\lambda}\hat{D}\right)\hat{\boldsymbol{J}}^{-1}$$

where

$$\hat{\boldsymbol{J}} = \frac{1}{N}\sum_{i=1}^{N}\tilde{\boldsymbol{X}}_i\tilde{\boldsymbol{X}}_i^\top$$

$$\hat{A} = \frac{1}{T}\sum_{t=1}^{T}\frac{1_{A_t=0}}{\pi_t(0)}(\boldsymbol{X}_t(Y_t - \boldsymbol{X}_t^\top\hat{\boldsymbol{\theta}}_{0,t}))^{\otimes 2} - \left(\frac{1}{T}\sum_{t=1}^{T}\frac{1_{A_t=0}}{\pi_t(0)}(\boldsymbol{X}_t(Y_t - \boldsymbol{X}_t^\top\hat{\boldsymbol{\theta}}_{0,t}))\right)^{\otimes 2}$$

$$\hat{B} = \frac{1}{T}\sum_{t=1}^{T}\frac{1_{A_t=0}}{\pi_t(0)^2}(Y_t - \boldsymbol{X}_t^\top\hat{\boldsymbol{\theta}}_{0,t})^2\boldsymbol{X}_t\boldsymbol{X}_t^\top$$

$$\hat{C} = \frac{1}{T}\sum_{t=1}^{T}\frac{1_{A_t=0}}{\pi_t(0)^2}(f_0(\boldsymbol{X}_t) - \boldsymbol{X}_t^\top\hat{\boldsymbol{\theta}}_{0,t})^2\boldsymbol{X}_t\boldsymbol{X}_t^\top$$

$$\hat{D} = 2\left(\frac{1}{T}\sum_{t=1}^{T}\frac{1_{A_t=0}}{\pi_t(0)^2}(Y_t - \boldsymbol{X}_t^\top\hat{\boldsymbol{\theta}}_{0,t})(f_0(\boldsymbol{X}_t) - \boldsymbol{X}_t^\top\hat{\boldsymbol{\theta}}_{0,t})\boldsymbol{X}_t\boldsymbol{X}_t^\top\right) + \left(\frac{1}{T}\sum_{t=1}^{T}\frac{1_{A_t=0}}{\pi_t(0)}(f_0(\boldsymbol{X}_t) - \boldsymbol{X}_t^\top\hat{\boldsymbol{\theta}}_{0,t})\boldsymbol{X}_t\right)^{\otimes 2}$$

We construct $1 - \alpha$ confidence intervals for the first component of $\boldsymbol{\theta}_0^*$ via

$$\hat{\boldsymbol{\theta}}_0^{(0)} \pm z_{1-\alpha/2}\sqrt{\frac{\hat{\Sigma}_{0,0}}{T}}, \tag{38}$$

where $z_q$ denotes the $q$th quantile of the standard normal distribution.

## E. Movie Recommendation Experiment Details

### E.1. Movie Recommendation Reward Model

To construct the reward model, we trained a random forest regressor to predict the rating for a user-movie pair $(u, m)$ based on the following five features:

1. (User Mean) The average rating provided by user $u$.

2. (Movie Mean) The average rating received by movie $m$.

3. (Interaction: User × Movie) The product of the user mean and the movie mean.

4. (Genre Affinity) The user's historical average rating for the specific genres associated with movie $m$.

5. (Interaction: Genre × Movie) The product of the genre affinity and the movie mean.

We then round the output of the random forest to $\{0.5, 1.0, 1.5, 2.0, 2.5, 3.0, 3.5, 4.0, 4.5, 5.0\}$ to make it a valid rating. The model was initially trained on $80\%$ of the data. On the remaining $20\%$ test set, the model achieved a Mean Absolute Error of $0.597$, indicating that simulated ratings typically deviate by approximately half of a star from observed ratings. Following this validation, we retrained the model on the full dataset to serve as our simulation environment.

### E.2. Experiment Steps

We simulate our method as follows: At time step $t = 1, \ldots, 1000$,

1. We sample the context $\boldsymbol{X}_t$ of the $t$th user.

2. We apply an $\epsilon$-greedy policy with $\epsilon = 0.1$ to choose the movie $m(A_t)$ to show the user. With probability $\epsilon$, we select $A_t \in \{0, 1\}$ randomly; with probability $1 - \epsilon$, we select $A_t = \arg\max_a \boldsymbol{\beta}_a^\top \boldsymbol{X}_t$.

3. We query the reward model to compute the rating $Y_t = y(\boldsymbol{X}_t, m(A_t))$.

4. We draw 10 unlabeled user contexts $\tilde{\boldsymbol{X}}_{10(t-1)+1}, \ldots, \tilde{\boldsymbol{X}}_{10t}$ and compute predicted ratings for both movies $m(0), m(1)$ by querying Gemini.

5. With the labeled data $(\boldsymbol{X}_\tau, A_\tau, Y_\tau)_{\tau=1}^t$ and unlabeled data $(\tilde{\boldsymbol{X}}_i, a, f(\tilde{\boldsymbol{X}}_i))_{i=1}^{10t}$ for $a = 0, 1$, we compute the PPAI least squares estimator (7).

To get predicted ratings from Gemini, we use prompts of the following form: "Predict how many stars (1-5) a viewer would give X-Men (2000) given that they gave these ratings: Independence Day (1996): 4.5, The Lord of the Rings: The Return of the King (2003): 3.0, Pulp Fiction (1994): 2.5, Shrek (2001): 5.0. State your predicted rating within brackets, e.g. [4.5]."

### E.3. Hypothesis Tests

We apply hypothesis tests to demonstrate how PPAI can enable discoveries. First, since $\boldsymbol{\beta}_0^{(3)} \approx 0.17$ is well separated from zero, it suggests a non-zero relationship between ratings of *Star Wars* and *X-Men*. We therefore test $H_0 : \boldsymbol{\beta}_0^{(3)} = 0$ against $H_1 : \boldsymbol{\beta}_0^{(3)} \neq 0$. With PPAI, we (correctly) reject in $90.0\%$ of trials, whereas with WLS, we reject in just $19\%$ of trials; this shows our method gives greater power against $H_0$.

To test estimates of $\boldsymbol{\beta}_1^{(1)}$, we first note the proximity of $\boldsymbol{\beta}_1^{(1)} \approx 0.03$ to zero demonstrates a negligible relationship between ratings of *Good Will Hunting* and *Independence Day*. We therefore employ a negligibility test: the null hypothesis is $H_0 : \boldsymbol{\beta}_1^{(1)} \notin [-\delta, \delta]$ ($\boldsymbol{\beta}_1^{(1)}$ is not negligible), for some $\delta > 0$, and the alternative is $H_1 : \boldsymbol{\beta}_1^{(1)} \in [-\delta, \delta]$ ($\boldsymbol{\beta}_1^{(1)}$ is negligible). The procedure is to reject when the $90\%$ confidence interval is fully contained in $[-\delta, \delta]$. The choice of $\delta > 0$ is somewhat arbitrary and depends on domain knowledge, but to demonstrate our method, we set $\delta = 0.2$. For this $\delta$, our method rejects (correctly concludes negligibility) in $82.5\%$ of trials, whereas WLS *never* rejects, as its confidence intervals are consistently too large to fit in $[-\delta, \delta]$.

## F. Extension to Covariate Shift

In practice, the unlabeled covariates may come from a different distribution than the labeled covariates. In this section, we briefly discuss how our framework can be extended to accommodate such covariate shift.

Specifically, suppose

$$\boldsymbol{X}_t \overset{\text{i.i.d.}}{\sim} \mathcal{P}_X^\ell, \qquad \tilde{\boldsymbol{X}}_i \overset{\text{i.i.d.}}{\sim} \mathcal{P}_X^u,$$

where $\mathcal{P}_X^\ell$ and $\mathcal{P}_X^u$ denote the labeled and unlabeled covariate distributions, with corresponding densities $p_\ell$ and $p_u$, respectively. We assume that $\mathcal{P}_X^\ell$ is absolutely continuous with respect to $\mathcal{P}_X^u$, and define the density ratio

$$w(\boldsymbol{x}) := \frac{p_\ell(\boldsymbol{x})}{p_u(\boldsymbol{x})}.$$

We further assume that $w$ is bounded almost surely.

We first consider the case where $w$ is known. The key idea is to reweight the unlabeled data so that expectations under the unlabeled covariate distribution are converted into expectations under the labeled covariate distribution. Indeed,

$$\mathbb{E}_{\mathcal{P}_X^u}\left[w(\tilde{\boldsymbol{X}})\, \boldsymbol{g}(\tilde{\boldsymbol{X}}, f_a(\tilde{\boldsymbol{X}}); \boldsymbol{\theta}^*)\right] = \int \frac{p_\ell(\tilde{\boldsymbol{x}})}{p_u(\tilde{\boldsymbol{x}})} \boldsymbol{g}(\tilde{\boldsymbol{x}}, f_a(\tilde{\boldsymbol{x}}); \boldsymbol{\theta}^*)\, p_u(\tilde{\boldsymbol{x}})\, d\tilde{\boldsymbol{x}},$$

$$= \int \boldsymbol{g}(\tilde{\boldsymbol{x}}, f_a(\tilde{\boldsymbol{x}}); \boldsymbol{\theta}^*) \, p_\ell(\tilde{\boldsymbol{x}}) \, d\tilde{\boldsymbol{x}} = \mathbb{E}_{\mathcal{P}_{\tilde{X}}^\ell} \left[ \boldsymbol{g}(\tilde{\boldsymbol{X}}, f_a(\tilde{\boldsymbol{X}}); \boldsymbol{\theta}^*) \right].$$

Accordingly, when $w$ is known, we modify the estimating equation by reweighting the unlabeled term:

$$\boldsymbol{G}_T^{\lambda_T, w}(\boldsymbol{\theta}) = \frac{1}{T} \sum_{t=1}^{T} \frac{\mathbf{1}\{A_t = a\}}{\pi_t(a)} \, \boldsymbol{g}(\boldsymbol{X}_t, Y_t; \boldsymbol{\theta}) + \lambda_T \left[ \frac{1}{N} \sum_{i=1}^{N} w(\tilde{\boldsymbol{X}}_i) \, \boldsymbol{g}(\tilde{\boldsymbol{X}}_i, f_a(\tilde{\boldsymbol{X}}_i); \boldsymbol{\theta}) - \frac{1}{T} \sum_{t=1}^{T} \frac{\mathbf{1}\{A_t = a\}}{\pi_t(a)} \, \boldsymbol{g}(\boldsymbol{X}_t, f_a(\boldsymbol{X}_t); \boldsymbol{\theta}) \right].$$

In practice, the density ratio $w$ is typically unknown and must be estimated. Suppose $\hat{w}$ is an estimator constructed from an auxiliary sample independent of $\{(\boldsymbol{X}_t, A_t, Y_t)\}_{t=1}^{T}$ and $\{\tilde{\boldsymbol{X}}_i\}_{i=1}^{N}$. One simple way to obtain such an auxiliary sample is to reserve a small fraction of the labeled and unlabeled data for this purpose. When $\hat{w}$ is a reasonable estimator for $w$, say $\|\hat{w} - w\|_\infty = o_p(T^{-1/2})$, then we may replace $w$ by $\hat{w}$ in the estimating equation:

$$\boldsymbol{G}_T^{\lambda_T, \hat{w}}(\boldsymbol{\theta}) = \frac{1}{T} \sum_{t=1}^{T} \frac{\mathbf{1}\{A_t = a\}}{\pi_t(a)} \, \boldsymbol{g}(\boldsymbol{X}_t, Y_t; \boldsymbol{\theta}) + \lambda_T \left[ \frac{1}{N} \sum_{i=1}^{N} \hat{w}(\tilde{\boldsymbol{X}}_i) \, \boldsymbol{g}(\tilde{\boldsymbol{X}}_i, f_a(\tilde{\boldsymbol{X}}_i); \boldsymbol{\theta}) - \frac{1}{T} \sum_{t=1}^{T} \frac{\mathbf{1}\{A_t = a\}}{\pi_t(a)} \, \boldsymbol{g}(\boldsymbol{X}_t, f_a(\boldsymbol{X}_t); \boldsymbol{\theta}) \right].$$

A full theoretical treatment of this covariate-shift extension is left for future work.

