# OpenReview forum: "Prediction-Powered Adaptive Inference with Pretrained AI Models for Contextual Bandits"
_ICML.cc/2026/Conference — ICML 2026 regular_

### Official Review · Reviewer_5HB9 · 2026-03-12

**Soundness:** 2
**Presentation:** 3
**Significance:** 2
**Originality:** 2
**Overall Recommendation:** 3
**Confidence:** 3

**Summary:**

This paper studies statistical inference in contextual bandits with access to unlabeled contexts and pretrained AI reward predictors. The authors propose a Prediction-Powered Adaptive Inference (PPAI) estimator that combines inverse propensity weighting (IPW) with prediction-powered inference (PPI++). A data-driven tuning parameter $\lambda$ adaptively weights AI predictions based on their informativeness. Theoretical results establish asymptotic normality under adaptive data collection and potential model misspecification. Numerical experiments on a two-armed bandit with exponential rewards demonstrate variance reduction compared to a weighted least squares baseline.

**Compliance With Llm Reviewing Policy:**

Affirmed.

**Final Justification:**

After carefully reviewing the authors’ rebuttal and their engaged discussion, I maintain my original score of Weak Reject. The authors have thoughtfully responded to the raised concerns and showed constructive engagement throughout the process. However, I still hold clear reservations about the novelty of the proposed approach. The work does not deliver sufficiently distinct contributions or meaningful advances over existing literature to meet ICML’s acceptance criteria.

**Key Questions For Authors:**

### 1. What new technical challenges arise from incorporating IPW into PPI++?

The paper argues that combining IPW with PPI++ requires new theoretical analysis. Please clarify: what specific new technical challenges emerge from this combination that are not already present in either the adaptive inference literature or the prediction-powered inference literature?

I ask because, on the surface, the IPW is applied independently to each labeled term, and the PPI++ structure is preserved. Please help reviewers understand what makes this combination theoretically non-trivial beyond the straightforward application of existing tools.

### 2. Fair comparison

Can you provide results comparing PPAI against a "naive PPI" baseline that uses unlabeled predictions without the correction term ($\lambda=1$ with only the unlabeled term)? Additionally, please include a fixed-$\lambda$ version of PPAI to demonstrate the benefit of adaptive tuning. This would help establish that the correction mechanism is necessary and that adaptive $\lambda$ provides meaningful improvement.

### 3. Misspecification analysis

Despite "misspecification" being a central claim, the experiments use only one fixed nonlinear reward function. How does the performance of PPAI (and the estimated $\lambda$) vary with the degree of misspecification? Consider reward functions of the form $Y = X^\top \beta + c \cdot h(X)$ where $h$ is nonlinear and $c$ varies from 0 (correct specification) to large values. How does $\lambda$ adapt across this spectrum?

### 4. Real-world validation

Do you plan to validate the method on any real-world datasets or simulators? Even an offline evaluation using logged bandit data from platforms like MovieLens, Yahoo! News, or clinical trial simulators would significantly strengthen the empirical contribution and demonstrate practical utility beyond synthetic settings.

### 5. Sensitivity to assumption violations

How sensitive is the method to violations of Assumption 3.2 (minimum sampling probability)? In practice, propensity scores $\pi_t(a)$ can become very small for suboptimal arms. Does this cause numerical instability or violate the theoretical guarantees? Have you tested scenarios where the positivity assumption is nearly violated?

### 6. Computational considerations

What is the computational complexity of Algorithm 1? How does it scale with dimension $d$, number of arms $K$, and time horizon $T$? This information is important for practitioners considering real-time deployment.

### 7. Theory-experiment gap

The experiments report coverage rates but do not validate key theoretical predictions such as:

- The accuracy of asymptotic variance estimation (e.g., comparing estimated vs. theoretical variance)
- The convergence rate of $\hat{\lambda}$ to $\lambda^*$
- Whether assumptions (e.g., policy convergence, bounded moments) hold in the simulations

Can you provide evidence addressing these gaps?

### 8. Interpretation of $\lambda$

The $\lambda$ values in Figure 3 converge to approximately 0.986 and 0.115. What do these numbers imply about the quality of the AI predictions? Can $\lambda$ be calibrated to an interpretable measure of prediction accuracy (e.g., correlation between $f$ and $Y$, or prediction MSE)? This would help users understand what $\lambda$ values mean in practice.

**Limitations:**

This paper presents a technically competent but conceptually incremental contribution. The PPAI estimator is a mechanical combination of two established techniques (IPW + PPI++), and while the theoretical re-validation in the adaptive setting is non-trivial, the combination lacks genuine synergy; the components operate independently without creating new statistical insights or emergent properties. The empirical evaluation is insufficient: unfair baseline comparison, lack of systematic misspecification analysis, and failure to validate key theoretical predictions. The paper has merits (rigorous theory, safety guarantee, online updates), but the weaknesses outweigh them. Major revisions, particularly a redesigned empirical evaluation with fair baselines and systematic misspecification analysis, would be needed before this work can be meaningfully built upon by others.

**Strengths And Weaknesses:**

## Strengths

1. **Timely and relevant problem**: The question of how to safely leverage AI predictions to improve inference in adaptive experiments is practically important, especially in domains like personalized medicine and online recommendations.
2. **Clean methodological formulation**: The PPAI estimator (Equation 8) provides an intuitive way to integrate IPW with prediction-powered corrections. The $\lambda$ parameter offers a natural mechanism to balance prediction quality.
3. **Rigorous theoretical validation**: The authors provide a complete asymptotic theory, including asymptotic normality (Theorem 3.7), derivation of optimal $\lambda$ (Theorem 3.8), and consistency of estimated $\lambda$ (Corollary 3.9). The use of martingale theory is technically sound.
4. **Safety guarantee**: The asymptotic variance never exceeds the labeled-only baseline (Corollary 3.10), a practically important property for deployment.
5. **Online update capability**: Algorithm 1 enables efficient online computation, making the method feasible for real-time applications.

## Weaknesses

### 1. Lack of genuine methodological synergy: A mechanical "A+B" combination

This is the core weakness. The paper combines two existing techniques (IPW from adaptive inference and PPI++ from offline prediction-powered inference) but they function independently rather than synergistically.

- IPW handles the adaptive data problem: It ensures unbiasedness under adaptive collection by reweighting observations.
- PPI++ handles the prediction utilization problem: It uses unlabeled data and predictions to improve efficiency via a correction term.

In the proposed PPAI estimator, these two components operate in parallel without meaningful interaction. The IPW is applied separately to each term involving labeled data, and the PPI++ structure is preserved intact. There is no emergent property that arises specifically from their combination, no new insight about how predictions might help correct adaptive bias, or how adaptive sampling might inform better use of predictions.

For a combination to be genuinely innovative, it should demonstrate that the whole is greater than the sum of its parts. This paper does not. The estimator could have been designed by anyone familiar with both literatures in a straightforward manner: take the PPI++ estimating equation and replace every average over labeled data with an IPW-weighted average. While the theoretical re-validation of this combined estimator is non-trivial, the conceptual step is predictable and lacks surprise.

A true methodological synergy would require, for example:

- Showing that predictions can help reduce the variance of IPW estimators in ways not achievable by either technique alone.
- Demonstrating that adaptive sampling creates opportunities for more efficient use of predictions (e.g., by focusing unlabeled data on informative regions).
- Revealing that the presence of predictions relaxes assumptions needed for adaptive inference (e.g., allowing smaller $\pi_{\min}$).

The paper offers none of these. The combination is additive, not multiplicative.

### 2. Unfair experimental comparison

The baseline (WLS) does not use unlabeled data or AI predictions. Comparing against a method with strictly less information and claiming superiority is not meaningful. A fair comparison requires:

- A "naive PPI" baseline ($\lambda = 1$ without correction) to demonstrate the necessity of the correction term.
- A fixed-$\lambda$ version of PPAI to demonstrate the benefit of adaptive tuning.

### 3. Inadequate evaluation of misspecification

Despite "misspecification" being a central claimed contribution, the experiments:

- Use only one fixed nonlinear reward function.
- Do not systematically vary the degree of misspecification.
- Do not show how $\lambda$ adapts to different misspecification levels.

### 4. Toy experimental setup

The simulation is limited to two arms, 3D contexts, and 5000 steps. No real-world datasets or higher-dimensional settings are considered, undermining generalizability claims.

### 5. Theory-experiment gap

The experiments do not validate key theoretical predictions:

- Accuracy of asymptotic variance estimates.
- Convergence rate of $\hat{\lambda}$ to $\lambda^*$.
- Sensitivity to assumption violations (e.g., small $\pi_t(a)$).
- Whether assumptions (policy convergence, bounded moments) hold in simulations.

### 6. Missing analysis of $\lambda$ behavior

The reported $\lambda$ values (0.115, 0.986) are not interpreted in terms of prediction quality. No confidence bands or comparisons with theoretical $\lambda^*$ are provided.

---

> ### Author Rebuttal · Authors · 2026-03-30
>
> We thank the reviewer for their constructive feedback. We are encouraged that they find our problem timely, the estimator intuitive, and the theoretical guarantees meaningful.
>
> **(1) Why this is not merely "IPW + PPI++".**
> We agree that the estimator has a simple form, but the technical issue is the *inferential object* being analyzed. Our estimator solves $G\_T^{\\lambda\_T}(\\hat\\theta\_T)=0$, with
>
> $$G\_T^{\\lambda\_T}(\\theta) = \\frac1T\\sum\_{t=1}^T\\frac{\\mathbf 1\\{A\_t=a\\}}{\\pi\_t(a)}g(X\_t,Y\_t;\\theta) + \\lambda\_T\\left[\\frac1N\\sum\_{i=1}^N g(\\tilde X\_i,f(\\tilde X\_i);\\theta) - \\frac1T\\sum\_{t=1}^T\\frac{\\mathbf 1\\{A\_t=a\\}}{\\pi\_t(a)}g(X\_t,f(X\_t);\\theta)\\right].$$
>
> The challenge is to analyze a *mixed* estimating equation with: (a) an adaptive IPW-weighted labeled component, (b) an i.i.d. unlabeled prediction component, and (c) a data-driven tuning parameter $\\hat\\lambda\_T$. This is why Theorem 3.7 requires a martingale CLT tailored to the prediction-powered adaptive setting, rather than a direct application of existing results.
>
> **(2) Interaction between predictions and adaptivity.**
> When $\\lambda\\approx 1$ (strong predictions), the estimating equation is
>
> $$G\_{a,T}^{1}(\\theta) = \\frac1N\\sum\_{i=1}^N g(\\tilde X\_i,f\_a(\\tilde X\_i);\\theta) + \\frac1T\\sum\_{t=1}^T\\frac{\\mathbf 1\\{A\_t=a\\}}{\\pi\_t(a)}\\Big(g(X\_t,Y\_t;\\theta)-g(X\_t,f\_a(X\_t);\\theta)\\Big).$$
>
> Strong predictions make the IPW-weighted object a *small residual* rather than the raw score. This shows that *strong predictors allow valid inference with smaller $\\pi\_{\\min}$* (hence less regret).
>
> Another key interaction is that $\\lambda^\\ast$ depends jointly on prediction quality and the policy $\\pi$, showing how optimal prediction weighting depends on the decision-making.
>
> **(3) Ablation 1.**
> We added an ablation comparing PPAI to a version *without* the correction term (mean absolute error over 300 trials):
>
> | | PPAI w/o correction | PPAI |
> |---|---|---|
> | $f^{\\mathrm{poor}}$ | 0.222 | 0.005 |
> | $f^{\\mathrm{strong}}$ | 0.013 | 0.003 |
>
> **(4) Ablation 2.**
> We also added a fixed-$\\lambda$ ablation to show how $\\hat{\\lambda}\_T$ reduces variance (CI width and coverage):
>
> | | PPAI with $\\lambda=0.5$ | PPAI with $\\hat\\lambda\_T$ |
> |---|---|---|
> | $f^{\\mathrm{poor}}$ | 0.355 (88%) | 0.021 (89%) |
> | $f^{\\mathrm{strong}}$ | 0.054 (93%) | 0.011 (93%) |
>
> **(5) Misspecification Analysis.**
> We experiment with $Y=X^T\\beta + c\\cdot h(X)$, where $\\beta = [0.3,0.1,0.2]$, $h(x) = \\exp(0.2 x^{(1)} -0.1 x^{(2)}+0.3 x^{(3)})$, and $c \\in \\{0,0.5,1\\}$ varies. With both $f^{\\text{strong}}(x):= x^T\\beta+\\exp(0.29 x^{(1)}+0.11 x^{(2)}+0.19x^{(3)})$ and $f^{\\text{poor}}(x):= 1$, PPAI CIs outperform WLS under misspecification:
>
> | $c$ | WLS | PPAI ($f^{\\mathrm{poor}}$) | PPAI ($f^{\\mathrm{strong}}$) |
> |---|---|---|---|
> | 0.0 | 0.009 (91%) | 0.009 (91%) | 0.009 (90%) |
> | 0.5 | 0.054 (91%) | 0.031 (89%) | 0.010 (88%) |
> | 1.0 | 0.107 (91%) | 0.063 (89%) | 0.011 (92%) |
>
> **(6) Real data.**
> We conducted a movie-recommendation experiment using *The Movies Dataset* (26M ratings) by using Gemini to predict users' ratings. PPAI substantially improved efficiency and testing power. Further details: [supplement](https://www.dropbox.com/scl/fi/wxt8ry5vb7mmz1qvzj1sg/Rebuttal-Supplement-3.pdf?rlkey=6znvtw8gox5jp613vu3wvsung&st=ypeppq5j&dl=0).
>
> **(7) Sensitivity to exploration.**
> To show strong predictions enable inference with substantially less exploration, we experimented with smaller $\\epsilon$ and a stronger model $f^{\\text{excellent}}(x):=\\exp(0.201x^{(1)}-0.097x^{(2)}+0.398x^{(3)})$ (average CI width with coverage in parentheses):
>
> | $\\epsilon$ | WLS | PPAI ($f^{\\mathrm{excellent}}$) |
> |---|---|---|
> | 0.01 | 0.47 (89%) | 0.03 (88%) |
> | 0.05 | 0.19 (90%) | 0.02 (89%) |
>
> **(8) Other experimental gaps.**
> Supplemental plots ([link](https://www.dropbox.com/scl/fi/wxt8ry5vb7mmz1qvzj1sg/Rebuttal-Supplement-3.pdf?rlkey=6znvtw8gox5jp613vu3wvsung&st=ypeppq5j&dl=0)) illustrate the convergence of $\\hat{\\theta}$ and the variance estimator. Convergence of $\\hat{\\theta}$ implies policy convergence. Bounded moment assumptions hold as both the reward and AI models themselves have bounded moments. We will clarify these points in the revision.
>
> **(9) Interpretation of $\\lambda^\\ast$.**
> Values near $0$ and $1$ indicate uninformative and highly informative prediction quality, respectively. This is why our experiments have $\\lambda^\\ast\\approx 0.115$ for $f^{\\mathrm{poor}}$ and $\\lambda^\\ast\\approx 0.986$ for $f^{\\mathrm{strong}}$. We also clarify that these are Monte Carlo approximations to the oracle optimum, not $\\hat\\lambda\_T$.
>
> **(10) Computational complexity.**
> The per-step complexity of Algorithm 1 is $\\mathcal O(1)$ in the horizon $T$, $\\mathcal O(d^3)$ in the feature dimension, and $\\mathcal O(1)$ in the number of arms $K$ when updating a single arm (or $\\mathcal O(K)$ for all arms).

---

> > ### Author Rebuttal · Reviewer_5HB9 · 2026-04-02
> >
> > Thank you for the response. I do see that the paper has some clear merits—there is value in the problem setting and the theoretical effort is not trivial. However, I remain unconvinced that the combination of IPW and PPI++ with a data-driven $\lambda$ is anything more than a predictable, incremental extension. The components you list (adaptive IPW, i.i.d. unlabeled data, tuning λ) are all borrowed from existing work, and the way they are put together does not strike me as a genuine conceptual synergy. I therefore maintain my original score.

---

> > > ### Author Response · Authors · 2026-04-05
> > >
> > > ## Methodological Synergy
> > >
> > > We thank the reviewer for their response. In the original review, it was noted that a "true methodological synergy" would be demonstrated by "showing that predictions can help reduce the variance of IPW estimators in ways not achievable by either technique alone" or "revealing that the presence of predictions relaxes assumptions needed for adaptive inference (e.g., allowing smaller $\\pi\_{\\min}$)."
> > >
> > > We reiterate that our work demonstrates **both** of these emergent properties, both analytically and empirically:
> > >
> > > - **Analytically:** Our estimating equation shows that strong predictions transform the IPW-weighted term from a raw score into a small residual. Thus, PPAI uses the AI model to ensure these large weights only multiply a residual near zero, reducing the variance impact of small inverse propensity weights and allowing for smaller $\\pi\_{\\min}$.
> > > - **Empirically:** When exploration is nearly absent ($\\epsilon = 0.01$), the inverse propensity weighting in standard adaptive inference causes the variance to explode, resulting in an uninformative interval (Width = 0.47). By contrast, PPAI with a strong AI model produces intervals 15 times smaller (Width = 0.03).
> > >
> > > In practice, a decision-maker can evaluate the quality of $f\_a$ during a brief burn-in period and then set $\\pi\_{\\min}$ accordingly. We believe this directly addresses the reviewer's own criteria for methodological synergy.
> > >
> > > Overall, PPAI achieves a level of statistical precision, accuracy, and low regret unattainable by either PPI++ or adaptive inference techniques in isolation. Specifically, PPAI improves upon standard IPW-based methods by reducing the variance impact of small weights, enabling valid inference with significantly less regret, and improves upon PPI++ by explicitly correcting for the adaptive bias inherent in bandit data.
> > >
> > > ## Theoretical Novelty
> > >
> > > The theoretical contribution lies in the analysis of a **mixed** estimating equation consisting of: (1) an adaptive IPW-weighted labeled component; (2) an i.i.d. unlabeled prediction component; and (3) a data-driven tuning parameter $\\hat{\\lambda}\_T$. This is why Theorem 3.7 requires a martingale CLT argument tailored to the prediction-powered adaptive setting, rather than a direct application of existing results.
> > >
> > > ## Significance
> > >
> > > In response to the "incremental" critique, we emphasize that PPAI enables decision-makers to incur lower regret (as explained above) and that the magnitude of improvement in statistical precision is substantial:
> > >
> > > In our movie-recommendation experiment using Gemini to predict user ratings, PPAI produced confidence intervals roughly $6\\times$ tighter than the WLS baseline while simultaneously improving coverage. PPAI also had vastly superior testing power: for the null hypothesis $H\_0 : \\beta\_0^{(3)} = 0$, PPAI achieved a $90.0\\%$ rejection rate compared to only $19\\%$ for WLS, and for $H\_0:|\\beta\_1^{(1)}|>0.2$, it correctly identified negligibility in $82.5\\%$ of trials where WLS failed entirely ($0\\%$). These results show how PPAI can enable reliable statistical discoveries in settings where standard adaptive methods lack the necessary precision and where PPI is not applicable.

---

### Official Review · Reviewer_CkaD · 2026-03-12

**Soundness:** 3
**Presentation:** 3
**Significance:** 2
**Originality:** 2
**Overall Recommendation:** 3
**Confidence:** 4

**Summary:**

This paper presents a way to do prediction power inference where the labeled data are collected adaptively.

**Compliance With Llm Reviewing Policy:**

Affirmed.

**Final Justification:**

While I feel this paper makes some contribution by intersecting the areas of PPI and inference for adaptively collected data, my concerns about the strength of its assumptions and the technical novelty were largely reinforced by the rebuttal, so I am maintaining my score.

**Key Questions For Authors:**

N/A

**Limitations:**

Yes.

**Strengths And Weaknesses:**

I feel it is a worthy topic to intersect the ideas of prediction powered inference and inference for adaptively collected data and this paper does so in a natural methodological way, but I have some reservations about its assumptions and potential impact. First, though the results don’t seem to be straightforward extensions of results in either branch of literature, it seemed they may be relatively straightforward in light of the union of these two branches of literature, i.e., it’s not clear to me that the theory in this paper isn’t a relatively straightforward combination of existing theory in the two branches. Second, a number of the assumptions seem significantly limiting in practice:
1. The authors assume the covariate distribution for unlabeled, non-adaptively collected data is the same as that for the labeled, adaptively collected data, which seems like it may not hold even in many of the proposed applications in section 1.
2. Assumption 3.1 basically ensures that, at least asymptotically, the data is NOT adaptively collected; this assumption is not always satisfied, e.g., in two-arm bandits with identical reward distributions, many bandit algorithms will not converge to a single policy.
3. Assumption 3.2 precludes any regret-optimal bandit algorithm, despite the paper’s language talking a lot about bandits (I note experiments use epsilon-greedy with epsilon=0.3, suggesting the adaptivity must really be far from converging to an optimal arm in order for the proposed method to perform well).
4. The authors seem to assume that the asymptotic variance they find can be consistently estimated, as this is needed (in addition to the CLT) for asymptotically valid inference. Note this consistency is particularly nontrivial for adaptively collected data.

---

> ### Author Rebuttal · Authors · 2026-03-30
>
> We thank the reviewer for their careful reading and for raising important questions about assumptions and scope. We agree that these points should be clarified more explicitly.
>
> **(i) Covariate shift.**
> We agree that, in practice, unlabeled covariates may come from a different distribution from the experimental contexts. If the shift is known or estimable, the unlabeled term can be reweighted by the density ratio
>
> $$w(\\tilde X\_i)=\\frac{p\_{\\mathrm{labeled}}(\\tilde X\_i)}{p\_{\\mathrm{unlabeled}}(\\tilde X\_i)},$$
>
> leading to the modified estimating equation
>
> $$G\_{a,T}^{\\lambda\_{a,T}}(\\theta) = \\frac1T\\sum\_{t=1}^T\\frac{\\mathbf 1\\{A\_t=a\\}}{\\pi\_t(a)}g(X\_t,Y\_t;\\theta) + \\lambda\_{a,T}\\left[\\frac1N\\sum\_{i=1}^N w(\\tilde X\_i)\\,g(\\tilde X\_i,f\_a(\\tilde X\_i);\\theta) - \\frac1T\\sum\_{t=1}^T\\frac{\\mathbf 1\\{A\_t=a\\}}{\\pi\_t(a)}g(X\_t,f\_a(X\_t);\\theta)\\right].$$
>
> We will add this extension and discussion in the revision.
>
> **(ii) Policy convergence.**
> We agree that policy convergence can fail in edge cases, e.g. ties in the reward distributions. Our claim is not that *all* bandit algorithms always converge, but rather that policy convergence is a standard regularity condition in adaptive inference under misspecification. This assumption appears in Guo & Xu (2025) and related work precisely because inference is much harder if the policy does not stabilize. We will clarify both the scope and limitations of this assumption.
>
> **(iii) Minimum sampling probability and regret-optimality.**
> We agree that the minimum-sampling condition excludes fully regret-optimal exploitation. This is a known tradeoff in adaptive inference: if suboptimal actions are sampled too rarely, one cannot make valid inference on their parameters. Our contribution is not to eliminate this tension, but to improve efficiency within the persistent-exploration regime.
>
> Importantly, the lower variance of PPAI can reduce the amount of exploration needed to achieve a target precision level when the AI model is strong. To illustrate this, we added results for $\\epsilon\\in\\{0.1,0.2,0.3\\}$; each entry reports average width and coverage of 300 nominal $90\\%$ confidence intervals:
>
> | $\\epsilon$ | WLS | PPAI ($f^{\\mathrm{poor}}$) | PPAI ($f^{\\mathrm{strong}}$) |
> |---|---|---|---|
> | 0.1 | 0.188 (87%) | 0.033 (87%) | 0.017 (90%) |
> | 0.2 | 0.133 (86%) | 0.024 (89%) | 0.013 (92%) |
> | 0.3 | 0.111 (90%) | 0.021 (89%) | 0.011 (93%) |
>
> To further illustrate, we also experiment with $\\epsilon=0.01$ and $f^{\\text{excellent}}(x) = \\exp(0.201x^{(1)}-0.097x^{(2)}+0.398x^{(3)})$:
>
> | $\\epsilon$ | WLS | PPAI ($f^{\\mathrm{excellent}}$) |
> |---|---|---|
> | 0.01 | 0.470 (89%) | 0.031 (88%) |
>
> Thus when the AI model is excellent, PPAI retains substantial efficiency gains and competitive coverage, even with very little exploration.
>
> **(iv) Consistent estimation of the asymptotic variance.**
> We agree this should be stated more explicitly. In Appendix C.2 we already give a plug-in estimator in the least-squares case, and the general $Z$-estimation variance estimator is analogous. We will add the general plug-in estimator $\\hat\\Sigma$ and its consistency proof in the revision; the proof follows the same structure as the consistency proof for $\\hat\\lambda\_T$ in Theorem 3.8.
>
> **(v) Why the theory is not a straightforward combination.**
> The novelty is not just that we use tools from two literatures. The key new point is that the asymptotic variance depends jointly on (a) the policy $\\pi$, through the adaptive IPW term, and (b) the predictive error of $f$, through the correction term. This coupling is absent from offline PPI and from labeled-only adaptive inference. In particular:
>
> 1. strong predictions can support the same inferential precision with less exploration;
> 2. the optimal tuning parameter $\\lambda^\\ast$ and its estimate $\\hat\\lambda\_T$ depend on the adaptive policy $\\pi$;
> 3. Theorem 3.7 characterizes asymptotic normality for a mixed adaptive/i.i.d. estimating equation, not a standard labeled-only estimator;
> 4. Algorithm 1 turns prediction-powered inference into an online update procedure rather than an offline batch correction.
>
> We will sharpen this explanation in the revision.

---

> > ### Author Rebuttal · Reviewer_CkaD · 2026-04-02
> >
> > I thank the authors for engaging meaningfully with my comments. Responding to them by in turn:
> > (i) It is good to have this extension to a known shift, but in practice having a covariate shift with known likelihood ratio is I think only slightly more general a setting than having no shift at all. For instance, I still don't think this setting would cover many of the proposed applications in section 1.
> > (ii) Thank you for acknowledging this limitation. I agree this is a common assumption and that it makes inference much easier, but it is nonetheless a significant limiting assumption. I don't agree that "ties in the reward distributions" is an "edge case", but rather a quite common case, at least approximately, whenever the difference between reward distributions is small relative to the sample size, i.e., the signal size is small / hard to find.
> > (iii) Thank you for acknowledging this limitation. I don't agree that "if suboptimal actions are sampled too rarely, one cannot make valid inference on their parameters". For instance, concentration inequalities like Hoeffding's bound can be made uniform over time so that they always cover, even in adaptively collected data and in finite time (this is a standard argument, e.g., in the analysis of UCB, which uses such inference as part of its procedure, and uses the inference's exact coverage in its regret analysis). Such an approach is of course conservative, but shows that there is no information-theoretic barrier that makes it impossible to do valid inference for rarely sampled actions.
> > (iv) Thank you for addressing this issue. I can see that, under the assumption made, there should be no issue with consistent variance estimation.
> > (v) I'm not sure the rebuttals response to this point really contradicts or changes my view. The main point it seems to make is that the novelty arises in the coupling of terms/behavior from both PPI and adaptively collected data, which I agree with. My point was only that the technical tools used to obtain these results may be a straightforward combination of existing technical tools from the two fields.
> >
> > In sum, the rebuttal has largely confirmed my concerns about the paper, so I will maintain my score of a 3.

---

> > > ### Author Response · Authors · 2026-04-05
> > >
> > > We thank the reviewer for their continued engagement. We would like to offer these clarifications:
> > >
> > > **(i)** We emphasize that our extension also applies if the density ratio is estimable, i.e., $\\hat w(\\tilde X\_i) \\xrightarrow{p} \\frac{p\_{\\text{labeled}}(\\tilde X\_i)}{p\_{\\text{unlabeled}}(\\tilde X\_i)}$. While estimating this shift can be challenging, without some estimable structure linking the two distributions (e.g., via the density ratio), incorporating unlabeled data into a consistent estimator is generally intractable.
> > >
> > > **(ii)** We note that it is straightforward to design policies that converge in practice. As a simple principle, a policy will converge if it is based on summary statistics (like empirical means) that reach a limit where the policy mapping is continuous (Guo & Xu, 2025). This covers a broad class of commonly used algorithms, like $\\epsilon$-greedy, UCB, and Thompson Sampling (with minimum sampling probabilities).
> > >
> > > We also appreciate the reviewer's point about regimes where reward differences are small relative to the sample size. We agree that stability may not be reached in finite samples if there is a near-tie. However, PPAI is more robust to this instability than competing methods: in the estimator of Guo & Xu, these fluctuating weights $\\pi\_t(a)$ would divide the scores $g(X\_t, Y\_t; \\theta)$, causing significant instability in the estimator, whereas in PPAI with a strong AI model ($\\lambda \\approx 1$), the weights divide the tiny residuals $(g(X\_t, Y\_t; \\theta) - g(X\_t, f\_a(X\_t); \\theta))$, allowing PPAI to maintain precision even when $\\pi\_t$ has not yet converged.
> > >
> > > **(iii)** We agree that concentration bounds like Hoeffding's provide valid coverage. However, there is a categorical difference between those conservative bounds and our goal of asymptotically exact inference: because Hoeffding-style bounds are so conservative, they are too loose for the precise hypothesis testing and interval estimation required in scientific studies.
> > >
> > > The trade-off between statistical inference and regret minimization is well-recognized: if one only prioritizes regret, a suboptimal arm may only be pulled $O(\\log T)$ times, but valid exact inference requires more frequent sampling to ensure the estimator achieves the $\\sqrt{T}$-consistency and asymptotic normality needed for precise intervals ([1], [2], [3]). Thus, persistent exploration assumptions are typically required in adaptive inference, and a minimum sampling probability is a standard such condition under misspecification. Moreover, our framework demonstrates that AI predictions allow practitioners to significantly relax this exploration floor (hence incur less regret) while maintaining exact valid inference.
> > >
> > > **(v)** To clarify our technical challenge, our estimator solves $G\_T^{\\lambda\_T}(\\hat\\theta\_T)=0$, with
> > >
> > > $$G\_T^{\\lambda\_T}(\\theta) = \\frac1T\\sum\_{t=1}^T\\frac{\\mathbf 1\\{A\_t=a\\}}{\\pi\_t(a)}g(X\_t,Y\_t;\\theta) + \\lambda\_T\\left[\\frac1N\\sum\_{i=1}^N g(\\tilde X\_i,f(\\tilde X\_i);\\theta) - \\frac1T\\sum\_{t=1}^T\\frac{\\mathbf 1\\{A\_t=a\\}}{\\pi\_t(a)}g(X\_t,f(X\_t);\\theta)\\right].$$
> > >
> > > Hence the challenge is to analyze a *mixed* estimating equation with: (a) an adaptive IPW-weighted labeled component, (b) an i.i.d. unlabeled prediction component, and (c) a data-driven tuning parameter $\\hat\\lambda\_T$. This is why Theorem 3.7 requires a martingale CLT argument tailored to the prediction-powered adaptive setting, rather than a straightforward application of existing tools.
> > >
> > > [1] Simchi-Levi, D., \& Wang, C. (2025). Multi-armed bandit experimental design: Online decision-making and adaptive inference. Management Science, 71(6), 4828–4846.
> > >
> > > [2] Zhang, K. W., Janson, L., \& Murphy, S. A. (2020). Inference for batched bandits. Advances in Neural Information Processing Systems, 33, 9818–9829.
> > >
> > > [3] Zhang, K. W., Janson, L., \& Murphy, S. A. (2021). Statistical inference with M-estimators on adaptively collected data. Proceedings of the 35th International Conference on Neural Information Processing Systems (NIPS '21), 7460–7471.

---

### Official Review · Reviewer_xSRz · 2026-03-13

**Soundness:** 3
**Presentation:** 2
**Significance:** 3
**Originality:** 3
**Overall Recommendation:** 3
**Confidence:** 2

**Summary:**

This paper proposes Prediction-Powered Adaptive Inference (PPAI) for contextual bandits with access to unlabeled contexts and pretrained reward predictors. The main idea is to extend prediction-powered inference from the i.i.d. offline setting to adaptively collected bandit data by combining inverse propensity weighting with a prediction-powered correction term in a single estimating equation. It analyzes an important concept: how to use black-box predictions safely for valid inference under adaptive data collection.

**Compliance With Llm Reviewing Policy:**

Affirmed.

**Final Justification:**

The rebuttal helps clarify the paper, but I am still concerned about the paper’s conceptual positioning and empirical scope. My overall assessment and score remain unchanged.

**Key Questions For Authors:**

1. Can the paper better clarify its relation to active statistical inference, and explain more explicitly why the present setting is related but distinct?
2. Can the empirical section be expanded beyond the current synthetic setup?

**Limitations:**

see above.

**Strengths And Weaknesses:**

Strengths:
S1: The paper addresses a meaningful and timely problem: inference in adaptive experiments with scarce labeled data but abundant unlabeled contexts and external predictive models.
S2: The methodological contribution is substantive. The proposed estimating equation is a nontrivial extension of PPI to the adaptive bandit setting, rather than a straightforward plug-in.

Weaknesses:
W1: Figure 1 has relatively low information density. It conveys the workflow, but it does not reveal the key structure of the estimator or the role of inverse propensity weighting and prediction correction.
W2: Algorithm 1 is difficult to parse in the main text. Maintaining many intermediate quantities $B, C, D, E, F$ may be and not reader-friendly. The assumptions appear standard for adaptive Z-estimation, but Assumptions 3.5 and 3.6 are presented in a rather heavy way.
W3: The empirical evaluation is fairly limited.

---

> ### Author Rebuttal · Authors · 2026-03-30
>
> We thank the reviewer for the thoughtful comments. We agree that the paper would benefit from improved presentation, empirical breadth, and discussion of related work.
>
> **(i) Presentation clarity.**
> We will revise Figure 1 and Algorithm 1. In particular, we will make Figure 1 explicitly separate the three pieces of the estimating equation,
>
> $$\\underbrace{\\frac1T\\sum\_{t=1}^T\\frac{\\mathbf 1\\{A\_t=a\\}}{\\pi\_t(a)}g(X\_t,Y\_t;\\theta)}\_{\\text{adaptive IPW-labeled term}} + \\lambda\_T\\left(\\underbrace{\\frac1N\\sum\_{i=1}^N g(\\tilde X\_i,f(\\tilde X\_i);\\theta)}\_{\\text{unlabeled prediction term}} - \\underbrace{\\frac1T\\sum\_{t=1}^T\\frac{\\mathbf 1\\{A\_t=a\\}}{\\pi\_t(a)}g(X\_t,f(X\_t);\\theta)}\_{\\text{labeled prediction correction}}\\right),$$
>
> so that the role of inverse propensity weighting and the role of prediction calibration are visually clear.
>
> **(ii) Relation to active statistical inference (ASI).**
> We agree this comparison should be made more explicit. The key distinction is not only that our setting uses unlabeled data and predictions; it is that the *source of adaptivity is fundamentally different*. In ASI, the learner adaptively chooses which *contexts* to label under a labeling budget, with the goal of maximizing inferential efficiency. In our problem, the learner adaptively chooses *actions* $A\_t$ for incoming contexts $X\_t$ under a reward-seeking bandit policy $\\pi\_t$, so the labels arise from reward optimization rather than inferential query selection. This creates a different stochastic structure: our labeled sample is biased by action-dependent adaptive assignment, which is why the prediction-powered correction must be combined with inverse propensity weighting. We will add this distinction to the related-work section.
>
> **(iii) Additional empirical validation.**
> To strengthen practical relevance, we conducted a movie-recommendation experiment based on *The Movies Dataset* (26M ratings), using Gemini to predict ratings. Since the full rating matrix is unobserved, we construct a surrogate ground-truth environment $y(u,a)$ using a Random Forest regressor. We use a 5-dimensional context $x$ consisting of standardized ratings on five representative movies, and let the action set be $\\mathcal A=\\{0,1\\}$ corresponding to recommending *X-Men* or *Good Will Hunting*. We fit the linear reward model
>
> $$y=\\mathbf 1\\{A\_t=0\\}\\beta\_0^\\top x+\\mathbf 1\\{A\_t=1\\}\\beta\_1^\\top x$$
>
> of ratings as a function of user context and simulate an online experiment with $T=1000$ and $\\epsilon=0.1$. At each round, we also sample 10 unlabeled contexts and query Gemini for predicted ratings $f_a(\\tilde X)$. PPAI substantially improves efficiency:
>
> | Target | WLS Var | PPAI Var | WLS CI (Cov.) | PPAI CI (Cov.) |
> |---|---|---|---|---|
> | $\\beta\_0^{(3)}$ | 0.170 | 0.003 | 1.23 (82.6%) | 0.21 (95.1%) |
> | $\\beta\_1^{(1)}$ | 0.304 | 0.002 | 1.44 (80.5%) | 0.22 (99.0%) |
>
> Thus PPAI achieves roughly $6$--$15\\times$ smaller variance and about $3\\times$ tighter confidence intervals while improving coverage. It also improves testing power: for $H\_0:\\beta\_0^{(3)}=0$, PPAI rejects in $90.0\\%$ of trials versus $19\\%$ for WLS; for the negligibility test $H\_0:|\\beta\_1^{(1)}|>0.2$, PPAI correctly identifies negligibility in $82.5\\%$ of trials versus $0\\%$ for WLS.

---

> > ### Author Rebuttal · Reviewer_xSRz · 2026-04-04
> >
> > Thank you for your response and clarification.

---

> > > ### Author Response · Authors · 2026-04-05
> > >
> > > We appreciate your confirmation that our previous response and new empirical results fully resolved your concerns. We kindly ask you to consider raising your score to reflect this updated assessment. To briefly reiterate the core novelty and contributions of our work:
> > >
> > > - **Theoretical Results:** We establish asymptotic normality for the PPAI estimator via a martingale CLT argument tailored to our mixed estimating equation, and we prove that our data-driven parameter $\\hat{\\lambda}\_T$ achieves the optimal weighting of predictions.
> > > - **Reduced Regret:** Our framework enables valid, high-precision inference even with minimal exploration, allowing decision-makers to achieve statistical guarantees while incurring significantly less regret.
> > > - **Computational Efficiency:** Our estimation procedure can be updated in an online fashion, making PPAI a computationally efficient solution for real-time adaptive experiments.
> > > - **Substantial Empirical Gains:** We demonstrate through both our synthetic simulations and our movie-recommendation experiment with Gemini that PPAI yields a substantial variance reduction and vastly superior testing power compared to the labeled-only baseline.
> > >
> > > Lastly, we make a small correction to our description of active statistical inference: the presence of unlabeled data and predictions is a *similarity*, not a difference, between PPAI and ASI; the difference is in the source of adaptivity, as explained.

---

### Official Review · Reviewer_jCTC · 2026-03-14

**Soundness:** 4
**Presentation:** 4
**Significance:** 2
**Originality:** 3
**Overall Recommendation:** 4
**Confidence:** 4

**Summary:**

This paper explores how machine learning predictions can be used to derive tighter confidence intervals when estimating parameterized reward functions in the contextual bandits setting. The problem setup blends the standard contextual bandits setting with a PPI-style adaptation. In particular, at each time step $T$, nature reveals a co-variate $X_T$, and the decision maker subsequently samples from a distribution over actions and selects a corresponding arm. A reward $Y_T$ is then observed. In addition, the decision maker also has access to a large set of unlabeled covariates and a machine learning model of unknown quality that predicts the reward for each action.

Within this setup, the authors propose an extension of the PPI++-style estimator that incorporates the ML model and an unlabeled dataset to produce better estimates of the reward parameters. The estimator is an adaptation of the PPI/PPI++ estimator that accounts for the adaptive nature of the setting by incorporating an ipw-style reweighting of the correction terms. Like PPI++, the authors include an adaptive parameter $\lambda$ that weights the predictions based on their quality. The authors show that their estimator is asymptotically normally distributed and characterize its variance. They further derive the optimal $\lambda$ to minimize the variance of the resulting estimator and present simulation results demonstrating its coverage probability.

**Compliance With Llm Reviewing Policy:**

Affirmed.

**Key Questions For Authors:**

1. Can you explain why PPAI(f^{poor}) performs relatively well compared to WLS, even though the predictions are bad? The gap between PPAI(f^{poor}) and PPAI(f^{strong}) is much smaller than PPAI(f^{poor}) and WLS. I would have assumed that this would make lambda close to zero, so we recover the WLS estimator.

2. Can you explain more explicitly your work’s relation to (Guo & Xu 2025)? What aspects of your theoretical results (e.g., Theorem 3.7) are similar vs different compared to this work?

**Limitations:**

See comments.

**Strengths And Weaknesses:**

Strengths
-----------------------

- The paper is technically sound. The asymptotic distribution of the estimator looks correct and seems to follow similarly to existing results on contextual bandits (Guo & Xu 2025). The assumptions are fairly standard assumptions for Z-estimation (i.e., Vaart 1998).

- Overall, the paper is well written and nicely laid out. The prior work on contextual bandits and existing estimators is explained with sufficient detail, and the assumptions are appropriately justified. There are a few areas (e.g., Assumption 3.5, Algorithm 1) where I believe the formatting could be improved to include more space and thus increase readability.


Comments - Questions
-----------------------------------
1. I believe this paper represents a nice adaptation of the PPI framework to the setting of contextual bandits. However, the paper feels like it is missing an appropriate justification for why this setting is important to study in practice. There is a brief mention of its relevance to clinical trials, but I would have liked to see this example expanded upon and/or more examples. Along these lines, it would have been nice to see these methods applied to real-world data (like in the original PPI setting) to provide further support for the fact that these methods are useful in practice.


2. Overall, I believe this paper contains sufficient original elements to justify the result’s novelty. The authors provide the first results on statistical inference in the contextual bandit setting with ML predictions. The estimator has a novel form, though it is clearly built from an amalgamation of existing ideas (IPW weighting combined with ppi style correction). The analysis of Theorem 3.7 appears to be non-trivial; however, I am not sure the extent to which it follows from existing theorems of (Guo & Xu 2025).

---

> ### Author Rebuttal · Authors · 2026-03-30
>
> We thank the reviewer for the positive assessment and for the helpful suggestions regarding practical motivation, empirical relevance, and positioning relative to Guo & Xu (2025).
>
> **(i) Real-world justification.**
> A key motivation of our work is that in many adaptive experiments, labels are expensive or slow to collect, while AI predictions are cheap and abundant. Concretely:
>
> 1. in clinical trials, pretrained EHR models can predict health trajectories and reduce ethical/financial costs of poor treatment assignment;
> 2. in education, engagement-based models can forecast student performance and support earlier intervention;
> 3. in recommendation systems, foundation models can cheaply predict user interest from logs before new online feedback arrives.
>
> Our paper gives, to our knowledge, the first rigorous framework for using such *external predictions* in *adaptive* experiments while preserving valid inference under misspecification and non-i.i.d. data collection.
>
> **(ii) Real-data experiment.**
> To strengthen practical relevance, we conducted a movie-recommendation experiment based on *The Movies Dataset* (26M ratings), and we used Gemini to predict ratings. Since the full rating matrix is unobserved, we construct a surrogate ground-truth environment $y(u,a)$ using a Random Forest regressor. We use a 5-dimensional context $x$ consisting of standardized ratings on five representative movies, and let the action set be $\\mathcal A=\\{0,1\\}$ corresponding to recommending *X-Men* or *Good Will Hunting*. We fit the linear reward model
>
> $$y=\\mathbf 1\\{A\_t=0\\}\\beta\_0^\\top x+\\mathbf 1\\{A\_t=1\\}\\beta\_1^\\top x,$$
>
> of ratings as a function of user context and simulate an online experiment with $T=1000$ and an $\\epsilon$-greedy policy, where $\\epsilon=0.1$. At each round, we also sample 10 unlabeled contexts and query Gemini for predicted ratings $f_a(\\tilde X)$. PPAI substantially improves efficiency:
>
> | Target | WLS Var | PPAI Var | WLS CI (Cov.) | PPAI CI (Cov.) |
> |---|---|---|---|---|
> | $\\beta\_0^{(3)}$ | 0.170 | 0.003 | 1.23 (82.6%) | 0.21 (95.1%) |
> | $\\beta\_1^{(1)}$ | 0.304 | 0.002 | 1.44 (80.5%) | 0.22 (99.0%) |
>
> Thus PPAI achieves roughly $6$--$15\\times$ smaller variance and about $3\\times$ tighter confidence intervals while improving coverage. It also improves testing power: for $H\_0:\\beta\_0^{(3)}=0$, PPAI rejects in $90.0\\%$ of trials versus $19\\%$ for WLS; for the negligibility test $H\_0:|\\beta\_1^{(1)}|>0.2$, PPAI correctly identifies negligibility in $82.5\\%$ of trials versus $0\\%$ for WLS.
>
> **(iii) Why does PPAI still help when $f^{\\mathrm{poor}}$ is weak?**
> In our experiments with $f^{\\text{poor}}$, the optimal weight is not zero: we estimate $\\lambda^\\ast\\approx 0.115$. So PPAI does *not* collapse to WLS. Intuitively, "weak" does not mean "completely useless": the large unlabeled sample can still contribute some signal, and the right choice of $\lambda$ shrinks this contribution to a safe level.
>
> **(iv) Relation to Guo & Xu (2025).**
> Guo & Xu solve a labeled-only adaptive estimating equation
>
> $$G\_T(\\hat\\theta\_T)=0, \\qquad G\_T(\\theta)=\\frac1T\\sum\_{t=1}^T\\frac{\\mathbf 1\\{A\_t=a\\}}{\\pi\_t(a)}\\,g(X\_t,Y\_t;\\theta).$$
>
> Our estimator instead solves $G\_T^{\\lambda\_T}(\\hat\\theta\_T)=0$, where
>
> $$G\_T^{\\lambda\_T}(\\theta) = \\frac1T\\sum\_{t=1}^T\\frac{\\mathbf 1\\{A\_t=a\\}}{\\pi\_t(a)}g(X\_t,Y\_t;\\theta) + \\lambda\_T\\left[\\frac1N\\sum\_{i=1}^N g(\\tilde X\_i,f(\\tilde X\_i);\\theta) - \\frac1T\\sum\_{t=1}^T\\frac{\\mathbf 1\\{A\_t=a\\}}{\\pi\_t(a)}g(X\_t,f(X\_t);\\theta)\\right].$$
>
> The novelty is not merely an extra term. Once this prediction-powered correction is introduced, the inferential object itself changes: the asymptotic analysis must jointly handle (a) an adaptive IPW-weighted labeled component, (b) an i.i.d. unlabeled prediction component, and (c) the stochastic estimation error of $\\hat\\lambda\_T$. This is why Lemma B.4 is substantially more involved than the labeled-only setting: deriving the asymptotic distribution of $\\sqrt{T}G\_T^{\\lambda\_T}(\\theta^\\ast)$ requires combining a martingale CLT for the adaptive term with an independent-sample limit for the unlabeled term and handling the randomness of $\\hat\\lambda\_T$. Our Theorem 3.7 reduces to Guo & Xu's result when $\\lambda=0$, but Theorem 3.8 and Corollary 3.10 (optimal/safe tuning) have no counterpart in the labeled-only framework.

---

> > ### Author Rebuttal · Reviewer_jCTC · 2026-04-06
> >
> > Thank you for all the answers. I am satisfied with almost everything, expect the real-world applications. The reason that real-world applications are important in this context is that the theoretical properties of PPI are not profoundly deep. The main advantage of it is its empirical performance in real-data. I understand that you now have a real-data application but I believe that adding this would need a significant change in the paper, more than what is expected after a rebuttal.
> >
> > I remain positive about the paper and I won't object acceptance.

---

> > > ### Author Response · Authors · 2026-04-06
> > >
> > > We thank the reviewer for the positive assessment and the constructive dialogue. We agree that demonstrating utility on real-world data is an important justification for the framework's significance. To address the concern regarding the scope of such an update, we clarify that the movie-recommendation experiment (described above) is already completed and written into a two-page report; this content is prepared as an addition to the appendix and will include a detailed summary integrated into the main body. We also highlight that these results do not change the core of the paper, but rather corroborate our theorems and further support the simulation results. In summary, our real-data results are fully completed and required minimal modification to the manuscript.
> > > Given these clarifications, we kindly ask the reviewer to consider if an increase in the significance score is warranted.

---

### Decision · Program_Chairs · 2026-04-30

**Decision:**

Accept (regular)

**Comment:**

The paper proposes PPAI, a statistical inference framework for bandit data that effectively leverages unlabeled offline datasets, pretrained AI models, and labeled but non-i.i.d. bandit data. The asymptotic distribution of the proposed estimator is derived, which enables hypothesis testing and confidence interval construction. The method provides a safe and efficient way to conduct valid inference using black-box predictions, and coverage probabilities are demonstrated through experiments.

However, there are concerns regarding the practical relevance and the gap between theory and experiments. Some reviewers also find that the theoretical contribution is predictable and incremental relative to existing works. The authors are strongly encouraged to include a real-data analysis in their next revision, reduce the theory–experiment gap, and more clearly highlight their theoretical contributions.